# Robustness Auditing for Linear Regression: To Singularity and Beyond

**Ittai Rubinstein & Samuel B. Hopkins**
Department of Electrical Engineering and Computer Science
Massachusetts Institute of Technology
Cambridge, MA, USA
`{ittair,samhop}@mit.edu`

## Abstract

It has recently been discovered that the conclusions of many highly influential econometrics studies can be overturned by removing a very small fraction of their samples (often less than $0.5\%$). These conclusions are typically based on the results of one or more Ordinary Least Squares (OLS) regressions, raising the question: given a dataset, can we certify the robustness of an OLS fit on this dataset to the removal of a given number of samples? Brute-force techniques quickly break down even on small datasets. Existing approaches which go beyond brute force either can only find candidate small subsets to remove (but cannot certify their non-existence) Broderick et al. (2020); Kuschnig et al. (2021), are computationally intractable beyond low dimensional settings Moitra & Rohatgi (2022), or require very strong assumptions on the data distribution and too many samples to give reasonable bounds in practice Bakshi & Prasad (2021); Freund & Hopkins (2023). We present an efficient algorithm for certifying the robustness of linear regressions to removals of samples. We implement our algorithm and run it on several landmark econometrics datasets with hundreds of dimensions and tens of thousands of samples, giving the first non-trivial certificates of robustness to sample removal for datasets of dimension $4$ or greater. We prove that under distributional assumptions on a dataset, the bounds produced by our algorithm are tight up to a $1 + o(1)$ multiplicative factor.

## 1 Introduction

Consider a supervised learning problem with feature vectors $X_1, \ldots, X_n \in \mathbb{R}^d$ and labels $Y_1, \ldots, Y_n \in \mathbb{R}$, to which we fit a model $f : \mathbb{R}^d \to \mathbb{R}$. Robustness auditing addresses the question:

*How would $f$ have differed if we had been missing a small fraction of the data?*

We study this question in the context of ordinary least squares (OLS) linear regression, where $f(X) = \langle \beta, X \rangle$ is the linear function minimizing the mean squared error $\frac{1}{n} \sum_{i \le n} (f(X_i) - Y_i)^2$. We focus on OLS for two reasons. First, OLS is a statistics workhorse, with widespread use across economics, social science, finance, machine learning, and beyond. Second, its relative simplicity affords us the opportunity to design algorithms with provable guarantees, and offers a stepping stone to more complex models (logistic regression, kernel methods, neural networks).

**Problem 1** (Robustness Auditing for OLS Regression). *Given a linear regression instance* $(X_1, Y_1), \ldots, (X_n, Y_n) \in \mathbb{R}^{d+1}$, *a direction* $e \in \mathbb{R}^d$, *and an integer* $k \le n$, *what is*

$$\Delta_k(e) = \max_{\substack{S \subseteq [n] \\ |S| = n-k}} \langle \beta_{[n]} - \beta_S, e \rangle \tag{1}$$

*where for* $T \subseteq [n]$, $\beta_T \in \mathbb{R}^d$ *denotes the vector of OLS coefficients for the dataset* $\{(X_i, Y_i)\}_{i \in T}$?

*In particular, for a threshold* $\theta \in \mathbb{R}$ *what is the minimal number of removals* $k_\theta(e)$ *for which* $\Delta_k(e) > \theta$?

**Context and Applications for Robustness Auditing**   Problem 1 was introduced in this form by Broderick, Giordano, and Meager Broderick et al. (2020), who use a heuristic algorithm, AMIP, to identify very small subsets of landmark datasets from econometrics which can be removed to overturn important conclusions of the respective studies Finkelstein et al. (2012); Angelucci & De Giorgi (2009); often this can be achieved by removing less than $0.5\%$ of a dataset. Researchers have subsequently used AMIP to audit a wide range of recent studies in economics Martinez (2022); Di & Xu (2022); Davies et al. (2024); Zachmann et al. (2023); Burton & Roach (2023); Beuermann et al. (2024); Bondy et al. (2023). Subsequent algorithmic works Kuschnig et al. (2021); Moitra & Rohatgi (2022); Freund & Hopkins (2023) develop additional algorithms for auditing a similar notion of robustness; we discuss prior work in detail below.

It is important that, similar to these prior works, we focus on robustness to a shift of $\beta$ in only a single user-specified direction $e$. This is because the main conclusion of a regression is often determined by the projection of its result on a particular axis. For instance:

ROBUSTNESS OF PARAMETER ESTIMATE   A researcher may want to estimate the correlation $\langle \beta, e \rangle$ between a specific explanatory variable and a target variable, controlled for additional factors, where $e$ is the indicator vector corresponding to the explanatory variable. Moreover, the sign and statistical significance of $\langle \beta, e \rangle$ is often of greatest interest.

This correlation can have a causal interpretation. For instance, in a randomized control trial, where $e$ is the indicator for the treatment variable, the projection $\langle \beta, e \rangle$ can be used to estimate the "average treatment effect" (ATE) of a new drug or policy on the outcome $Y$, while including the control variables in the regression can help reduce the variance of this estimate. Even more complex causal inferences (e.g., instrumental variables regression) can often be decomposed into a small number of OLS regressions, where the result of the causal inference depends on a single coefficient from the result of each regression.

Conclusions from a study where this shift $\Delta_k(e)$ is large when $k \ll n$ are therefore driven by a small number of data points, meriting at minimum reinspection of a dataset, and perhaps casting doubt on generalizability. In many real world regressions, the sign of $\langle \beta, e \rangle$ is not robust to a small number of removals, even though it is statistically significant Broderick et al. (2020). *Non-existence* of a small set of highly influential samples indicates robustness of the measured effect to an interesting class of distribution shifts – any removal of a small fraction of the population.

DATA ATTRIBUTION   Suppose that instead of looking for the effect of a particular feature on the label $Y$, instead we use the linear model $f$ to predict the label of a fresh point $X_{\text{new}}$, and we want to identify what part of the training data led to the prediction that $Y_{\text{new}} \approx f(X_{\text{new}})$. Following the counterfactual formulation of this data attribution problem from Koh & Liang (2017); Ilyas et al. (2022), we arrive again at robustness auditing: using $e = X_{\text{new}}$, we can find the smallest set of whose removal would significantly shift $f(X_{\text{new}})$. We can evaluate the *brittleness* of the prediction by measuring the size of the smallest set of samples we could remove to cause $f(X_{\text{new}})$ to cross a decision boundary.

**Intractability, Heuristics and Upper Bounds**   As soon as $k$ exceeds single digits, robustness auditing by brute-force search over all $|S| = n - k$ takes times $\binom{n}{k}$, which is computationally intractable.

The works Broderick et al. (2020); Kuschnig et al. (2021) relax the goal to finding upper bounds on $k_\theta(e)$, for which they use greedy/local search algorithms. This approach leaves open the risk that $k_\theta(e)$ might be much smaller than the upper bound suggests. Indeed, later experiments by us (see Figure 1), Moitra & Rohatgi (2022), and Freund & Hopkins (2023) uncover numerous real-world examples where local search techniques give loose upper bounds.

Following Moitra & Rohatgi (2022), we aim for algorithms which provide *unconditionally valid upper and lower bounds* on $k_\theta(e)$ for every dataset $\{(X_i, Y_i)\}_{i \in [n]}$, and which return *high-quality* upper/lower bounds (as close to matching each other as possible) under reasonable assumptions on $\{(X_i, Y_i)\}_{i \in [n]}$. Prior approaches to go beyond greedy algorithms and provide lower bounds on $k_\theta(e)$ Klivans et al. (2018); Bakshi & Prasad (2021); Moitra & Rohatgi (2022); Freund & Hopkins (2023) so far don't yield results in practice for datasets of dimension 4 or greater, due to running times which scale exponentially in $d$ and/or prohibitively strong assumptions on $X_1, \ldots, X_n$.

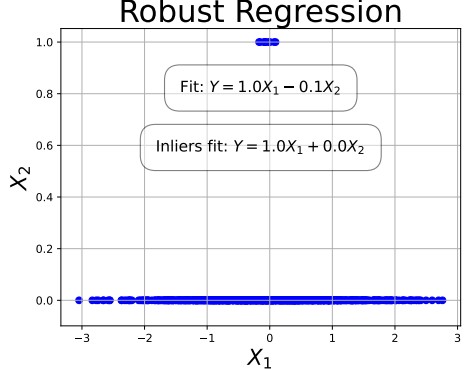
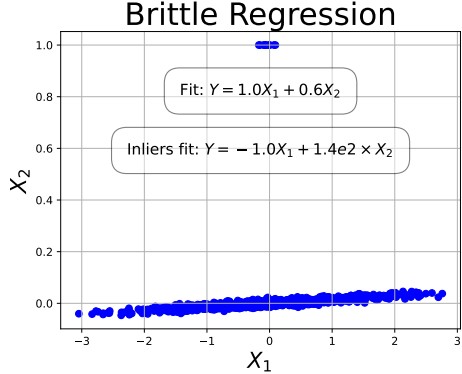

(a) A robust dataset whose robustness cannot be certified by continuous algorithms such as Bakshi & Prasad (2021); Freund & Hopkins (2023).

(b) A small perturbation on 1a creates brittleness which is not detected by AMIP (Broderick et al. (2020)) or Kuschnig et al. (2021).

Figure 1: A comparison of two regressions. Figure 1a shows a regression from a main variable $X_1$ and an indicator variable $X_2$ which is set to 1 on only a very small subset of the samples ($\approx 1\%$). The labels $Y$ are drawn from a normal distribution around $X_1$, resulting in an OLS vector $\beta$ whose first coefficient is positive and whose sign is robust to removing $\geq 15\%$ of its samples. As shown in Claim G.1, we can perturb only the $X_2$ values of the inlier samples to produce an extremely brittle regression (Figure 1b). Because most current approaches to estimating the robustness of a linear regression produce outputs which vary smoothly with the input dataset (such as gradient descent Broderick et al. (2020), semidefinite programming Bakshi & Prasad (2021), or spectral decompositions Freund & Hopkins (2023)), they cannot differentiate between these cases.

## 1.1 OUR CONTRIBUTIONS

We present and analyze two new algorithms, ACRE and OHARE , which provide *lower bounds* on $k_\theta(e)$, *certifying robustness* of OLS regression. Our algorithms provide the first nontrival bounds on the number of samples which must be removed to flip the signs of important parameter estimates for benchmark datasets studied in prior work (including regressions with dimension $d > 200$ and with $n > 30000$ samples). We evaluate our algorithms experimentally and in theory.

ACRE (**A**lgorithm for **C**ertifying **R**obustness **E**fficiently) takes as input a dataset $X \in \mathbb{R}^{n \times d}, Y \in \mathbb{R}^n$ and a vector $e \in \mathbb{R}^d$, runs in time $O\left(n^2 d + n^2 \log(n)\right)$, and outputs a set of upper and lower bounds $U, L \in \mathbb{R}^n$ on the removal effects such that $U_k \geq \Delta_k(e) \geq L_k$. In particular, this runtime avoids exponential dependence on $k$, $d$, and $n$. The upper and lower bounds ACRE provides are valid even making *no assumptions whatsoever* on $X$ and $Y$. When $X$ and $Y$ are drawn from a sufficiently "nice" distribution (such as a linear model with subgaussian features and labels), then the bounds are also *good*, meaning that the upper and lower bounds are close to matching (see Theorem 1.2). We present ACRE in Section 3.

However, there is a very important class of datasets on which ACRE still provides very loose bounds: those using one-hot encodings (also known as indicator or dummy variables) to express categorical features. Even though one-hot encoded datasets can yield robust regressions, certifying this is challenging because of singularities which emerge in the covariance when samples are removed (see Figure 1).

OHARE (**O**ne-**H**ot aware **A**lgorithm for certifying **R**obustness **E**fficiently) extends ACRE to certify robustness of datasets with a mix of continuous and categorical features. It uses dynamic programming in conjunction with a fine-grained linear-algebraic analysis of the contribution of categorical features. OHARE takes as input a dataset $X \in \mathbb{R}^{n \times (d+m)}$ (where $m$ of the features represent a one-hot encoding and the other $d$ are some continuous features) and a direction of interest $e \in \mathbb{R}^d$ within the continuous features, runs in time $O\left(n^2(d+m) + n^2 m \log(n)\right)$ and outputs upper and lower bounds on the removal effect along the axis $e$. We present an overview of OHARE in Section 3.5, with a detailed description deferred to Appendix F.

The most important property of both ACRE and OHARE is that the bounds they produce are *valid* regardless of any assumptions on the dataset:

**Theorem 1.1** (Correctness). *Given $e, X$, and $Y$, ACRE and OHARE output lists of upper/lower bounds $U, L \in \mathbb{R}^n$ s.t.*

$$\forall k \in [n] \quad L_k \leq \Delta_k(e) = \max_{\substack{S \subseteq [n] \\ |S|=n-k}} \langle \beta - \beta_S, e \rangle \leq U_k$$

The proof of Theorem 1.1 is given in Section 3.3 for ACRE and Section A for OHARE . On its own, Theorem 1.1 says little about the usefulness of the upper and lower bounds $L_k, U_k$. We provide two types of evidence that the bounds produced by ACRE and OHARE are interesting. First, we demonstrate on real-world econometric datasets studied in prior work on robustness auditing that ACRE and OHARE produce significantly better lower bounds bounds than were previously known (see Figure 2 and Table 1). Second, we prove that both ACRE and OHARE produce nearly-matching upper and lower bounds under relatively mild distributional assumptions on $X$ and $Y$, for the interesting range of $k$.

INTERESTING VALUES OF $k$   For a direction $e$, we emphasize two values of $k$: $k_{2\sigma}(e)$, the number of removals needed to shift $\langle \beta, e \rangle$ outside its 95% confidence interval, and $k_{\text{sign}}(e) = k_{\langle \beta, e \rangle}$, the number to flip the sign of $\langle \beta, e \rangle$. In the parameter estimation setting, $\langle \beta, e \rangle$ and $2\sigma$ are often of similar magnitude, because rejecting a null hypothesis often involves placing the estimator $\langle \beta, e \rangle$ in a confidence interval which does not contain 0.

**Experimental Results**   For comparison with prior works, we focus our experiments on $k_{\text{sign}}(e)$. We provide lower bounds on $k_{\text{sign}}(e)$ for benchmark datasets drawn from important studies in economics and social sciences, first investigated in the context of robustness auditing by Broderick et al. (2020).

We study real-world datasets corresponding to each of the parameter estimation use-cases listed above: *Nightlights* Martinez (2022) (correlation controlled for additional features), *Cash Transfer* Angelucci & De Giorgi (2009) (randomized control trial), and the *Oregon Health Insurance Experiment (OHIE)* Finkelstein et al. (2012) (IV regression), with 14 distinct linear or instrumental-variables regressions drawn from the corresponding papers, all of which appeared in top econometrics journals. In many cases, our lower bounds match known upper bounds up to a factor of 2 or 3, where no nontrivial lower bounds were previously known; see Figure 2 and Table 1.

To illustrate, consider Nightlights: Martinez (2022) studies whether democractic countries publish more accurate economic growth estimates than dictatorships, after controlling for variables like regional stability and wealth in natural resources. Martinez formulates this as a linear regression with dimension $d = 209$, over 200 of which correspond to one-hot encoded categorical variables. The sign and statistical significance of a single coordinate of $\beta$ govern the conclusion of the study. Algorithms from prior work find a subset of 2.8% of the samples which can be removed to reverse Martinez (2022)'s main conclusion, but prior algorithms could not rule out the existence of much smaller subsets. Our algorithm OHARE provides a certificate that no subset of $\leq 0.7\%$ of the samples would reverse the study's conclusion. See Table 1 and Appendix C for our results on Nightlights and the 13 other regressions we audit, and a detailed discussion of our experiments.

An implementation of our algorithms is available via Github. Our implementation is efficient enough to run our algorithms with $n$ up to $3 \times 10^4$ and $d$ up to $10^3$ on a single CPU core with $< 64GB$ of RAM. The main bottleneck in practice is storing 3 floating-point matrices of size $n \times n$ each.

**Theory for ACRE**   Without some assumptions on $X, Y$, finding matching upper and lower bounds on $\Delta_k(e)$ is computationally intractable, under standard complexity assumptions Moitra & Rohatgi (2022). So, we analyze tightness of the bounds produced by ACRE and OHARE under some relatively mild distributional assumptions on $X$ and $Y$. Our main assumption for ACRE will be that the samples $X_1, \ldots, X_n$ are drawn iid from a well-behaved distribution:

**Definition 1** (Well-Behaved Distribution). *We say that a mean-zero distribution $\mathcal{X}$ on $\mathbb{R}^d$ with covariance $\Sigma = \mathbb{E}_{X \sim \mathcal{X}} [XX^{\mathsf{T}}]$ is well-behaved if it has exponentially decaying tails in the sense that*

$$\exists C > 0 \ \forall v \in \mathbb{S}^{d-1}, t > 0 \quad \Pr_{X \sim \mathcal{X}} \left[ \left| \left\langle v, \Sigma^{-1/2}X \right\rangle \right| > t \right] \leq \exp\left(-\Omega\left(t^C\right)\right).$$

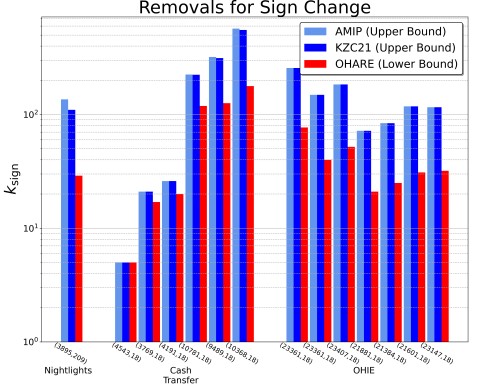

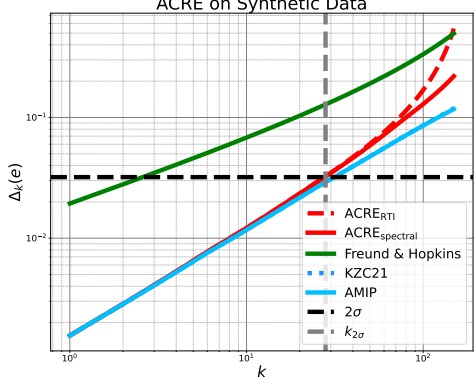

(a) A comparison of OHARE and known upper bounds on benchmark datasets.

(b) A comparison of ACRE and known bounds on a synthetic dataset (AMIP and KZC coincide).

Figure 2: A comparison of our ACRE and OHARE algorithms with previous techniques. In Figure 2a, we plot the number of removals required to flip the sign of several linear regressions from landmark econometrics studies Martinez (2022); Angelucci & De Giorgi (2009); Finkelstein et al. (2012). Each of these studies contains a number of linear regression central to their analyses, which include several applications of linear regression, such as estimating correlation controlled for additional covariates, treatment effects, and instrumental variables regression. For each regression, we run AMIP Broderick et al. (2020) and KZC Kuschnig et al. (2021) to obtain base-line upper bounds on $k_{\text{sign}}$ and compare the results to lower bounds produced by OHARE. We list the number of samples and the dimension of each regression below the plot. In Figure 2b, we consider a synthetic dataset comprised of $n = 4000$ samples in dimension $d = 50$, and plot bounds on the removal effects $\Delta_k(e)$. In this plot, the roles are reversed, with AMIP and KZC representing lower bounds on the removal effects, while our ACRE algorithm gives the first practical upper bound. We compare the bounds produced by ACRE to the previous state-of-the-art for efficiently computable upper bounds Freund & Hopkins (2023). Moreover, to ground the scale of the plot, consider the different bounds on $k_{2\sigma}$ (the number of removals required to shift the regression results outside of their 95% confidence intervals). The ACRE algorithm has two possible backends – spectral or RTI (see Section 4). RTI is more efficient and performs better in practice, while the spectral analysis which uses ideas from Freund and Hopkins' algorithm has a somewhat slower runtime ($\widetilde{O}(n^3)$) and offers a logarithmic advantage in some synthetic datasets. The bound produced by ACRE is almost tight on this range of values of $k$, while Freund and Hopkins' algorithm yields a trivial bound.

Note that the class of well-behaved distributions contains all subgaussian ($C = 2$) and all subexponential distributions ($C = 1$), but is not limited to these sets.

We will assume that the labels $Y_1, \ldots, Y_n$ are drawn according to a ground-truth linear model specified by an unknown vector $\beta_{\text{gt}} \in \mathbb{R}^d$. Concretely, we assume $Y_i = \langle \beta_{\text{gt}}, X_i \rangle + \epsilon_i$, where $\epsilon_1, \ldots, \epsilon_n$ are iid from $\mathcal{N}(0, \sigma^2)$ for an unknown variance parameter $\sigma^2 > 0$.

**Theorem 1.2** (Tightness of ACRE, Proof in Section E). *Let $n, d \in \mathbb{N}$, $e, \beta_{\text{gt}} \in \mathbb{R}^d$, $X, Y$, and $\sigma > 0$ be as above. There exists $k_{\text{threshold}} = \widetilde{\Theta}\left(\min\left\{\frac{n}{\sqrt{d}}, \frac{n^2}{d^2}\right\}\right)$ such that for all $k \leq k_{\text{threshold}}$, the bounds $L_k, U_k$ produced by ACRE satisfy*

$$\frac{U_k}{L_k} = 1 + \widetilde{O}\left(\frac{d + k\sqrt{d}}{n}\right). \tag{2}$$

Note that this theorem holds even when the covariance $\Sigma$ of $\mathcal{X}$ is unknown *a priori*.

RELATIONSHIP OF $k$, $d$, AND $n$ Theorem 1.2 guarantees that ACRE gives nearly optimal bounds so long as $n \gg d$ and the number of removals is at most $k \leq \widetilde{\Theta}(\min\{n/\sqrt{d}, n^2/d^2\})$; note that in this regime the RHS of (2) is $1 + o(1)$. First of all, OLS is only appropriate when $n \geq d$, and all of the datasets on which we perform experiments have $n$ well in excess of $d$. The range of interesting values

of $k$ is more subtle – to illustrate, consider $k_{2\sigma}(e)$. Assuming reasonably well-behaved samples, we expect $k_{2\sigma}(e) = O(\sqrt{n})$, so as long as we also have $n \gg d^{4/3}$, ACRE gives nearly tight bounds on $k_{2\sigma}(e)$. This represents a mild multiplicative overhead of $d^{1/3}$ samples compared to the $n \geq d$ required anyway to use OLS regression.

**Theory for OHARE**    Even though they are mild, the assumptions for ACRE are not satisfied by the real-world datasets used in our experiments, because of the presence of one-hot encoded categorical features, which is the reason we design OHARE in the first place. We also prove a tightness theorem for OHARE – we now turn to the assumptions on $X, Y$ which underlie it.

We study regression in $d + m$ dimensions – that is, $X \in \mathbb{R}^{n \times (d+m)}$. The block of $m$ coordinates will be a one-hot encoding of a categorical variable, while the block of $d$ coordinates will act as in the ACRE setting. Formally, let $n \in \mathbb{N}$ and let $B_1, \ldots, B_m$ partition $[n]$ into $m$ *buckets*. Let $e, \beta_{\mathrm{gt}} \in \mathbb{R}^{d+m}$, with $e$ supported only on the first $d$ coordinates, and $\sigma > 0$. For $i \in [n]$, let $X'_1, \ldots, X'_n$ be iid draws from a $d$-dimensional well-behaved distribution and let $X_i$ be $X'_i$ concatenated with the $j$-th indicator vector, where $j = j(i)$ is such that $i \in B_j$. Finally, let $Y_i = \langle \beta_{\mathrm{gt}}, X_i \rangle + \epsilon_i$ where $\epsilon_1, \ldots, \epsilon_n \sim \mathcal{N}(0, \sigma^2)$.

**Theorem 1.3** (Tightness of OHARE , informal, see Section F). *Let $X, Y, e$ be as described above. For any arbitrarily small $\varepsilon > 0$, if all buckets have sizes $n^\varepsilon \sqrt{d} < |B_j| < 0.49n$, and $n \geq d^{5/4+o(1)}$, then with high probability, for all $k < k_{\mathrm{threshold}}$, where $k_{\mathrm{threshold}} = \widetilde{\Theta}\left(\min\left\{\frac{n}{\sqrt{d}}, \frac{n^2}{d^2}, n^{1-\varepsilon}\right\}\right)$, the upper and lower bounds produced by OHARE satisfy*

$$\frac{U_k}{L_k} = 1 + O\left(\frac{1}{\sqrt{\log n}}\right).$$

Careful analysis is needed to prove Theorem 1.3 under the relatively weak assumptions $|B_j| > n^\varepsilon \sqrt{d}$ and $n \geq d^{5/4+o(1)}$. The stronger assumptions $|B_j| > d$ or $n \geq d^2$ would simplify the analysis. But neither assumption would be valid for all of our real-world datasets. Getting away with such weak assumptions ultimately requires us to put together a number of technical tools, including novel matrix concentration arguments (e.g. Lemma F.12).

Additionally, the error term $1/\sqrt{\log n}$, which we believe is nearly tight, goes to zero slowly compared to the error term in Theorem 1.2 – to capture this fine-grained behavior of $U_k/L_k$, our proof carefully exploits Gaussian anticoncentration.

## 2    PRELIMINARIES

Let $(X_1, Y_1), \ldots, (X_n, Y_n) \in \mathbb{R}^{(d+1)}$ represent features/covariates and labels/target variables of a linear regression instance. We always assume $n \geq d$. Let $\Sigma = X^\mathsf{T} X \in \mathbb{R}^{d \times d}$ denote the (un-normalized) empirical second moment of $X$, let $\beta$ denote the OLS fit $\beta = \Sigma^{-1} X^\mathsf{T} Y \in \mathbb{R}^d$, and let $R = Y - \beta X \in \mathbb{R}^n$ denote the residuals on the complete regression. Finally, let $e \in \mathbb{R}^d$ be some "direction of interest" along which we wish to certify the robustness of $\beta$. Using standard normalization techniques, we may ensure that $e \in \mathbb{S}^{d-1}$ has norm 1 and $\Sigma = I$.

For any $S \subseteq [n]$ representing some subset of the samples, let $X_S, Y_S$ represent the samples limited to only those whose indices lie in $S$. Similarly, let $\Sigma_S, \beta_S$ denote the empirical second moment and regression when using only the samples in $S = \overline{T}$. Note that while we could set $\Sigma = I$, removing some of the samples may change the covariance matrix $\Sigma_S = \Sigma - \Sigma_T \neq \Sigma$.

Finally, we use standard asymptotic notation $O(\cdot), \Theta(\cdot)$, and we write $f(n) = \widetilde{O}(g(n))$ if there is a constant $C$ such that $f(n) = O(g(n) \cdot \log^C n)$, and similarly for $\widetilde{\Theta}$.

## 3    ACRE: CERTIFYING ROBUSTNESS WITHOUT CATEGORICAL FEATURES

In this section we present ACRE (**A**lgorithm for **C**ertifying **R**obustness **E**fficiently), our algorithm for certifying robustness of regressions without categorical features. At the end of this section we give an overview of OHARE , but due to space constraints we defer the details of OHARE to Appendix A.

### 3.1 Separating First Order and Higher Order Effects on the OLS fit $\beta$

We split the effect of data removal on $\beta$ into a first order term and a higher order correction. The first order term is linear in the datapoints $X_i$, allowing us to analyze it exactly. For well-behaved datasets, the higher order term is smaller in magnitude, so even loose bounds on the higher order will suffice to generate tight bounds on the overall removal effect.

More concretely, let $T \subseteq [n]$ be a set of $k = |T| \ll n$ samples we might remove from the regression data, and set $S = [n] \setminus T$. A simple analysis yields the identity

$$\beta - \beta_S = \Sigma_S^{-1} \sum_{i \in T} R_i X_i \tag{3}$$

where $R$ denotes the residuals of the original regression and $\Sigma_S$ denotes the empirical 2nd moment of the retained samples $\Sigma_S = X_S^\mathsf{T} X_S = \sum_{i \in S} X_i X_i^\mathsf{T}$.

The difficulty in analyzing equation (3) is the effect induced by the non-linear matrix inversion operation. Recall that we normalized our datasets so that the empirical second moment over the entire dataset $\Sigma$ is the identity matrix. $\Sigma_S$ is generated by removing some of the samples, so it is no longer normalized.

Because only a very small number of samples were removed, we might hope $\Sigma_S = I - \Sigma_T$ is still close to the identity matrix. Therefore, it makes sense to try to develop equation (3) in orders of $\Sigma_T$.

More concretely, we use the identity $(I - \Sigma_T)^{-1} = I + (I - \Sigma_T)^{-1} \Sigma_T$ to derive:

$$\beta - \beta_S = \Sigma_S^{-1} \sum_{i \in T} R_i X_i = \underbrace{\sum_{i \in T} R_i X_i}_{\text{first order term}} + \underbrace{\Sigma_S^{-1} \Sigma_T \sum_{i \in T} R_i X_i}_{\text{higher order correction}} \tag{4}$$

Projecting the first order term onto some axis $e \in \mathbb{S}^{d-1}$ yields the gradients / influence scores used by the AMIP algorithm of Broderick et al. (2020). Our analysis will focus on bounding the higher order term.

### 3.2 Maximal Subset Sum Norm (MSN) – The Backend of ACRE

Under the hood of ACRE is a simple algorithm which places upper/lower bounds on the following optimization problem.

**Problem 2** (**M**aximal **S**ubset sum **N**orm (MSN)[1]). *Given a set of vectors $v_1, \dots, v_n \in \mathbb{R}^d$ with Gram matrix $G = (\langle v_i, v_j \rangle)_{i,j \in [n]}$, we define*

$$\forall k \in [n] \quad \mathrm{MSN}_k(G) = \max_{\substack{T \subseteq [n] \\ |T| = k}} \left\{ \left\| \sum_{i \in T} v_i \right\|_2 \right\} = \max_{\substack{T \subseteq [n] \\ |T| = k}} \left\{ \sqrt{1_T^\mathsf{T} G 1_T} \right\}$$

A constant-factor approximation to MSN would refute the small-set expansion hypothesis Hopkins & Li (2019), so we aim for *MSN-bounding* algorithms which place upper and lower bounds on the optimal value, with the aim that these bounds are close to tight on well-behaved $v_i$s. For our purposes, a simple algorithm in Section 4 gives useful bounds, but ACRE can use any MSN-bounding algorithm as a subroutine, and improved MSN-bounding algorithms will lead to improved performance for ACRE .

For now, we treat MSN-bounding as a black-box, and show how ACRE produces its upper and lower bounds $U_k, L_k$ by making a few calls to an MSN-bounding algorithm.

### 3.3 Reducing Robustness Certification to MSN-bounding

Recall equation (4) for the effect of removals on an OLS regression. Projecting onto $e$, we have

---

[1]When the vectors have 0 mean ($\sum_{i \in [n]} v_i = 0$), MSN is equivalent to resilience Steinhardt et al. (2017).

$$\langle e, \beta - \beta_S \rangle = \left\langle e, \Sigma_S^{-1} \sum_{i \in T} R_i X_i \right\rangle = \underbrace{\sum_{i \in T} R_i \langle X_i, e \rangle}_{\text{first order term}} + \underbrace{\left\langle \Sigma_S^{-1} \Sigma_T e, \sum_{i \in T} R_i X_i \right\rangle}_{\text{high order term}} \tag{5}$$

We compute the first order term of equation (5) exactly for all $k \in [n]$ via a greedy algorithm in time $O(n \log(n))$. For the rest of our analysis, we focus on bounding the maximum value of the high order term. We use the following bound:

$$|\text{high order term}| \le \max(\lambda(\Sigma_S^{-1})) \, \|(\Sigma_S - I)e\| \left\| \sum_{i \in T} X_i R_i \right\| \tag{6}$$

where $\max(\lambda(\Sigma_S^{-1}))$ is the largest eigenvalue of $\Sigma_S^{-1}$. We show that each of the three terms in the RHS of equation (6) can be upper-bounded by the value of an MSN problem. The last term, $\|\sum_{i \in T} X_i R_i\|$, is already corresponds to an MSN problem with the Gram matrix $G_{RX} = \text{diag}(R) G_X \text{diag}(R)$, so we focus on the other two terms.

**MSN bound for** $\max(\lambda(\Sigma_S^{-1}))$    We start by simplifying the $\Sigma_S^{-1}$ term. Recall that $\Sigma_S = I - \Sigma_T$, allowing us to use

$$\max \left\{ \lambda \left( \Sigma_S^{-1} \right) \right\} \le \frac{1}{1 - \max \{ \lambda(\Sigma_T) \}} \tag{7}$$

so long as $\max \{ \lambda(\Sigma_T) \} < 1$. Over the next few lines we bound $\|\Sigma_T\|$, allowing us to verify this assumption. We apply the inequality

$$\max \left\{ \lambda \left( \Sigma_T \right) \right\} \le \sqrt{\sum_{\lambda_i \in \lambda(\Sigma_T)} \lambda_i^2} = \|\Sigma_T\|_F = \left\| \sum_{i \in T} X_i \otimes X_i \right\|$$

where $\|M\|_F$ is the Frobenius norm of a matrix $M$ ($\ell_2$ norm of the vector of eigenvalues) and $\otimes$ denotes tensor product. So we have an MSN problem with the vectors $X_i \otimes X_i$.

Even though this MSN is represented by $n$ vectors of dimension $d^2$, its Gram matrix representation $G_{X \otimes X}$ can still be computed in time $O(n^2 d)$, by computing the Gram matrix $G_X = (\langle X_i, X_j \rangle)_{i,j \in [n]}$, and squaring its entries to obtain

$$G_{X \otimes X} = (\langle X_i \otimes X_i, X_j \otimes X_j \rangle)_{i,j \in [n]} = \left( \langle X_i, X_j \rangle^2 \right)_{i,j \in [n]}$$

**MSN bound for** $\Sigma_T e$    Recall $\Sigma_T = \sum_{i \in T} X_i X_i^{\mathsf{T}}$. Let $G_{X \langle X, e \rangle}$ be the Gram matrix of $\{X_i \langle X_i, e \rangle\}_{i \in [n]}$. So,

$$\|\Sigma_T e\| = \left\| \sum_{i \in T} X_i \langle X_i, e \rangle \right\| \le \text{MSN} \left( G_{X \langle X, e \rangle} \right) ,$$

### 3.4   Algorithm

Combining the results of the analysis above yields the following expression and the corresponding Algorithm 1:

$$|\Delta_k(e) - (\text{first-order term})_k| \le \frac{1}{1 - \text{MSN}_k \left( G_{X \otimes X} \right)} \cdot \text{MSN}_k \left( G_{XR} \right) \cdot \text{MSN}_k \left( G_{XZ} \right)$$

---

**Algorithm 1:** ACRE (**A**lgorithm for **C**ertifying **R**obustness **E**fficiently)

---

**Input:** Linear regression problem $X \in \mathbb{R}^{n \times d}$, $Y \in \mathbb{R}^d$, direction of interest $e \in \mathbb{R}^d$,
       MSN-bounding algorithm $\mathcal{M}$
**Output:** Upper and lower bounds $U, L \in \mathbb{R}^n$ such that $U_k \geq \Delta_k(e) \geq L_k$

1   Compute the Gram matrix $G_X = XX^{\mathsf{T}} \in \mathbb{R}^{n \times n}$;
2   Compute the projection of the samples on the direction of interest $Z = Xe$;
3   Define $A = $ `cumulative sum(sorted(`$RZ$`))`;

4   $G_{X \otimes X} = $ pointwise square of the entries of $G_X$;
5   $G_{XR} = \operatorname{diag}(R) \, G_X \operatorname{diag}(R)$;
6   $G_{X\langle X, e \rangle} = \operatorname{diag}(Z) \, G_X \operatorname{diag}(Z)$;

7   Run the MSN-bound algorithm $\mathcal{M}$ on these $G_{X \otimes X}, G_{XR}, G_{X\langle X, e \rangle}$ to compute upper bounds
    $M_{X \otimes X}, M_{XR}, M_{X\langle X, e \rangle} \in \mathbb{R}^n$;

8   **return** $U, L = A \pm \frac{1}{1 - M_{X \otimes X}} M_{XR} M_{X\langle X, e \rangle}$ ;         `// pointwise operations`

---

### 3.5 OHARE: Awareness of Categorical Features

We present an overview of OHARE (**O**ne-**H**ot aware **A**lgorithm for certifying **R**obustness **E**fficiently),
deferring details to the appendix. Suppose that $X_1, \ldots, X_n \in \mathbb{R}^{d+m}$ consist of $d$ continuous-valued
features and a single categorical feature with $m$ categories, one-hot encoded.

The bounds computed by ACRE are valid for such $X_1, \ldots, X_n$. But, consider removing the samples
$T$ comprising any single category, encoded in coordinate $i$; let $S = [n] \setminus T$. The matrix $\Sigma_S$ is
singular, since all variance from category $i$ has been removed. So our approach to bounding the
high-order term from (6) by pulling out $\max(\lambda(\Sigma_S^{-1}))$ is doomed to failure once $k$ exceeds the size
of the smallest category, since $\max(\lambda(\Sigma_S^{-1})) \to \infty$. We will assume that the direction of interest $e$
lies orthogonal to the one-hot features, so $e \in \mathbb{R}^d$.

The key idea in OHARE is to rephrase the OLS algorithm as a two-phase process, first explicitly
controlling for the categorical feature by computing a "controlled" dataset $\{(\widetilde{X}_i, \widetilde{Y}_i)\}_{i \in [n]} \subseteq \mathbb{R}^{d+1}$,
then performing OLS on the controlled dataset to arrive at $\beta \in \mathbb{R}^d$. We show that this process yields
the same $\beta$ which would have been produced by running OLS on the original dataset $\{(X_i, Y_i)\}_{i \in [n]}$
(Claim A.1) and restricting to the span orthogonal to the one-hot encoding.

We derive explicit formulae for $\widetilde{X}_i, \widetilde{Y}_i$ by analyzing the Gram-Schmidt orthogonalization process
which we call "reaveraging". The upshot is that we replace each term in (6) as well as the first
order effect, with two terms: one corresponding to the direct effect of sample removals on the
$\widetilde{X}, \widetilde{Y}$ regression, and one corresponding to effects of removing $X_T, Y_T$ on the remaining controlled
samples $\widetilde{X}_S, \widetilde{Y}_S$ through reaveraging. Crucially, this process allows us to certify that the matrix $\widetilde{\Sigma}_S$
is nonsingular even in cases where $\Sigma_S$ can be singular.

To bound the new correction term coming from the influence of categorical features on the sample-
removal effect, we use a knapsack-style dynamic program to combine bounds on the influence of data
removals from each category into a single bound by searching over partitions $k = k_1 + \ldots + k_m$.

## 4 MSN-Bounding Algorithms

In this section we discuss the simple MSN-bounding algorithm we use as the backend of ACRE and
OHARE for all our experiemnts on real-world data. We call this algorithm **R**efined **T**riangle **I**nequality
(RTI). We implemented other MSN-bounding algorithms, in particular one based on eigenvalues and
eigenvectors of the Gram matrix $G$, but found that they improved over RTI only on synthetic data, so
we defer them to the appendix.

RTI relies on the following inequality, evaluating the RHS via a greedy algorithm:

$$\max_{|T|=k} \left\| \sum_{i \in T} X_i \right\|^2 = \max_{|T|=k} \sum_{i, j \in T} \langle X_i, X_j \rangle \leq \max_{|T|, |S_1|, \ldots, |S_k|=k} \sum_{i \in T, j \in S_i} \langle X_i, X_j \rangle. \tag{8}$$

---

**Algorithm 2:** Refined Triangle Inequality

---

**Input:** Gram matrix $G$ of size $n \times n$ where $G_{ij} = \langle Z_i, Z_j \rangle$
**Output:** A vector $V$ of length $n$, where $V_k$ is an upper bound on the $\ell_2$ norm of the sum of any $k$ vectors in $Z$, for $k = 1$ to $n$

1 Sort each row of $G$ in decreasing order;
2 Compute the cumulative sum of the rows of $G$ and store the result in $C$;
3 Sort the columns of $C$ in decreasing order;
4 Compute the cumulative sums of the columns of $C$ and store the results in $S$;
5 **for** $k \leftarrow 1$ **to** $n$ **do**
6 $\quad$ Set $V_k$ to be $\sqrt{S_{k,k}}$;

7 **return** $V$;

---

**Greedy Algorithm for Diagnosing Robustness Failures via MSN**  We also implement a simple greedy algorithm to compute *lower* bounds on MSN, discussed in Section B. This algorithm is not used in ACRE or OHARE , but can be used to diagnose robustness failures by finding subsets of influential samples, thereby complementing AMIP.

## 5  PRIOR WORK AND FUTURE DIRECTIONS

**Prior Work**  Robustness in linear regression is too vast to survey here, so we restrict attention to recent works in robustness auditing. As previously discussed, our work is preceded by Broderick et al. (2020); Kuschnig et al. (2021); Moitra & Rohatgi (2022); Freund & Hopkins (2023), all studying robustness auditing for least-squares regression. Unlike any of these prior works, our algorithms provide nontrivial lower bounds on $k_\theta$ in practice for datasets with tens/hundreds of dimensions. All these algorithms and ours fit into the broader tradition of influence functions as a measure of robustness in regression, dating at least to Cook & Weisberg (1980).

Our algorithms are partly inspired by recent developments in algorithmic robust statistics – see Diakonikolas & Kane (2023) and references therein. While the main goal in robust statistics differs somewhat from robustness auditing, modern algorithms for robust regression typically contain subroutines for tasks very similar to robustness auditing. But the subroutines in recent breakthroughs in robust regression Klivans et al. (2018); Bakshi & Prasad (2021) are not practically implementable due to reliance on semidefinite programming.

**Future Directions**  We propose several directions for future work:

BEYOND OLS  Given the prevalence of regression beyond ordinary least squares in machine learning, an important next step is to design algorithms to certify robustness of other regression methods which arise by minimizing a convex loss – e.g. logistic regression and LASSO Tibshirani (1996).

APPLICATIONS TO DIFFERENTIAL PRIVACY (DP)  DP Dwork et al. (2006) is the gold-standard mathematically rigorous approach to protecting privacy of individuals represented in a dataset. Tighter certificates of robustness for regression have great potential to improve *privacy-accuracy tradeoffs* in private regression Dwork & Lei (2009); poor privacy-accuracy tradeoffs are a major roadblock to widespread adoption of private data analysis techniques.

INTERPRETATIONS OF ROBUSTNESS CERTIFICATES  It is an appealing intuition that statistical conclusions which are robust to removing many samples should generalize better – formalizations of the relationship between stability and generalization have been highly influential, e.g. Bousquet & Elisseeff (2002). Can this intuition be formalized? Can robustness certificates yield e.g. tighter empirical confidence intervals, or out-of-distribution generalization guarantees?

BEYOND SAMPLE REMOVAL  Removing a small set of datapoints is just one of many potential ways to perturb a dataset. Can we certify robustness of OLS or other regression algorithms to other types of dataset perturbation, e.g. $\ell_2$ or $\ell_\infty$-bounded perturbations of the feature vectors?

ACKNOWLEDGMENTS

This work was supported by NSF CAREER award no. 2238080 and CSAIL Alliances.

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

## A    CATEGORICAL-AWARE ROBUSTNESS ANALYSIS (OHARE)

ACRE encounters an issue when the regression includes a one-hot encoding of a categorical feature. This is because removing all the samples from one category of the one-hot encoding could cause $\Sigma_S$ to have a singularity, making our bound, which depends linearly on $\max\{\lambda(\Sigma_S^{-1})\}$, effectively meaningless.

Moreover, as we show in Appendix G, it is possible to perturb datasets with one-hot encodings, even those which are robust to many sample removals, to make them extremely brittle to sample removals. Therefore, any "continuous" algorithm which does not somehow utilize the discreteness of the one-hot encoding is doomed to fail.

### A.1    REGRESSING OVER A CATEGORICAL FEATURE

Consider a linear regression over a set of feature vectors, $F_i \in \mathbb{R}^d$ for $i \in n$, and an additional $m$ dimensions corresponding to a one-hot encoding with buckets $B_1 \sqcup \cdots \sqcup B_m = [n]$, where we want to fit our labels $L_i$ to a model of the form

$$L_i \approx \sum_{j \in [d]} \beta_j F_{i,j} + \sum_{j \in [m]} t_j 1_{i \in B_j}$$

This linear regression would be represented by a matrix $X \in \mathbb{R}^{n \times (m+d)}$ whose rows are the samples $X_i = (F_i, 1_{i \in B_1}, \ldots, 1_{i \in B_m})$. Then, if we are interested in the correlation between one of the continuous features $F_{:,j}$ and the labels $Y$ (controlled for the rest of our categorical / continuous features), we may compute this correlation by running an OLS regression $\beta = (X^\intercal X)^{-1} X^\intercal Y$ and output the relevant entry $\beta_j$ of the fit.

Analyzing the robustness of this process is challenging, and we proceed by performing the Gram Schmidt orthogonalization between the dummy variables and the continuous features explicitly. Before delving into the details of this analysis, we note that OHARE can be used to certify robustness for a slightly more general class of regressions. Using this more general notation will help motivate our orthogonalization process and is crucial for certifying the robustness of weighted regressions with categorical features (such as the ones in the OHIE study - see Appendix C.4).

Indeed, let $u_1, \ldots, u_m$ be the columns representing the dummy variables. So long as for any sample $i \in [n]$ there is a unique $j = b(i)$ (representing the bucket to which the $i$th sample belongs) such that $u_{j,i} \neq 0$ (and for all $j \neq j'$, we have $u_{j',i} = 0$). This property ensures that for any $S \subseteq [n]$, these columns are still perpendicular to one another $u_{j,S} \perp u_{j',S}$.

1. For each bucket $B_j \subseteq [n]$, we compute the weighted averages over the features $f_{j,i} = \frac{uu^\intercal}{\|u\|^2} F = \frac{\sum_{i \in B_j} u_{j,i} f_{i,:}}{\|u_j\|^2} u_{j,i}$ and over the labels $\ell_j = \frac{\sum_{i \in B_j} u_{j,i} \ell_i}{\|u_j\|^2} u_{j,i}$ for samples from this bucket. When the samples are unweighted $u_{j,i} = 1_{i \in B_j}$, these are equal to the averages over the features / labels $f_j = \mathbb{E}_{i \in B_j}[F_{i,:}]$ and $\ell_j = \mathbb{E}_{i \in B_j}[L_i]$, and when the samples are weighted, these are the projections of the continuous features / labels onto the space spanned by the indicator columns.

2. Compute the normalized features $X \in \mathbb{R}^{d \times n}$ obtained by subtracting the feature averages of each bucket from its samples, and the normalized labels $Y \in \mathbb{R}^n$ obtained by subtracting the average label from each bucket:

$$X_i = F_{i,:} - f_{b(i),i}$$
$$Y_i = L_i - \ell_{b(i),i}$$

3. Perform a linear regression on $Y$ as a function of $X$ and output the fit $\beta = (X^\intercal X)^{-1} X^\intercal Y$.

**Claim A.1.** *The output of the process described above $\beta$ is equal to the coefficients for the continuous features on a full OLS with the dummy variables.*

The proof of Claim A.1 follows directly by performing the Gram-Schmidt orthogonalization (to compute $\Sigma^{-1}$) explicitly while taking the one-hot encoding columns into account (see Section A.6). This approach of explicitly writing out a Gram-Schmidt orthogonalization to remove the effects of

some of the controls could potentially be used whenever we have two sets of features, one of which is easier to analyze than the other.

Our focus on one-hot encodings is due to their prevalence in econometrics datasets, along with the fact that, as we will see over the next few pages, the perpendicular structure of the indicator variables makes them particularly amenable to such a divide and conquer strategy (and much harder to deal with using previous techniques).

For ease of notation, we will normalize the dummy variables so that $\|u_j\| = 1$ for all $j \in [m]$.

## A.2 FIRST ORDER EFFECTS

Claim A.1 allows us to reduce the problem of performing a regression on $d$ continuous feature and a categorical feature with $m$ potential values into a regression on just a reaveraged version of the $d$ continuous features. Using this reduction, we can split the effects of removing samples from the smaller linear regression into direct removal effects that change the regression fit directly by removing samples from the $d$-dimensional regression $X, Y$, and reaveraging effects that shift the regression fit by causing the expectation of the data within each bucket to change (effectively shifting the values of the retained rows of $X, Y$).

In particular, we may again write

$$\langle e, \beta - \beta_S \rangle = - \left\langle e, \hat{\Sigma}_S^{-1} \sum_{i \in S} \hat{X}_i \hat{R}_i \right\rangle \tag{9}$$

where $\hat{\cdot}$ is the value of $\cdot$ after the new reaveraging due to the removal of the elements in $T$. More concretely, we have

$$\hat{X}_i = X_i - x_{b(i)} u_{b(i),i} \text{ where } x_j = \sum_{i' \in B_j \cap S} X_i u_{j,i'}$$

$$\hat{R}_i = R_i - r_{b(i)} u_{b(i),i} \text{ where } r_j = \sum_{i' \in B_j \cap S} R_i u_{j,i'} \tag{10}$$

From here, we can derive closed form formula for the new versions of all the terms in the previous analysis. For instance, the "first order effect" (i.e., the effect we would have seen on the regression, had $\Sigma_S$ been equal to $I$), which was previously given by First Order$_{\text{continuous}} = \sum_{i \in T} R_i \langle e, X_i \rangle$ is now given by

$$\text{First Order}_{\text{one-hot}} = - \left\langle e, \sum_{i \in S} \hat{X}_i \hat{R}_i \right\rangle = - \left\langle e, \sum_{i \in S} (X_i - x_{b(i)} u_{b(i),i}) \hat{R}_i \right\rangle =$$

$$= - \left\langle e, \sum_{j \in [m]} \sum_{i \in B_j \cap S} (X_i - x_j u_{j,i}) \hat{R}_i \right\rangle =$$

$$= - \left\langle e, \sum_{j \in [m]} \sum_{i \in B_j \cap S} X_i \hat{R}_i \right\rangle - \left\langle e, \sum_{j \in [m]} x_j \underbrace{\sum_{i \in B_j \cap S} u_{j,i} \hat{R}_i}_{=0} \right\rangle =$$

$$= - \left\langle e, \sum_{j \in [m]} \sum_{i \in B_j \cap S} X_i R_i \right\rangle + \left\langle e, \sum_{j \in [m]} r_j \underbrace{\sum_{i \in B_j \cap S} u_{j,i} X_i}_{= -\sum_{i \in B_j \cap T} X_i u_{j,i}} \right\rangle =$$

$$= \underbrace{\left\langle e, \sum_{i \in T} X_i R_i \right\rangle}_{\text{First Order}_{\text{continuous}}} + \underbrace{\sum_{j \in [m]} \frac{1}{\|u_{j,S}\|^2} \left( \sum_{i \in T \cap B_j} R_i u_{j,i} \right) \left( \sum_{i \in T \cap B_j} \langle e, X_i \rangle u_{j,i} \right)}_{\text{Correction Term}}$$

$$\tag{11}$$

In order to bound the right-hand-side of equation (11), we consider each of the 3 terms $\sum_{i\in T\cap B_j} R_i \langle e, X_i\rangle$, $\sum_{i\in T\cap B_j} R_i u_{j,i}$ and $\sum_{i\in T\cap B_j} \langle e, X_i\rangle u_{j,i}$ separately. Using a greedy algorithm, we can maximize / minimize each of these terms separately as a function of $k_j = |T\cap B_j|$. We can then combine these into a bound of the form

$$\text{bound}_j^{\pm}(k_j) = d_j(k_j) + \frac{c_j^{\pm}(k_j)}{\min_{S_j\in\binom{B_j}{k_j}}\left\{\left\|u_{j,S_j}\right\|^2\right\}}$$

where

$$d_j(k_j) = \max_{\substack{T_j\subseteq B_j\\|T_j|=k_j}} \sum_{i\in T_j} R_i \langle e, X_i\rangle$$

$$c_j^+(k_j) = \max\left\{\left(\max_{\substack{T_j\subseteq B_j\\|T_j|=k_j}}\sum_{i\in T_j} R_i u_{j,i}\right)\left(\max_{\substack{T_j\subseteq B_j\\|T_j|=k_j}}\sum_{i\in T_j} \langle e, X_i\rangle u_{j,i}\right), \left(\min_{\substack{T_j\subseteq B_j\\|T_j|=k_j}}\sum_{i\in T_j} R_i u_{j,i}\right)\left(\min_{\substack{T_j\subseteq B_j\\|T_j|=k_j}}\sum_{i\in T_j} \langle e, X_i\rangle u_{j,i}\right)\right\}$$

$$c_j^-(k_j) = \min\left\{\left(\min_{\substack{T_j\subseteq B_j\\|T_j|=k_j}}\sum_{i\in T_j} R_i u_{j,i}\right)\left(\max_{\substack{T_j\subseteq B_j\\|T_j|=k_j}}\sum_{i\in T_j} \langle e, X_i\rangle u_{j,i}\right), \left(\max_{\substack{T_j\subseteq B_j\\|T_j|=k_j}}\sum_{i\in T_j} R_i u_{j,i}\right)\left(\min_{\substack{T_j\subseteq B_j\\|T_j|=k_j}}\sum_{i\in T_j} \langle e, X_i\rangle u_{j,i}\right)\right\}$$

For $k_j = n_j$, there is no reaveraging effect and $\text{bound}_j^{\pm}(n_j) = d_j(n_j)$.

We can then combine these bounds on the individual buckets by using a dynamic programming algorithm to solve the integer knapsack problem of

$$\max_{k_1+\cdots+k_m=k} \sum_{j\in[m]} \text{bound}_j^{\pm}(k_j).$$

From equation (11), it is clear that these maximizations yield upper and lower bounds on the maximal first order removal effect

$$\max k_1+\cdots+k_m = k \sum_{j\in[m]} \text{bound}_j^-(k_j) \leq \max_{S\in\binom{[n]}{n-k}}\left\{-\left\langle e, \sum_{i\in S}\hat{X}_i\hat{R}_i\right\rangle\right\} \leq \max_{k_1+\cdots+k_m=k} \sum_{j\in[m]} \text{bound}_j^+(k_j).$$

The upper bound on the overall first order effect tends to be larger than the continuous first order effect but only by a $1 + o(1)$ factor. This is because $d_j$ tends to be the dominant effect, while $c_j$ are typically much smaller.

For instance, consider an unweighted regression on normally distributed samples $(X_i, Y_i)$, and let $\sigma_R = \sqrt{\frac{1}{n}\sum_{i\in[n]} R_i^2}$ denote the root-mean-square (RMS) / scale of the residuals. With high probability we can find removal sets that would have $d_j \approx \frac{k_j}{\sqrt{n}}\sigma_R \log(n/k)$ (by taking only samples which are on the $\varepsilon$ tail end of the distribution of having both large residual and large inner product with the direction of interest). On the other hand, for sufficiently large buckets $|B_j| \gg \log(n)$, we expect to have $c_j = O\left(\frac{k_j^2}{|B_j|\sqrt{n}}\sigma_R\right) < O\left(\frac{k_j}{\sqrt{n}}\sigma_R\right)$, so they tend to be somewhat smaller than $d_j$.

Moreover, by using a dynamic programming algorithm, we can enforce the constraint that the number of samples removed from each bucket $k_j$ has to be the same for the direct effects and for the reaveraging effect. This constraint limits causes the contribution of the reaveraging effects to be even smaller, as they require taking many samples from the same bucket to enjoy the quadratic scaling of $k_j$, while the more dominant direct effects are typically optimized by selecting samples evenly from the buckets.

---

**Algorithm 3:** Dynamic Programming Algorithm for Integer Knapsack

---

**Input:** List of bounds $\{\text{bound}_j(k_j)\}_{j=1}^m$, maximal budget $k_{\max}$
**Output:** Array $F$ where $F[k]$ is the highest total score possible for budget $k$

1  $m \leftarrow |\{\text{bound}_j(k_j)\}|$;
2  $k_{\max} \leftarrow k_{\max} + 1$;
    // Adjusting for 0-based indexing in NumPy arrays
3  Initialize $F$ array of size $(m+1) \times k_{\max}$ with $-\infty$;
4  Set the first column of $F$ to 0;
5  **for** $j \leftarrow 1$ **to** $m$ **do**
6      $\text{bound} \leftarrow \text{bound}_j$;
7      $F[j, 0:\texttt{len}(\text{bound})] \leftarrow \text{bound}[0:k_{\max}]$;
8      $F[j, :] \leftarrow \max(F[j, :], F[j-1, :])$;
9      **for** $\delta_k \leftarrow 1$ **to** $\min(k_{\max}, \texttt{len}(\text{bound})) - 1$ **do**
10         $F[j, \delta_k :] \leftarrow \max(F[j, \delta_k :], F[j-1, :-\delta_k] + \text{bound}[\delta_k])$;

11  **return** $F[m, : k_{\max} - 1]$

---

### A.3 HIGH ORDER TERMS

We continue our analysis of equation (9). As in the continuous analysis, we have

$$
\begin{aligned}
\langle e, \beta - \beta_S \rangle &= -\left\langle e, \hat{\Sigma}_S^{-1} \sum_{i \in S} \hat{X}_i \hat{R}_i \right\rangle = \\
&= -\left\langle e, I \sum_{i \in S} \hat{X}_i \hat{R}_i \right\rangle - \left\langle e, \left(\hat{\Sigma}_S^{-1} - I\right) \sum_{i \in S} \hat{X}_i \hat{R}_i \right\rangle = \\
&= \text{First Order}_{\text{one-hot}} + \left\langle \hat{\Sigma}_S^{-1}\left(I - \hat{\Sigma}_S\right) e, \sum_{i \in S} \hat{X}_i \hat{R}_i \right\rangle \leq \\
&\leq \text{First Order}_{\text{one-hot}} + \max\left\{\lambda\left(\hat{\Sigma}_S^{-1}\right)\right\} \times \left\|\left(I - \hat{\Sigma}_S\right)e\right\| \times \left\|\sum_{i \in S} \hat{X}_i \hat{R}_i\right\|
\end{aligned}
\tag{12}
$$

To analyze the first 2 higher order terms, we begin with an analysis of $\hat{\Sigma}_S$. This analysis shows that $\hat{\Sigma}_S$ essentially behaves like $\Sigma_S$ with a minor correction for each bucket:

$$
\begin{aligned}
\hat{\Sigma}_S &= \sum_{i \in S} \hat{X}_i \hat{X}_i^{\mathsf{T}} = \sum_{j \in [m]} \sum_{i \in B_j \cap S} (X_i - x_j u_{j,i}) \hat{X}_i^{\mathsf{T}} = \\
&= \sum_{j \in [m]} \sum_{i \in B_j \cap S} X_i (X_i - x_j u_{j,i})^{\mathsf{T}} - \sum_{j \in [m]} x_j \underbrace{\sum_{i \in B_j \cap S} \hat{X}_i^{\mathsf{T}} u_{j,i}}_{=0} = \\
&= \sum_{j \in [m]} \sum_{i \in B_j \cap S} X_i X_i^{\mathsf{T}} - \sum_{j \in [m]} \left(\sum_{i \in B_j \cap T} X_i u_{j,i}\right)\left(\sum_{i \in B_j \cap T} X_i\right)^{\mathsf{T}} = \\
&= I - \Sigma_T - \sum_{j \in [m]} \frac{1}{\|u_{j,S}\|^2} \left(\sum_{i \in B_j \cap T} X_i u_{j,i}\right)\left(\sum_{i \in B_j \cap T} X_i u_{j,i}\right)^{\mathsf{T}}
\end{aligned}
\tag{13}
$$

In order to bound $\max\left\{\lambda\left(\hat{\Sigma}_S^{-1}\right)\right\} = \min\left\{\lambda\left(\hat{\Sigma}_S\right)\right\}^{-1}$ from above, we begin by bounding $\max\left\{\lambda\left(\Sigma_T\right)\right\}$ from above (using the same MSN-bounding reductions from Section 3). We then use a MSN-bounding algorithm and the same dynamic programming as above to bound the term in

equation (14) (MSN-bounding to bound the individual terms and dynamic programming to combine them).

$$
\max\left\{\lambda\left(\sum_{j\in[m]}\frac{1}{\|u_{j,S}\|^2}\left(\sum_{i\in B_j\cap T}X_iu_{j,i}\right)\left(\sum_{i\in B_j\cap T}X_iu_{j,i}\right)^{\mathsf{T}}\right)\right\}=
$$

$$
=\max_{k_1+\cdots+k_m=k}\sum_{j\in[m]}\frac{1}{\min_{S_j}\left\{\|u_{j,S_j}\|^2\right\}}\max_{\substack{T_j\subseteq B_j\\|T_j|=k_j}}\left\{\left\|\sum_{i\in T_j}X_iu_{j,i}\right\|^2\right\}
\tag{14}
$$

For each $j$, we can bound $\max_{\substack{T_j\subseteq B_j\\|T_j|=k_j}}\left\{\left\|\sum_{i\in T_j}X_iu_{j,i}\right\|\right\}$ from above by performing a call to an MSN bounding algorithm. However, we can refine the results of this MSN bounding call (especially with regards to larger values of $k_j\geq\frac{n_j}{2}$), by utilizing the fact that $\sum_{i\in B_j}X_i=0$, which implies that

$$
\max_{\substack{T_j\subseteq B_j\\|T_j|=k_j}}\left\{\left\|\sum_{i\in T_j}X_iu_{j,i}\right\|\right\}=\max_{\substack{T_j\subseteq B_j\\|T_j|=n_j-k_j}}\left\{\left\|\sum_{i\in T_j}X_iu_{j,i}\right\|\right\}.
$$

Therefore, if $M_j$ is the result of an MSN bounding algorithm (such as RTI) on the $n_j\times n_j$ Gram matrix of the samples in the $j$th bucket, we have the bound

$$
\max_{\substack{T_j\subseteq B_j\\|T_j|=k_j}}\left\{\left\|\sum_{i\in T_j}X_iu_{j,i}\right\|\right\}\leq\min\left\{M_j(k_j),M_j(n_j-k_j)\right\}.
$$

Similarly, we may bound the other terms. Using equation (13), we have

$$
\left(I-\hat{\Sigma}_S\right)e=\left(\Sigma_T+\sum_{j\in[m]}\left(\sum_{i\in B_j\cap T}X_iu_{j,i}\right)\left(\sum_{i\in B_j\cap T}X_iu_{j,i}\right)^{\mathsf{T}}\right)e=
$$

$$
=\sum_{i\in T}X_i\langle X_i,e\rangle+\sum_{j\in[m]}\frac{1}{\|u_{j,S}\|^2}\left(\sum_{i\in T\cap B_j}X_iu_{j,i}\right)\left(\sum_{i\in T\cap B_j}\langle X_i,e\rangle u_{j,i}\right)
\tag{15}
$$

As usual, we bound the norm of the first term $\sum_{i\in T}X_i\langle X_i,e\rangle$ using an MSN-bounding algorithm, and for each bucket, we bound $\left\|\sum_{i\in T\cap B_j}X_iu_{j,i}\right\|$ and $\left|\sum_{i\in T\cap B_j}\langle X_i,e\rangle u_{j,i}\right|$ as a function of $k_j=|T\cap B_j|$. We then combine these bounds using the triangle inequality, and the same symmetry and dynamic programming algorithm as above.

Finally, to bound $\left\|\sum_{i\in S}\hat{X}_i\hat{R}_i\right\|$, we use the same analytic techniques to write

$$
\sum_{i\in S}\hat{X}_i\hat{R}_i=\sum_{j\in[m]}\sum_{i\in B_j\cap S}(X_i-x_ju_{j,i})\hat{R}_i=\cdots
$$

$$
\cdots=\sum_{i\in T}X_iR_i+\sum_{j\in[m]}\frac{1}{\|u_{j,S}\|^2}\left(\sum_{i\in B_j\cap T}X_iu_{j,i}\right)\left(\sum_{i\in B_j\cap T}R_iu_{j,i}\right)
\tag{16}
$$

which can be bounded by the same MSN + symmetry + dynamic programming above.

## A.4 THE OHARE ALGORITHM

---

**Algorithm 4:** The OHARE Algorithm

---

**Input:** Samples $X_i \in \mathbb{R}^d$ (s.t. $X^\intercal X = I \in \mathbb{R}^{d \times d}$) and residuals $R_i \in \mathbb{R}$ for $i \in [n]$, vector $e \in \mathbb{R}^d$, separation of the samples into buckets $b : [n] \to [m]$ (based on some additional categorical feature), weights $w \in \mathbb{R}^n$ (by default all 1s).

**Output:** $U, L \in \mathbb{R}^n$ that bound the removal effects $\Delta_k(e)$.

```
/* Step 1:  Split the samples and residuals by their b value
   into buckets B_1,...,B_m ⊆ [n]                              */
```
1 **for** $j \leftarrow 1$ **to** $m$ **do**
2     **for** $k_j \leftarrow 1$ **to** $|B_j|$ **do**
3         Compute $\text{bound}_j(k_j) = d_j(k_j) + \frac{c_j(k_j)}{|B_j| - k_j}$ with $c_j$ and $d_j$ as defined above;

```
/* Step 2:  Compute Influences using the 1D Dynamic Programming
   Algorithm                                                    */
```
4 Use the "1D Dynamic Programming Algorithm" defined above to compute the bounds on the direct influences and store the result as Influences;

```
/* Step 3:  Compute upper bounds on M_{k_j}, U_{k_j}, ρ_{k_j}, and ζ_{k_j} for
   each bucket                                                  */
```
5 **for** $j \leftarrow 1$ **to** $m$ **do**
6     Use an MSN bounding algorithm to compute an upper bound on
    $M_{k_j} \geq \max_{T \subseteq B_j, |T| = k_j} \left\| \sum_{i \in T} X_i \right\|$ for all $k_j \in [|B_j|]$;
7     Use a sort + cumulative sum to compute $U_{k_j} = 1 - \max_{T \subseteq B_j, |T| = k_j} \left\{ \sum_{i \in T} u_{i,j}^2 \right\}$;
8     Use a similar sort + cumulative sum to compute $\rho_{k_j} = \max_{T \subseteq B_j, |T| = k_j} \left\{ \sum_{i \in T} |R_i u_{i,j}| \right\}$;
9     Use a similar sort + cumulative sum to compute $\zeta_{k_j} = \max_{T \subseteq B_j, |T| = k_j} \left\{ \sum_{i \in T} |Z_i u_{i,j}| \right\}$;
10     Use the symmetry to refine our bounds
    $M_{k_j} := \min \left\{ M_{k_j}, M_{n_j - k_j} \right\}; \quad \rho_{k_j} := \min \left\{ \rho_{k_j}, \rho_{n_j - k_j} \right\}; \quad \zeta_{k_j} := \min \left\{ \zeta_{k_j}, \zeta_{n_j - k_j} \right\};$

```
/* Step 4:  Compute indirect contributions using the 2D Dynamic
   Programming Algorithm                                        */
```
11 Use 3 calls to the 2D Dynamic Programming Algorithm 6 to generate arrays from $k, u$ to the maximum over the choice of $k_1, \ldots, k_m$ with total $k$ of which $u$ are non-zero of:

    1. Indirect CS Contribution: $\sum_{j \in [m]} \frac{M_{k_j}^2}{U_{k_j}}$

    2. Indirect XR Contribution: $\sum_{j \in [m]} \frac{M_{k_j} \rho_{k_j}}{U_{k_j}}$

    3. Indirect XZ Contribution: $\sum_{j \in [m]} \frac{M_{k_j} \zeta_{k_j}}{U_{k_j}}$

```
/* Step 5:  Compute direct contributions using the KU Triangle
   Inequality                                                   */
```
12 Use the KU Triangle Inequality 5 to also compute upper bounds on the Direct CS, XR, and XZ Contributions;

```
/* Step 6:  Compute final bounds and return the result         */
```
13 **return** $U_k, L_k = \text{Influences} \pm \max_u \frac{|\text{Direct XR} + \text{Indirect XR}| \times |\text{Direct XZ} + \text{Indirect XZ}|}{1 - \text{Direct CS} - \text{Indirect CS}}$

---

## A.5 SUPPLEMENTARY ALGORITHMS

We begin by an extension of the RTI Algorithm 2 that bounds the triangle inequality terms as a function of $k$ (number of removals) and $u$ (number of unique buckets from which samples were removed). The basic idea is the same as with the original RTI algorithm and requires only minor adaptations to track which row-column pairs are largest in their respective buckets.

---

**Algorithm 5:** KU Triangle Inequality

---

**Input:** Gram matrix $G$ of size $n \times n$ where $G_{ij} = \langle v_i, v_j \rangle$ representing an MSN-bounding
problem

**Output:** Matrix $V$ of size $u_{max} \times k_{max}$, where $V_{u,k}$ is an upper bound on the $\ell_2$ norm of the
sum of $k$ vectors taken from $u$ unique buckets.

```
/* Split each row of G into the largest and the non-largest
   entries per bucket.                                        */
```
1 Initialize $m$ as the number of buckets;
2 Initialize $n$ as the number of vectors;
3 Compute bucket indices from bucket sizes;
4 **for** $j \leftarrow 1$ **to** $m$ **do**
5      Sort entries in each bucket $j$ in decreasing order;
6      Store the largest entry of each bucket in $best\_entries$;
7      Remove these entries from the Gram matrix;

```
/* Compute the cumulative contributions of the largest and
   non-largest entries per bucket.                           */
```
8 Compute $best\_u\_contributions$ as cumulative sums of sorted $best\_entries$;
9 Compute $best\_kmu\_contributions$ as cumulative sums of the modified Gram matrix;
```
/* Compute the contributions of each sample for the triangle
   inequality.                                               */
```
10 Initialize $sample\_contributions$ as an array of $-\infty$;
11 **for** $u \leftarrow 1$ **to** $\min(u_{max}, k_{max})$ **do**
12      **for** $k \leftarrow u$ **to** $k_{max}$ **do**
13          Compute contributions by combining $best\_u\_contributions$ and
         $best\_kmu\_contributions$;

```
/* Enforce the constraint that T must use exactly u separate
   buckets.                                                  */
```
14 **for** $j \leftarrow 1$ **to** $m$ **do**
15      Move the largest elements of each bucket to the start using partition;
16      Copy the largest elements to $best\_contributions$;
17      Remove these elements from $sample\_contributions$;

18 Sort $best\_contributions$ and compute their cumulative sums as $cumsum\_best\_contributions$;
```
/* Compute the norms squared using the constraints.          */
```
19 Initialize $norms\_squared$ as an array of $-\infty$;
20 **for** $u \leftarrow 1$ **to** $u_{max}$ **do**
21      **for** $k \leftarrow u$ **to** $k_{max}$ **do**
22          Compute sum over $k - u$ largest elements of $sample\_contributions$ and $u$ largest
         elements of $cumsum\_best\_contributions$;
23          Update $norms\_squared[u, k]$;

```
/* Compute the norms and handle invalid values.             */
```
24 Compute $norms$ as the square root of $norms\_squared$;
25 **return** $norms$;

---

We also adapt Algorithm 3 to the case where we wish to keep track of both the total number of
removals and the number of unique buckets from which we remove samples.

---

**Algorithm 6:** Dynamic Programming 2D

---

**Input:** A list of bucket scores $B$

**Output:** A table $V$ where $V[u, k]$ is the maximal score for using $u$ unique buckets with total budget $k$

```
/* Initialize parameters and dynamic programming table       */
```
1   Compute cumulative sums of bucket lengths;
2   Initialize $dp\_table$ with $-\infty$ and set $dp\_table[:, 0, 0] = 0$;
```
/* Fill the dynamic programming table                         */
```
3   **for** $j \leftarrow 1$ **to** *length of* $B$ **do**
4      Get the current bucket $B[j]$;
5      Set $u$ as $\min(m, j)$;
6      Set $k$ as $\min(n, \text{cumsum\_bucket\_lengths}[j])$;
```
     /* Base case for the first bucket                        */
```
7      **if** $j == 0$ **then**
8         Set $dp\_table[0, 1, : k] = B[0][: k]$;
9      **else**
```
         /* Case where we do not update the table with new bucket
            values                                            */
```
10         Set $dp\_table[j, 1 : u, : k] = dp\_table[j - 1, 1 : u, : k]$;
```
         /* Case where we add values from the new bucket      */
```
11         **for** $\delta_k \leftarrow 1$ **to** $\min(\text{length of } B[j], k) - 1$ **do**
12            Update $dp\_table[j, 1 : u, \delta_k : k]$ with the maximum of the current value and
               $dp\_table[j - 1, : u - 1, : k - \delta_k] + B[j][\delta_k]$;

13   **return** $dp\_table[-1, :, :]$;

---

### A.6   PROOF OF CLAIM A.1

The main analytic idea we use in the OHARE algorithm is an alternative description of what happens when we perform a linear regression while controlling for a categorical feature. We formalise this process in Claim A.1, which we prove here:

*Proof of Claim A.1.* Let $X = (F \mid U)$ be the covariates for the original regression. Consider the Gram-Schmidt orthogonalization process, and define

$$\hat{X} = \left(F - D^{-1}U^\intercal F \mid U\right)$$

where $D = U^\intercal U$ is the diagonal matrix whose $j$th entry is $\|u_j\|$.

$\hat{X}$ now has a block-diagonal covariance matrix

$$\hat{\Sigma} = \hat{X}^\intercal \hat{X} = \begin{pmatrix} (F - D^{-1}U^\intercal F)^\intercal (F - D^{-1}U^\intercal F) & 0 \\ 0 & D \end{pmatrix}$$

Similarly, let $\hat{Y} = \hat{Y} = Y - D^{-1}U^\intercal Y$. These labels are now also perpendicular to the dummy variables $U$. Therefore, for $\hat{X} = F - D^{-1}U^\intercal F$, $\hat{\Sigma} = \hat{X}^\intercal \hat{X}$, we have

$$
\begin{aligned}
\hat{\beta} &= \hat{\Sigma}^{-1}\hat{X}^\intercal \hat{Y} \\
&= \begin{pmatrix} (F - D^{-1}U^\intercal F)^\intercal (F - D^{-1}U^\intercal F) & 0 \\ 0 & D \end{pmatrix}^{-1} \begin{pmatrix} F - D^{-1}U^\intercal F \\ U \end{pmatrix}^\intercal \left(Y - D^{-1}U^\intercal Y\right) \\
&= \begin{pmatrix} \widetilde{\Sigma}^{-1}\widetilde{X}^\intercal \widetilde{Y} \\ D^{-1}U^\intercal \left(Y - D^{-1}U^\intercal Y\right) \end{pmatrix} = \begin{pmatrix} \hat{\beta} \\ \star \end{pmatrix}
\end{aligned}
\tag{17}
$$

$\square$

## B  ADDITIONAL MSN-BOUNDING ALGORITHMS

### B.1  SPECTRAL DECOMPOSITION

At a high-level, this algorithm similarly tries to bound $\left\|\sum_{i \in T} Z_i\right\|^2 = 1_T^\mathsf{T} G 1_T$ for the same Gram matrix $G$, but here we use a spectral decomposition. We refine on the standard spectral algorithms which might output $\|1_T\| \max \lambda(G) = \sqrt{k} \max \lambda(G)$, by computing naive bounds on the inner products between $1_G$ and each of the top eigenvalues using a standard greedy algorithm.

---

**Algorithm 7:** Spectral Bound Algorithm

**Input:** A matrix $G$ of size $n \times n$ representing a Gram matrix of a set of vectors $G_{i,j} = \langle Z_i, Z_j \rangle$

**Output:** A vector $V$ whose $k$th entry is an upper bound on $\max_{|T|=k} \left\|\sum_{i \in T} Z_i\right\|$

1  Compute eigenvalues $\lambda_1 \geq \cdots \geq \lambda_n$ and corresponding eigenvectors $v_1, \ldots, v_n$ of $G$;

2  **for** $k = 1$ **to** $n-1$ **do**

```
/* For each of the eigenvectors, compute upper and lower
   bounds on αᵢ = ⟨1_T, vᵢ⟩.  To do this, we note that αᵢ is the
   sum of k entries of vᵢ, so it is bounded from below by ℓᵢ =
   sum over k smallest entries of vᵢ and from above by uᵢ =
   sum over k largest entries of vᵢ.                          */
```

3  $\quad$ Sort the entries of each $v_i$ in ascending order and store the cumulative sum in $\ell_i$;

4  $\quad$ Sort the entries of each $v_i$ in descending order and store the cumulative sum in $u_i$;

```
/* Compute upper bound on αᵢ² ≤ max{ℓᵢ², uᵢ²}.              */
```

5  $\quad$ Compute $b_i = \max\{\ell_i^2, u_i^2\}$;

```
/* Combine this with bound on overall weight of spectral
   decomposition ∑ᵢ αᵢ² = k.                               */
```

6  $\quad$ If $\sum_{j \leq i} b_i \geq k$, set $b_i = \max\{0, k - \sum_{j < i} b_i\}$;

7  $\quad$ Set $V_k = \sqrt{\sum_i \lambda_i b_i}$;

8  **return** $V$;

---

### B.2  GREEDY LOWER BOUND

In order to get some additional diagnostic capabilities and improve the interpretability of our results, we also implemented a simple greedy algorithm that helps us generate lower-bounds on the MSNs. Algorithm 8 initializes candidate set $T$ to contain the longest vector $v_i$ in our MSN instance, and greedily adds vectors to this set in an attempt to increase $\left\|\sum_{i \in T} v_i\right\|$ as fast as possible.

---

**Algorithm 8:** Greedy Algorithm for Lower Bound and Candidate Set

---

**Input:** A matrix $G$ of size $n \times n$ representing a Gram matrix of a set of vectors, where
  $G_{i,j} = \langle v_i, v_j \rangle$
**Output:** An ordering of the indices $T$, and a series of lower bounds
  $L_k = 1_{T:k}^\mathsf{T} G 1_{T:k} \leq \max_{T' \in \binom{[n]}{k}} 1_{T'}^\mathsf{T} G 1_{T'}$

1 Initialize an empty list $T := []$;
2 Initialize the scores array $\Delta$ where $\Delta_i = G_{i,i}$ for all $i \in [n]$;
  /* $\Delta$ maps each index $i \in [n]$ to the change in $1_T^\mathsf{T} G 1_T$ caused by
     adding $i$ to $T$.                                                    */
3 Initialize $L_0 := 0$;
4 **while** $|T| < n$ **do**
5 | Select $i := \text{argmax}_{j \notin T} \Delta_j$;
  | /* Choose the index $i$ not in $T$ with the maximum change in
  |    score.                                                           */
6 | Add $i$ to $T$ and update $L_k = L_{k-1} + \Delta_i$;
  | /* Update the lower bounds by adding the score of the newly
  |    added index.                                                     */
7 | Update the scores $\Delta := \Delta + 2 G_{i,:}$;
  | /* Adjust $\Delta$ to reflect the change in $1_T^\mathsf{T} G 1_T$ after adding $i$ to
  |    $T$.                                                             */
8 **return** $T$ and $L$;

---

In real-world datasets, we found that the upper bounds obtained by Algorithms 2 and 7 tend to be very close to the greedy lower bounds on their respective MSNs.

Moreover, when analyzing the Cash Transfer study (see Section C.3), we noticed that our analysis behaved poorly in some regimes. Using this greedy lower bound technique, we were able to identify that this behaviour was caused by a small number of households ($< 0.5\%$) dominating the variance in land-ownership ($> 80\%$), which was included in the study in linear scale. Converting the land-ownership to a logarithmic scale resolved this issue (allowing us to certify robustness to removing many more samples).

## C  Applied Experiments

Our main experimental results are detailed in Table 1:

| Paper | n | d | AMIP | KZC21 | OHARE | Runtime | Memory |
|---|---|---|---|---|---|---|---|
| Nightlights | 3895 | 209 | 136 | 110 | **29** | 25 s | 3.81 GiB |
| Cash Transfer | 4543 | 18 | 5 | 5 | **5** | 6 s | 1.78 GiB |
| | 3769 | 18 | 21 | 21 | **17** | 4 s | 2.80 GiB |
| | 4191 | 18 | 26 | 26 | **20** | 5 s | 1.73 GiB |
| | 10781 | 18 | 225 | 224 | **119** | 50 s | 11.20 GiB |
| | 9489 | 18 | 321 | 314 | **126** | 24 s | 5.37 GiB |
| | 10368 | 18 | 570 | 555 | **178** | 29 s | 6.59 GiB |
| OHIE | 23361 | 18 | 257 | 257 | **77** | 9 m 5 s | 34.19 GiB |
| | 23361 | 18 | 149 | 149 | **40** | 9 m 10 s | 43.50 GiB |
| | 23407 | 18 | 184 | 184 | **52** | 9 m 10 s | 43.55 GiB |
| | 21881 | 18 | 72 | 72 | **21** | 7 m 39 s | 38.64 GiB |
| | 21384 | 18 | 84 | 84 | **25** | 7 m 17 s | 45.77 GiB |
| | 21601 | 18 | 118 | 118 | **31** | 7 m 20 s | 46.21 GiB |
| | 23147 | 18 | 116 | 116 | **32** | 8 m 57 s | 51.53 GiB |

Table 1: A comparison of the lower bounds on $k_{\text{sign}}(e)$ produced by OHARE vs the corresponding upper bounds produced by AMIP and KZC. In all cases, no non-trivial lower bound was previously known. The runtimes listed are for a single core AMD processor.

Throughout this section, we will present the details of the data analysis and experimental procedures used to generate Table 1

## C.1    METHODOLOGY OF TABLE 1

In Table 1, we compare several bounds on $k_{\mathrm{sign}}(e)$ (the number of removals required to flip the sign of the "main" coefficient $e$) for a several regressions taken from highly influential econometrics datasets.

Before delving into the source of each regression we give a brief overview of the methodology used to construct each column of the table.

**Metadata**    The first few columns ("Paper", "Regression", $n$ and $d$) give some of the metadata for the regression.

The paper column details which paper each regression was drawn from, and for papers with multiple regressions we list the name of each regression in the paper. For more details about each of these papers see the following subsections.

The $n$ and $d$ columns list the number of samples and the dimension of each regression.

**AMIP**    The AMIP algorithm by Broderick et al. Broderick et al. (2020) can be used to estimate the robustness of a linear regression in one of two ways. The fastest option is to compute the AMIP gradients / influence scores, and estimate that the number of samples one must remove in order to change the fit by a certain amount by taking a cummulative sum over the sorted gradients.

However, this method both typically tends to overestimate the robustness of a dataset, and despite this does not even provide a formal upper bound on $k_{\mathrm{sign}}$ or $k_{2\sigma}$. Instead, for our experiments we sort the samples from most influential to least influential and remove them one at a time until the fit crosses the threshold we are considering.

**KZC**    The robustness estimation algorithm by Kusching et al. Kuschnig et al. (2021) utilizes a greedy approach – at each point we select the sample whose removal would have the greatest effect on our metric of interest and remove that sample. We repeat this process until the decision boundary has been crossed (in our case, until the sign of the regression coefficient has been flipped).

As with AMIP, we report the size of the smallest set produced by the algorithm that actually produced a sign flip.

**OHARE**    Assume WLOG that our coefficient of interest is positive in the original regression $\langle \beta, e \rangle > 0$ (otherwise we set $e' = -e$).

Running the OHARE algorithm yields a series of upper bounds $U_k$ on the removal effects. In particular, we know that for $k = k_{\mathrm{sign}}$, we must have $\Delta_k(e) > \langle \beta, e \rangle$, so by computing

$$k_{\mathrm{OHARE}}(e) = \min \{ k \in [n] \mid U_k > \langle \beta, e \rangle \}$$

yields a lower bound as $k_{\mathrm{sign}}(e) \geq k_{\mathrm{OHARE}}(e)$.

**Computational Resources**    In the columns "Runtime" and "Memory", we list the total runtime and memory cost of running the OHARE algorithm in this case. All experiments were run on a single core of an AMD EPYC 9654 96-Core Processor.

## C.2    HOW MUCH SHOULD WE TRUST THE DICTATOR'S GDP GROWTH ESTIMATES?

### C.2.1    PAPER OVERVIEW

Reliable estimates of GDP figures are crucial for analysts to assess the performance of an economy. However, leaders often have a variety of political and financial incentives manipulate GDP figures in order to improve their perception. Therefore, economists often use proxies to obtain independent estimates of these figures that may be harder to manipulate. One such method which has gained a lot of attention in recent years is to simply measure the amount of light emitted from a region at night (nightlights or NTL), as observed by satellite imaging Henderson et al. (2012).

Martinez Martinez (2022) uses this proxy in conjunction with several well-known measures for the democracy of a country (such as the freedom-in-the-world or FiW metric), in an effort to find evidence of GDP figure manipulation in autocratic regimes. One methods used by Martinez is to measure the "autocracy gradient in the NTL elasticity of GDP". In practice, this translates to running a regression of the form

$$\ln(\text{GDP})_{i,t} = \mu_i + \delta_t + \phi_0 \ln(\text{NTL})_{i,t} + \phi_1 \text{FiW}_{i,t} + \phi_2 \text{FiW}_{i,t}^2 + \phi_3 [\ln(\text{NTL}) \times \text{FiW}]_{i,t} + \varepsilon_{i,t} \quad (18)$$

where $i$ represents the index of a country, $t$ the year from which this sample was taken, $\mu_i$ and $\delta_t$ control for country-specific or time-specific effects, $\phi_0, \phi_1, \phi_2$ control for the direct correlation between nightlights and GDP, as well as for any $\leq$ 2nd order dependence between GDP and democracy. $\phi_3$ is the main effect we wish to observe, and a positive value of $\phi_3$ could be explained by autocratic regimes being more prone to embellishing their GDP figures.

Martinez reports a statistically significant positive value of $\phi_3$, as well as additional evidence for GDP figure manipulation in autocratic regimes (such as a larger difference between NTL-based estimates and reported GDP in years leading up to IMF evaluations of autocratic regimes). Martinez hypothesises that the separation of powers and cross-examination of figures by opposition contribute to making GDP figures more reliable in democratic countries.

### C.2.2 ROBUSTNESS

Martinez uses the AMIP tool Broderick et al. (2020) to assess the robustness of the regression (18). AMIP finds a set of 136 samples whose removal flips the sign of the OLS fit on the $[\ln(\text{NTL}) \times \text{FiW}]_{i,t}$ term. Kusching et al. Kuschnig et al. (2021) improve this upper bound on the stability by finding a set of 110 samples that flip the sign of the fit parameter.

Running our continuous regression toolkit on the Martinez dataset can only certify robustness to the removal of at most 7 samples, as the dataset contains a one-hot encoding of the country and it contains only 8 samples from Monetenegro. To overcome this, we apply our one-hot aware algorithms to the same data, and are able to certify the sign of this parameter in Martinez' regression requires at least $k_{\text{sign}} \geq 29$ to overturn.

### C.3 INDIRECT EFFECTS OF AN AID PROGRAM: HOW DO CASH TRANSFERS AFFECT INELIGIBLES' CONSUMPTION?

### C.3.1 PAPER OVERVIEW

Angelucci and De Giorgi Angelucci & De Giorgi (2009) study the indirect effect of the Progresa welfare program in Mexico. This program gave financial aid to eligible households within "treated" villages, and did not give aid to ineligible households or households from "untreated" villages.

Angelucci and De Giorgi then track spending patterns of eligible and ineligible households from both treated and untreated villages. They then use linear regressions to estimate the treatment effects on both eligible and ineligible households when controlling on various other features.

### C.3.2 MINOR DIFFICULTIES IN USING THE DATA

We found two different regression formula that have been attributed to this paper. The formula use slightly different control, with one option controlling for: head of household age, sex, literacy and education level, household poverty index and amount of land owned, local poverty index, number of households in the village and "average shock" (to the best of our knowledge, this was the regression used in Angelucci and De Giorgi's original paper). The regression used in the later paper by Broderick et al. Broderick et al. (2020) also controls for head of household marital status and which region the samples came from (using a one-hot encoding). Ultimately, these additional controls had very little effect on the fit parameters or their error bars, and we chose to focus on the latter to maintain consistency with previous benchmarks.

A more significant issue is that the dataset also contained many clear outliers. For instance, almost all the entries of the column for head of household sex were either "hombre" or "mujer" with a very small fraction (roughly 16 out of 59455 samples) having the value 9.0, and many columns had entries of "nr" (presumably no response).

We removed these "bad samples" for the sake of our analysis (both to avoid regressing over a clearly problematic dataset, and also because not doing so would cause the regression to include nearly empty categories in several different one-hot encodings which is beyond the scope of the OHARE algorithm). This had only a minor effect on the regression results.

| Column | Values Removed | Samples Removed |
|---|---|---|
| hhhsex | 9.0, nr | 16 |
| hhhalpha | nr | 21 |
| hhhspouse | nr, 2.0 | 546 |
| p16 | nr | 146 |
| hhhage | 97 y más, no sabe, nr | 205 |
| **Total** | | 921 |

Table 2: Number and values of outliers we removed from each of the covariates in Angelucci and De Giorgi's study Angelucci & De Giorgi (2009).

### C.3.3 ROBUSTNESS

We ran our tools on the 6 regressions from Angelucci and De Giorgi's paper (treatment effects on eligible / ineligible over 3 periods). Our robustness lower bounds nearly match the AMIP upper bound for the ineligible studies, but left room for improvement on the eligible regressions.

To find the root of this gap, we first noted that the main component of our bound that failed was that after removing $\approx 50$ samples, our bound on $\max\left\{\lambda(\Sigma_S^{-1})\right\}$ becomes very large. Using Algorithm 8, we were able to find the samples responsible for these loose bounds: $\approx 50$ households (out of $> 10000$) that account for more than $80\%$ of the variance in the amount of land owned by each household.

When one of the columns of a linear regression is heavy-tailed, it is not uncommon to replace this column with its logarithm. This presented a slight challenge in this case, as many of the household in the eligible studies owned no land at all. To overcome this, we replace the hectacres (amount of land owned) column with $\log(\text{hectacres} + \text{median hectacres})$. We then reran the regression and robustness analysis to see the effects of this change to produce the results in Table 1 and Figure 2.

### C.4 THE OREGON HEALTH INSURANCE EXPERIMENT: EVIDENCE FROM THE FIRST YEAR

#### C.4.1 PAPER OVERVIEW

The Oregon Health Insurance Experiment provides a unique opportunity to assess the effects of expanding access to public health insurance on low-income adults using a randomized controlled design. In 2008, Oregon implemented a lottery to select uninsured low-income adults to apply for Medicaid. This random assignment allows researchers to compare the outcomes of the treatment group (those selected by the lottery) with the control group (those not selected).

Finkelstein et al. Finkelstein et al. (2012) analyze data from the first year after the lottery to evaluate the impacts on health care utilization, financial strain, and health outcomes. The study finds that individuals in the treatment group were about 25 percentage points more likely to have health insurance compared to the control group. The results show that the treatment group experienced higher health care utilization, including increased primary and preventive care visits, and hospitalizations. They also faced lower out-of-pocket medical expenditures and medical debt, evidenced by fewer bills sent to collection agencies.

Moreover, the treatment group reported better physical and mental health than the control group. The authors suggest that the increase in health care utilization due to insurance coverage led to improved health outcomes and reduced financial strain. These findings provide significant evidence on the benefits of expanding Medicaid coverage to low-income populations.

#### C.4.2 REGRESSION ANALYSIS

The regression analysis conducted by Finkelstein et al. utilized instrumental variable (IV) regression to estimate the impact of Medicaid coverage on various health and financial outcomes. They used a

treatment variable as the instrument and several dummy variable controls corresponding to different medical metrics (e.g., "Not bad days physical").

IV regression is particularly useful in this context as it helps address potential endogeneity issues, where the treatment (Medicaid enrollment) might be correlated with unobserved factors affecting the outcomes. By using the lottery selection as an instrument for Medicaid enrollment, they ensured a more accurate estimation of the causal effect.

The IV regression can be computed as the ratio of two ordinary least squares (OLS) regressions: one estimating the relationship between the instrument (lottery selection) and the endogenous variable (Medicaid enrollment), and the other estimating the relationship between the instrument and the outcome (health or financial metrics). This implies that if both OLS regressions are robust to sign changes, so is their ratio. In this study, the correlation between the instrument and the endogenous variable was very strong and robust, making the primary robustness concern the OLS regression between the endogenous variable and the outcome.

Finkelstein et al. used weighted OLS regressions with several dummy variables, some of which had very sparse categories. This sparsity caused singularities after 13-14 removals, making it difficult to certify robustness with the ACRE algorithm. Therefore, the OHARE algorithm was used to certify the robustness of these regressions.

| Outcome | Type | ACRE | OHARE | AMIP | KZC21 |
|---|---|---|---|---|---|
| Health genflip | Instrument vs Endogenous | 14 | 752 | $\geq 10\%$ | $\geq 10\%$ |
| | Instrument vs Outcome | 14 | 77 | 257 | 257 |
| Health notpoor | Instrument vs Endogenous | 14 | 752 | $\geq 10\%$ | $\geq 10\%$ |
| | Instrument vs Outcome | 14 | 40 | 149 | 149 |
| Health change flip | Instrument vs Endogenous | 14 | 755 | $\geq 10\%$ | $\geq 10\%$ |
| | Instrument vs Outcome | 14 | 52 | 184 | 184 |
| Not bad days total | Instrument vs Endogenous | 13 | 685 | $\geq 10\%$ | $\geq 10\%$ |
| | Instrument vs Outcome | 13 | 21 | 72 | 72 |
| Not bad days physical | Instrument vs Endogenous | 13 | 669 | $\geq 10\%$ | $\geq 10\%$ |
| | Instrument vs Outcome | 13 | 25 | 84 | 84 |
| Not bad days mental | Instrument vs Endogenous | 13 | 676 | $\geq 10\%$ | $\geq 10\%$ |
| | Instrument vs Outcome | 13 | 31 | 118 | 118 |
| Nodep Screen | Instrument vs Endogenous | 14 | 742 | $\geq 10\%$ | $\geq 10\%$ |
| | Instrument vs Outcome | 14 | 32 | 116 | 116 |

# D  SYNTHETIC DATA EXPERIMENTS

## D.1  METHODOLOGY

We evaluate the performance of the ACRE algorithm on synthetic datasets drawn from two distinct distributions:

**Normally Distributed Regressions**   As a baseline, we test the algorithm on normally distributed covariates. Specifically, the covariates are drawn i.i.d. from a standard normal distribution, $\mathcal{N}(0, 1)$.

**Power-Law Distribution**   To stress-test the algorithm, we evaluate its performance on covariates drawn i.i.d. from a heavy-tailed power-law (Pareto) distribution. In this setting, we set the power $b = 4$ and draw covariates $X_{i,j}$ i.i.d. from a distribution with density:

$$f(x) \propto 1_{|x| \geq 1} \cdot |x|^{-(b+1)}.$$

This distribution has finite first, second, and third moments, but its fourth moment diverges, making it a challenging test case.

**Target Variable**   In both cases, the target variables (labels) $Y_i$ are drawn independently from a standard normal distribution: $Y_i \sim \mathcal{N}(0, 1)$.

## D.2 RESULTS

### D.2.1 COMPARISON OF KNOWN BOUNDS

We evaluate the performance of the ACRE algorithm on synthetic regression datasets drawn from the two distributions described earlier, with $n = 4000$ samples and $d = 50$ covariates. Specifically, we compare the upper bounds produced by ACRE, denoted $U_k$, with the state-of-the-art upper bounds by Freund and Hopkins, as well as the lower bounds produced by AMIP and KZC (see Figure 3).

As shown in the results, ACRE produces significantly tighter bounds than Freund and Hopkins across both normally distributed and power-law distributed covariates. Furthermore, ACRE's upper bounds closely approach the lower bounds from AMIP and KZC for values of $k$ up to removal effects that push the regression outside the $2\sigma$ confidence interval[2] $k_{2\sigma}$.

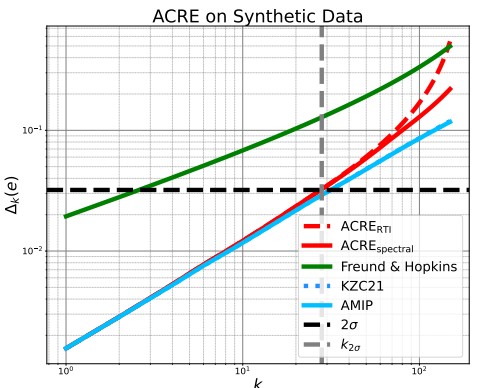
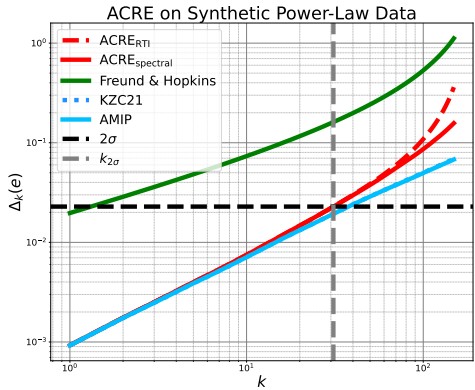

(a) Normally distributed covariates.   (b) Power-law distributed covariates.

Figure 3: Comparison of the upper bounds $U_k$ produced by the ACRE algorithm with the state-of-the-art bounds from Freund and Hopkins and the lower bounds from AMIP and KZC. Each dataset consists of $n = 4000$ samples and $d = 50$ covariates, with covariates drawn i.i.d. from either a normal distribution (Figure 3a) or a power-law distribution (Figure 3b). For each figure, we plot two versions of the ACRE algorithm: one that uses RTI as its MSN bounding component in the backend, and another that uses a spectral algorithm. On these synthetic datasets, the spectral bound slightly outperforms the RTI backend, but has a longer runtime.

### D.2.2 SCALING OF $k_{\text{threshold}}$ WITH $n$

In this experiment, we aim to analyze how $k_{\text{threshold}}$ (the maximal value of $k$ for which the bounds of ACRE are close to tight) scales with the number of samples $n$.

We fix the dimension $d = 20$ and draw $n_{\max} = 5000$ samples with covariates drawn i.i.d. from either a normal or a power-law distribution. The ACRE algorithm is then run on a series of regressions, corresponding to subsets of this dataset, with the number of samples varying from $n = 7d$ to $n = n_{\max}$.

For each regression, we compute the ACRE upper and lower bounds, $U_k$ and $L_k$, and use a heatmap to visualize the ratio $\frac{U_k}{L_k}$ as a function of $k$ and the sample size $n$. Contour plots indicate the regions where $\frac{U_k}{L_k}$ falls below specific thresholds (e.g., 1.1), highlighting the values of $k$ for which our bounds are close to tight.

As shown in Figure 4, for both normally and power-law distributed covariates, $k_{\text{threshold}}$ appears to scale approximately linearly with $n$. This is consistent with the scaling predicted by Theorem 1.2,

---

[2]$k_{\text{sign}}$ is not an appropriate metric for this experiment, as it is directly influenced by our choice of the ground truth model.

which gives:

$$k_{\text{threshold}} = \widetilde{\Omega}\left(\min\left\{\frac{n}{\sqrt{d}}, \frac{n^2}{d^2}\right\}\right) \approx \frac{n}{\sqrt{d}}.$$

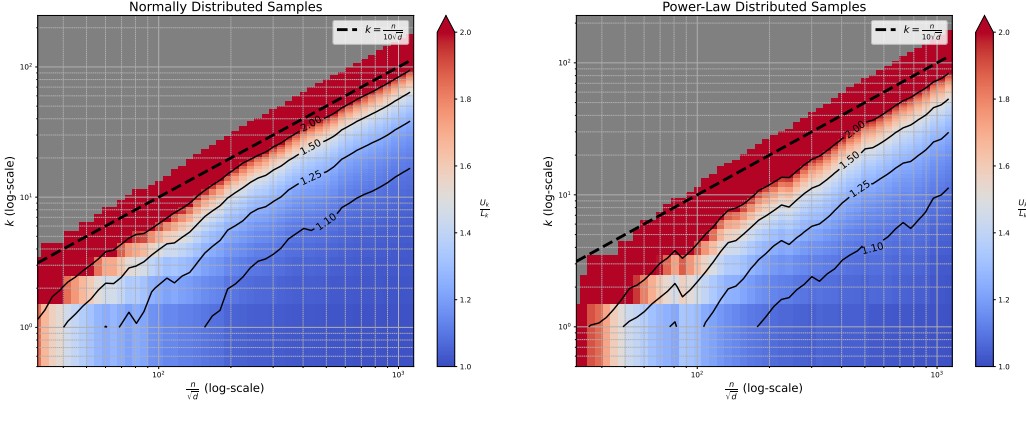

(a) Normally distributed covariates.  (b) Power-law distributed covariates.

Figure 4: The ratio $\frac{U_k}{L_k}$ between the upper and lower bounds produced by ACRE for synthetic linear regressions with covariates drawn from either a normal distribution (Figure 4a) or a power-law distribution (Figure 4b). Heatmaps show the tightness of the bounds as a function of $k$ and the number of samples $n$. Contour plots indicate regions where $\frac{U_k}{L_k}$ is below specific thresholds (e.g., 1.1), highlighting the scaling of $k_{\text{threshold}}$. In both cases, $k_{\text{threshold}}$ scales roughly linearly with $n$, consistent with the theoretical prediction of Theorem 1.2.

### D.2.3 SCALING OF $k_{\text{threshold}}$ WITH $d$

Finally, we examine how $k_{\text{threshold}}$ (the maximal value of $k$ for which the bounds of ACRE are close to tight) scales with the dimension $d$ of the regression for a fixed number of samples $n$.

In this experiment, we fix the number of samples to $n = 4000$ and sample a regression with $n$ samples and $d_{\max} = 500$ features, where the covariates are drawn i.i.d. from either a normal or a power-law distribution. We then vary $d$ by limiting each regression to subsets of the features, varying the dimension from $d = 5$ to $d = d_{\max}$.

Theoretically, we expect $k_{\text{threshold}}$ to scale as:

$$k_{\text{threshold}} = \widetilde{\Omega}\left(\min\left\{\frac{n}{\sqrt{d}}, \frac{n^2}{d^2}\right\}\right),$$

which predicts two regimes:

- For relatively small $d$, the first term dominates and we have $k_{\text{threshold}} \approx \frac{n}{\sqrt{d}}$ .

- For larger $d$, the second term dominates and we have $k_{\text{threshold}} \approx \frac{n^2}{d^2}$ .

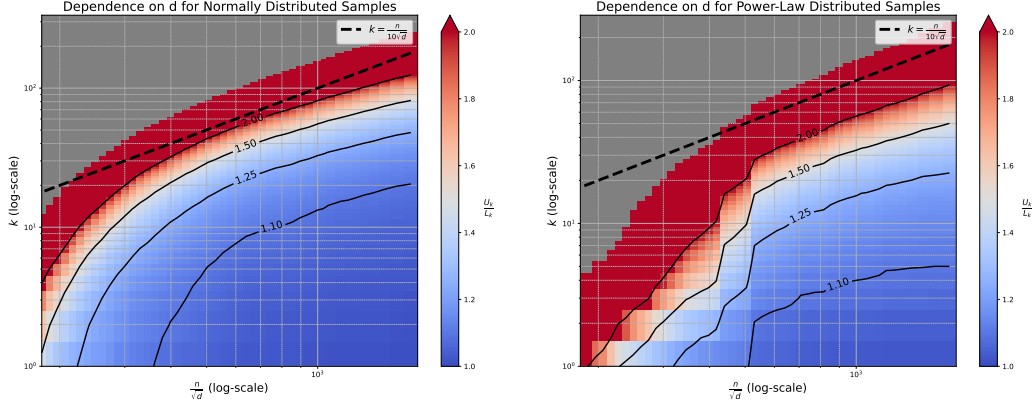

(a) Normally distributed covariates.

(b) Power-law distributed covariates.

Figure 5: The ratio $\frac{U_k}{L_k}$ between the upper and lower bounds produced by ACRE for synthetic linear regressions with covariates drawn from either a normal distribution (Figure 5a) or a power-law distribution (Figure 5b). Contour plots illustrate the values of $k_{\text{threshold}}$ for some given thresholds, as a function of the dimension $d$. For relatively small $d$ (right side of the plot), $k_{\text{threshold}}$ scales approximately as $\frac{n}{\sqrt{d}}$. Conversely, for larger $d$ (left side of the plot), $k_{\text{threshold}}$ decays faster, consistent with the $\frac{n^2}{d^2}$ scaling predicted by our theoretical analysis.

## E    TIGHTNESS OF ACRE

In this section, we will prove that the ACRE algorithm produces tight bounds on "well-behaved" distributions, proving Theorem 1.2.

### E.1    PRELIMINARIES

Throughout this section, we use $\widetilde{O}/\widetilde{\Theta}/\widetilde{\Omega}$ to denote big-O statements that hold up to a factor of polylog($n$). We will make no attempt to optimize the polylog($n$) factors in this analysis.

Moreover, we say that a term $\eta$ is *negligible* if $\eta^{-1} = n^{\omega(1)}$ is superpolynomial in $n$. Finally, we say an event happens with very high probability if it happens with probability $1 - \eta$ for negligible $\eta$.

### E.2    MAIN RESULT

Our main goal for this section will be to prove that ACRE produces good bounds with high probability when the regression data is drawn from a well behaved distribution:

**Theorem E.1** (ACRE Bounds are Tight on Well-behaved Data). *Let $X \in \mathbb{R}^{n \times d}, Y \in \mathbb{R}^n$ be a linear regression problem such that the covariates (i.e., the rows of $X$) are drawn iid from a well-behaved distribution $X_i \sim \mathcal{X}$ and the outcomes $Y$ are drawn iid from $Y \sim \mathcal{N}(X\beta_{\text{gt}}, I_n)$.*

*Then, for any axis $e \in \mathbb{S}^{d-1}$, with very high probability, the upper and lower bounds produced by ACRE on this regression are close to tight*

$$\frac{U_k}{L_k} = 1 + \widetilde{O}\left(\frac{d + k\sqrt{d}}{n}\right)$$

*for all $k < k_{\text{threshold}}$, where $k_{\text{threshold}} = \widetilde{\Theta}\left(\min\left\{\frac{n}{\sqrt{d}}, \frac{n^2}{d^2}\right\}\right)$*

In particular, if the samples $X_i, Y_i$ are drawn iid from some normal distribution $\mathcal{N}(0, \Sigma')$ (for some covariance $\Sigma' \in \mathbb{R}^{(d+1) \times (d+1)}$), or if the covariates $X_i$ are drawn iid from the hypercube or unit sphere and the target variable $Y_i \sim \langle \beta_{\text{gt}}, X_i \rangle + \mathcal{N}(0, \sigma_R)$ are drawn iid from a normal distribution around some ground truth model, then Theorem E.1 holds for them. Therefore, Theorem 1.2 follows from Theorem E.1.

Our goal for the rest of this section will be to prove Theorem E.1.

### E.3 PROOF SKETCH

First, we note that if $n = \widetilde{O}(d)$, Theorem E.1 holds vacuously, as $k_{\text{threshold}} < 1$. Therefore, we limit our analysis to the cases where $n > n_{\text{threshold}}$ for some $n_{\text{threshold}} = d \times \text{polylog}(n)$ (the exact power of this polylogarithmic factor will depend on the specific constant $C$ in the exponential decay assumption in Definition 1.

To prove Theorem E.1, we first define a set of condition under which we can prove that ACRE will produce good bounds:

**Definition 2** (ACRE-friendly). *Let $X, Y$ be the covariates and target variable of a regression. Let $\Sigma = X^\intercal X$ denote the unnormalized empirical covariance of the covariates, and let $R = Y - X\Sigma^{-1}X^T Y$ denote the residuals.*

*We say that this regression is* ACRE-friendly *for direction $e \in \mathbb{R}^d$ with $k$ removals and parameters $P_1, P_2, P_3, P_4, P_5 \geq 0$ if*

1. *The covariates are bounded in Mahalanobis distance $\max_{i \in [n]} X_i^\intercal \Sigma^{-1} X_i \leq P_1 \frac{d}{n}$.*

2. *The inner products of samples are bounded $\max_{i \neq j \in [n]} X_i^\intercal \Sigma^{-1} X_j \leq P_2 \frac{\sqrt{d}}{n}$.*

3. *The residuals are bounded $\max_{i \in [n]} R_i^2 \leq P_3 \sigma_R^2$, where $\sigma_R = \sqrt{\frac{1}{n} \sum_{i \in [n]} R_i^2}$.*

4. *The inner products between the covariates and the axis of interest are bounded $\max_{i \in [n]} \left(e^\intercal \Sigma^{-1} X_i\right)^2 \leq \frac{P_4}{n} \sum_{i \in [n]} \left(e^\intercal \Sigma^{-1} X_i\right)^2 = \frac{P_4}{n} e^\intercal \Sigma^{-1} e$.*

5. *Let $\alpha_i = e^\intercal \Sigma^{-1} X_i R_i$ be the AMIP influence scores. We require that the sum over the $k$ largest influence scores is at least*

$$a_k = \max_{T \in \binom{[n]}{k}} \left\{ \sum_{i \in T} \alpha_i \right\} \geq \frac{1}{P_5} \sigma_Z \sigma_R k$$

*where $\sigma_Z = \sqrt{\frac{1}{n} \sum_{i \in [n]} Z_i^2}$, for $Z_i = e^\intercal \Sigma^{-1} X_i$.*

*When $P_1, \ldots, P_5$ are at most polylogarithmic in $n$, we say that the regression is ACRE-friendly.*

Our proof of Theorem E.1 will have two main components. The first and smaller portion of the proof will be to show that the bounds produced by ACRE are close to tight when a regression is ACRE-friendly. It should not be surprising that Definition 2 is sufficient condition for producing good bounds with ACRE, since ACRE essentially checks a slightly more robust version of these conditions.

**Claim E.2** (ACRE bounds are nearly tight on ACRE-friendly regressions). *Let $X, Y, e$ be an ACRE-friendly regression with parameters $P_1, \ldots, P_5$ for all $k \leq k_0$. Then, there exists $k_{\text{threshold}} = \Theta \left( \min \left\{ \frac{n}{\sqrt{d}}, \frac{n^2}{d^2}, k_0 \right\} / \text{poly}\left(P_1, \ldots, P_5\right) \right)$ such that for all $k \leq k_{\text{threshold}}$, the bounds $L_k, U_k$ produced by* ACRE *satisfy*

$$\frac{U_k}{L_k} \leq 1 + O\left( \frac{d + k\sqrt{d}}{n} \times \text{poly}\left(P_1, \ldots, P_5\right) \right)$$

The second and longer portion of our proof will be devoted to showing that well-behaved distributions yield ACRE-friendly regressions with high probability.

**Claim E.3** (Well-behaved distributions yield ACRE-friendly regressions with high probability). *Let $n, d$ be as above, and let $X, Y$ be as in Theorem E.1.*

*Then, for any axis $e \in \mathbb{S}^{d-1}$, with very high probability, the regression $X, Y$ is ACRE-friendly for all $k \leq k_0$, for $k_0 = \widetilde{\Omega}(n)$.*

Combined, Claims E.2 and E.3 yield Theorem E.1.

### E.4 SYMMETRIES

Before proceeding to the proofs of Claims E.2 and E.3, we note that our definitions of well-behaved distributions and of ACRE-friendly regressions are normalized in a way that permits some symmetries. Indeed, it is easy to see that for any invertible matrix $L \in \mathbb{R}^{d \times d}$ and positive scalars $\alpha_e, \alpha_y \in \mathbb{R}^+$, a regression $X, Y, e$ is well-behaved / ACRE-friendly if and only if the regression $\widetilde{X} = XL, \widetilde{Y} = \alpha_y Y, \widetilde{e} = \alpha_e Le$ is well-behaved / ACRE-friendly. Moreover, such a reparametrizations also has no effect on the robustness of the regression, so we may apply this symmetry on the input and output of Theorem E.1 or Claims E.2 and E.3.

For Theorem E.1 or Claim E.3, we may apply this symmetry with $L = \Sigma^{-1/2}$, where $\Sigma$ is the *ground truth* covariance of the distribution $\mathcal{X}$, resulting in a distribution with covariance identity. Note that this renormalization is *not* the same as the renormalization process in the ACRE algorithm. In the ACRE algorithm, we renormalize by the empirical covariance, whereas the renormalization above is by the ground-truth covariance. One of the steps of our analysis will be to show that when the regression is drawn from a well-behaved distribution, the two are close to one another with high probability, but this is not immediate.

For the proof of Claim E.2, we renormalize the samples according to their empirical covariance. This would result in a set of samples $X$ such that $\Sigma^* = X^\intercal X = I$. Moreover, we use the $\alpha_Y$ symmetry to ensure that the residuals to have standard deviation $\sigma_R = 1$ and the $\alpha_e$ symmetry to ensure that $e$ has norm 1.

### E.5 PROOF OF CLAIM E.2

After this renormalization step above, the conditions on the normalized regression simplify to:

1. The covariates are bounded in $\ell_2$ norm $\max_{i \in [n]} \|X_i\|^2 \le P_1 \frac{d}{n}$

2. The inner products of samples are bounded $\max_{i \neq j \in [n]} \langle X_i, X_j \rangle \le P_2 \frac{\sqrt{d}}{n}$

3. The residuals are bounded $\max_{i \in [n]} R_i^2 \le P_3$.

4. The inner products between the covariates and the axis of interest are bounded $\max_{i \in [n]} \langle e, X_i \rangle^2 \le \frac{P_4}{n}$

5. Let $\alpha_i = \langle e, X_i \rangle R_i$ be the AMIP influence scores. We require that the sum over the $k$ largest influence scores is at least

$$ a_k = \max_{T \in \binom{[n]}{k}} \left\{ \sum_{i \in T} \alpha_i \right\} \ge \frac{1}{P_5} \times \frac{k}{\sqrt{n}} $$

We prove Claim E.2, working with the normalized regression. Recall that ACRE produces its bounds by combining RTI bounds on the following Gram matrices:

1. $G_{X \otimes X}$ whose entries are the squared entries of the Gram matrix $G_X$ of the original covariates. By our assumptions on the maximal norm and inner products of covariates, we know that the diagonal entries of this matrix are bounded by $\frac{d^2}{n^2} P_1^2$ and its off-diagonal entries are bounded by $\frac{d}{n^2} P_2^2$.

2. $G_{XR}$ whose entries are inner products between covariates multiplied by the product of two residuals. Therefore, the diagonal entries of this matrix are bounded (in absolute value) by $\frac{d}{n} P_1 P_3$ and its off-diagonal entries by $\frac{\sqrt{d}}{n} P_2 P_3$.

3. and $G_{XZ}$ whose entries are inner products between covariates, rescaled by the product of their weights on the axis of interest $e$. Therefore, its diagonal entries are bounded (in absolute value) by $\frac{d}{n^2} P_1 P_4$ and its off-diagonal entries by $\frac{\sqrt{d}}{n^2} P_2 P_4$.

Note that the output of the RTI algorithm is an upper bound on the resulting MSN problem. The output of the RTI algorithm squared is always equal to the sum of $k$ diagonal entries of the Gram matrix plus fewer than $k^2$ off-diagonal entries. Therefore, we have:

$$ \text{RTI Bound} \le \sqrt{k \times \text{Largest Diagonal Entry} + k^2 \times \text{Largest Off-Diagonal Entry}} \qquad (19) $$

**The $X \otimes X$ Term** Let $M_{X \otimes X}$ denote the MSN-bound obtained by running the RTI algorithm on the $G_{X \otimes X}$. Combining equation (19) with our knowledge of the diagonal and off-diagonal entries of $G_{X \otimes X}$, we clearly have:

$$M_{X \otimes X} \leq \sqrt{k \frac{d^2}{n^2} P_1^2 + k^2 \frac{d}{n^2} P_2^2}$$

Therefore, for all $k \leq k_{\text{threshold}} = \min\left\{\frac{n}{\sqrt{d} P_2 2}, \frac{n^2}{d^2 P_1^2 4}\right\}$, $M_{X \otimes X}$ is at most

$$M_{X \otimes X} \leq \sqrt{\frac{1}{2}} < 1$$

**The $XR$ and $XZ$ Terms** Similarly, let $M_{XR}$ and $M_{XZ}$ denote the MSN bounds obtained by the RTI algorithm on $G_{XR}$ and $G_{XZ}$ respectively. As before, we combine equation (19) with our bounds on the diagonal and off-diagonal terms of $G_{XR}$ and $G_{XZ}$ to show that

$$M_{XR} \leq \sqrt{k \frac{d}{n} P_1 P_3 + k^2 \frac{\sqrt{d}}{n} P_2 P_3} \leq \sqrt{k \frac{d}{n} P_1 P_3} + \sqrt{k^2 \frac{\sqrt{d}}{n} P_2 P_3}$$

and

$$M_{XZ} \leq \sqrt{k \frac{d}{n^2} P_1 P_4 + k^2 \frac{\sqrt{d}}{n^2} P_2 P_4} \leq \sqrt{k \frac{d}{n^2} P_1 P_4} + \sqrt{k^2 \frac{\sqrt{d}}{n^2} P_2 P_4}$$

**Wrapping Up** Therefore, for all $k \leq k_{\text{threshold}}$, we have

$$b_k = \frac{1}{1 - M_{X \otimes X}} M_{XR} M_{XZ} \leq \frac{\sqrt{2 P_3 P_4}}{\sqrt{2} - 1} \times \left(\frac{kd}{n^{3/2}} P_1 + 2 \frac{k^{3/2} d^{3/4}}{n^{3/2}} \sqrt{P_1 P_2} + \frac{k^2 \sqrt{d}}{n^{3/2}} P_2\right) = O\left(\frac{kd + k^2 \sqrt{d}}{n^{3/2}} \times \text{poly}(P_1, \ldots)\right.$$

Finally, applying the 5th condition, we have

$$\frac{U_k}{L_k} = \frac{a_k + b_k}{a_k - b_k} = 1 + O\left(\frac{b_k}{a_k}\right) = 1 + O\left(\frac{d + k\sqrt{d}}{n} \times \text{poly}(P_1, \ldots, P_5)\right)$$

completing the proof of Claim E.2.

### E.6 PROOF OF CLAIM E.3

We now move on to the main portion of the proof which will be devoted to showing that when the covariates $X_i \sim \mathcal{X}$ are drawn from a well-behaved distribution and the target variable $Y_i \sim N(X_i^\intercal \beta_{\text{gt}}, 1)$ are drawn from a normal distribution around some ground truth linear model, the resulting regression is ACRE-friendly with very high probability (i.e., to prove Claim E.3). Recall that as we showed in Section E.4, it suffices to prove this result for the case where the ground-truth covariance of the distribution $\mathcal{X}$ is equal to identity.

**Main Challenge and Proof Strategy** The main challenge will be that the requirements of Definition 2 (ACRE-friendly) require statements like $v^\intercal \Sigma^{-1} w = \text{small}$, where $v$ and $w$ are related to the samples of the regression (e.g., $v, w$ might be two samples $X_i, X_j$) and from here on out, $\Sigma = X^\intercal X$ denotes the unnormalized empirical covariance (recall that we normalized the ground truth covariance to be the identity).

Because of our assumption that $n > d \times \text{polylog}(n)$, we can use matrix Bernstein to prove that the empirical covariance is close to its expectation $\Sigma \approx nI$, but even for moderate dimensions $d$, our bounds on the overall error of this approximation (i.e., $\|\Sigma - nI\|$) would not be strong enough to prove Claim E.3. For any fixed $v, w \in \mathbb{R}^d$ that do not depend on the sample distributions, we can easily prove sufficiently strong concentration bounds on $v^\intercal \Sigma^{-1} w$. However, the $v, w$ pairs for which we will need to prove our concentration bounds *do* depend on the samples, creating the potential for an alignment between the large eigenvectors of $\Sigma - nI$ and this pair.

Our proof strategy will be to expand $\Sigma^{-1}$ into terms that depend on $v, w$ and terms that do not. The terms that do not depend on $v, w$ can be tightly bounded using simple concentration bounds, and the terms that do depend on $v, w$ will be so small to begin with that we can bound them very loosely using the matrix Bernstein inequality (see Lemma E.4).

### E.6.1 MATRIX BERNSTEIN INEQUALITY

Throughout our analysis we will often make use of the matrix Bernstein inequality:

**Lemma E.4** (Matrix Bernstein Tropp et al. (2015)). *Let $Z_1, \ldots, Z_n$ be independently-distributed random symmetric $d \times d$ matrices with mean $\mathbb{E}[Z_i] = 0$. Moreover, assume that $\|Z_i\| \leq L$ and $\left\|\sum_{i \in [n]} \mathbb{E}[Z_i^2]\right\| \leq \sigma^2$, then*

$$\Pr\left[\left\|\sum_{i \in [n]} Z_i\right\| \geq t\right] \leq 2d \cdot \exp\left(\frac{-t^2/2}{\sigma^2 + Lt/3}\right)$$

In many cases however, our input will not directly fit the assumptions of this inequality and instead of a statement of the form $\|Z_i\| \leq L$, we will only have a statement of the form $\Pr[\|Z_i\| \geq L] \leq \delta$ for some negligible probability $\delta$. Therefore it will be useful for us to have an extension of this theorem to cases when the assumption fails, but with a very small probability:

**Lemma E.5** (Approximate Matrix Bernstein). *Let $Z_1, \ldots, Z_n$ be independently-distributed random symmetric $d \times d$ matrices with mean $\mathbb{E}[Z_i] = 0$. Moreover, assume that for all $i$, $\Pr[\|Z_i\| \geq L] \leq \delta$ and $\left\|\mathbb{E}[Z_i^2]\right\| \leq \tau^2$, and that $\left\|\sum_{i \in [n]} \mathbb{E}[Z_i^2]\right\| \leq \sigma^2$, then*

$$\Pr\left[\left\|\sum_{i \in [n]} Z_i\right\| \geq t + n\tau\sqrt{\delta}\right] \leq 2d \cdot \exp\left(\frac{-t^2/2}{\sigma^2 + Lt/3}\right) + n\delta$$

*Proof of Lemma E.5.* Let $\mathcal{S}^d = \left\{Z \in \mathbb{R}^{d \times d} \big| Z = Z^T\right\}$ denote the space of symmetric real valued $d \times d$ matrices.

Consider the random variable $\zeta_i = Z_i \times 1_{\|Z_i\| \leq L}$. At first it might seem like we can directly apply the matrix Bernstein inequality to these new variables, and then use the fact that $Z_i = \zeta_i$ for all $i$ w.p. $\geq 1 - n\delta$.

The issue is that we no longer know that these $\zeta_i$ maintain the other assumptions of the matrix Bernstein inequality. In particular, we will bound $\|\mathbb{E}[\zeta_i]\|$ and $\left\|\sum_{i \in [n]} \mathbb{E}[\zeta_i^2]\right\|$ from above.

Let $f_i$ denote the probability density function of $Z_i$, and consider

$$\mathbb{E}[\zeta_i] = \mathbb{E}[\zeta_i] - \mathbb{E}[Z_i] = \int_{\|Z\| > L} Z f_i(Z)$$

Let $v, w \in \mathbb{S}^{d-1}$ be any two points on the unit sphere. Using the CS inequality, we have

$$v^\mathsf{T}\mathbb{E}[\zeta_i]w = \int_{\|Z\|>L} v^\mathsf{T} Z w f_i(Z) = \int_{Z \in \mathcal{S}^d} 1_{\|Z\|>L} v^\mathsf{T} Z w f_i(Z) \leq \sqrt{\int_{Z \in \mathcal{S}^d} 1_{\|Z\|>L} f_i(Z)} \times \sqrt{\int_{Z \in \mathcal{S}^d} (v^\mathsf{T} Z w)^2 f_i(Z)}$$

The first term in the RHS is bounded by

$$\sqrt{\int_{Z \in \mathcal{S}^d} 1_{\|Z\|>L} f_i(Z)} = \sqrt{\Pr[\|Z_i\| \geq L]} \leq \sqrt{\delta}$$

For the latter term, we use the CS inequality again:

$$\sqrt{\int_{Z \in \mathcal{S}^d} (v^\mathsf{T} Z w)^2 f_i(Z)} \leq \sqrt{\int_{Z \in \mathcal{S}^d} \|Zv\|^2 \|w\|^2 f_i(Z)} = \sqrt{\int_{Z \in \mathcal{S}^d} v^\mathsf{T} Z^2 v f_i(Z)} = \sqrt{v^\mathsf{T}\mathbb{E}[Z_i^2]v} \leq \tau$$

Therefore

$$\|\mathbb{E}[\zeta_i]\| \leq \tau\sqrt{\delta}$$

Finally, we bound the change in the variance of these variables.

$$\mathrm{Cov}\left(\zeta_i\right) = \mathbb{E}\left[\zeta_i^2\right] - \mathbb{E}\left[\zeta_i\right]^2 \preceq \mathbb{E}\left[\zeta_i^2\right] \preceq \mathbb{E}\left[Z_i^2\right] \Rightarrow \left\|\sum_{i\in[n]} \mathrm{Cov}\left(\zeta_i\right)\right\| \leq \left\|\sum_{i\in[n]} \mathbb{E}\left[Z_i^2\right]\right\| \leq \sigma^2$$

Therefore, we may apply the matrix Bernstein inequality (Lemma E.4) on the matrices $z_i = \zeta_i - \mathbb{E}\left[\zeta_i\right]$ to obtain the bound

$$\mathrm{Pr}\left[\left\|\sum_{i\in[n]} \zeta_i\right\| \geq t + n\tau\sqrt{\delta}\right] \leq \mathrm{Pr}\left[\left\|\sum_{i\in[n]} \zeta_i - \sum_{i\in[n]} \mathbb{E}\left[\zeta_i\right]\right\| \geq t\right] \leq 2d \cdot \exp\left(\frac{-t^2/2}{\sigma^2 + Lt/3}\right),$$

yielding the main claim.

$\square$

### E.6.2 A Useful Corollary of Lemma E.5

Recall that we defined a "well-behaved distribution" to be a distribution whose tails decay rapidly when projected on any direction. It will be beneficial to our use-case to work with a more relaxed condition on the decay of these tails, by allowing an additional poly($n$) factor. Note that any well-behaved distribution is clearly also almost well-behaved.

**Definition 3.** *We say that a mean-zero distribution $\mathcal{X}$ on $\mathbb{R}^d$ is* almost well-behaved *with respect to the scaling parameter $n$, if it has exponentially decaying tails in the sense that*

$$\exists C > 0 \ \ \forall v \in \mathbb{S}^{d-1}, t > 0 \quad \mathrm{Pr}_{X\sim\mathcal{X}}\left[\left|\left\langle v, \Sigma^{-1/2}X\right\rangle\right| > t\right] \leq \mathrm{poly}(n) \times \exp\left(-\Omega\left(t^C\right)\right)$$

*where*

$$\Sigma = \mathbb{E}_{X\sim\mathcal{X}}\left[XX^{\mathsf{T}}\right] = \mathrm{Covariance}(\mathcal{X})$$

In the previous section, we proved a generalisation of the matrix Bernstein inequality that can deal with a small probability that the norm bound assumption of the Bernstein inequality is violated. A corollary of this result is that almost well-behaved distributions are closed to summation over polynomially many iid samples. More concretely:

**Lemma E.6.** *Let $\mathcal{X}$ be a distribution that is almost well-behaved with respect to the scaling parameter $n$. Let $k = \mathrm{poly}(n)$ and let $X_1, \ldots, X_k \sim \mathcal{X}$ be iid random variables in $\mathbb{R}^d$, drawn from $\mathcal{X}$.*

*Then, the distribution of their empirical mean (i.e., of the variable $\hat{X} = \frac{1}{k}\sum_{i\in[k]} X_i$) is also almost well-behaved.*

*Proof of Lemma E.6.* Let $\hat{\mathcal{X}}$ denote the distribution of $\hat{X}$.

First, we note that from linearity of expectation $\mathbb{E}\left[\hat{X}\right] = \frac{1}{k}\sum_{i\in[k]} \mathbb{E}\left[X_i\right] = 0$, so $\hat{\mathcal{X}}$ is indeed a mean zero distribution. Let $\Sigma = \mathbb{E}_{X\sim\mathcal{X}}\left[XX^{\mathsf{T}}\right]$. Again, using linearity of expectation, we have

$$\hat{\Sigma} = \mathbb{E}_{X\sim\hat{\mathcal{X}}}\left[XX^{\mathsf{T}}\right] = \frac{1}{k^2}\sum_{i\in[k]} \Sigma = \frac{1}{k}\Sigma$$

Finally, we need to show that the tails of $\hat{\mathcal{X}}$ are bounded. Let $v \in \mathbb{S}^{d-1}$ be any vector on the unit sphere. From our assumption that $\mathcal{X}$ is well-behaved, for any $t > 0$, we have

$$\mathrm{Pr}_{X\sim\mathcal{X}}\left[\left|\left\langle v, \Sigma^{-1/2}X\right\rangle\right| \geq t\right] \leq \mathrm{poly}(n) \times \exp\left(-\Omega\left(t^C\right)\right)$$

Now, consider

$$Z = \left\langle v, \hat{\Sigma}^{-1/2}\hat{X}\right\rangle = \frac{1}{\sqrt{k}}\sum_{i\in[k]} \left\langle v, \Sigma^{-1/2}X_i\right\rangle.$$

We would like to bound the probability that $|Z|$ is greater than some threshold $T$.

Let $Z_i = \langle v, \Sigma^{-1/2} X_i \rangle$. These are iid random variables and our goal is to prove a concentration bound on their sum, so we may try to use Bernstein-type inequalities to do so.

Since $\mathbb{E}[X_i] = 0$ and $\mathbb{E}[X_i X_i^\mathsf{T}] = \Sigma$, we have that $\mathbb{E}[Z_i] = 0$ and $\mathbb{E}[Z_i]^2 = 1$. Moreover, from the assumption that $\mathcal{X}$ is well behaved, we have concentration bounds on these individual variables. Indeed, for any $t$, we have that

$$\delta_t \stackrel{\text{def}}{=} \Pr[|X_i| \geq t] = \text{poly}(n) \times \exp\left(-\Omega\left(t^C\right)\right)$$

Therefore, from the $d = 1$ dimensional case of the approximate matrix Bernstein inequality (Lemma E.5), we have that

$$\Pr\left[\left|\left\langle v, \hat{\Sigma}^{-1/2} \hat{\mathcal{X}} \right\rangle\right| \geq \tau + k\sqrt{\delta_t}\right] \leq \exp\left(-\frac{\frac{1}{2}\tau^2 k}{k + \frac{1}{3} t\tau\sqrt{k}}\right) + k \times \text{poly}(n) \times \exp\left(-\Omega\left(t^C\right)\right)$$

Setting $t = \sqrt{\frac{T}{2}}$, we consider two regimes. When $t < t_{\text{threshold}} = \Theta\left(\log(n)^{1/C}\right)$, the bound

$$\Pr\left[\left|\left\langle v, \hat{\Sigma}^{-1/2} \hat{X} \right\rangle\right| \geq T\right] \leq \text{poly}(n) \times \exp\left(-\Omega\left(t^C\right)\right)$$

holds vacuously, since the RHS is greater than $1$ and the LHS is a probability.

When $t \geq t_{\text{threshold}}$, we can ensure that $\delta_t < 1/k^2 = 1/\text{poly}(n)$. Therefore in this case for $\tau = t^2$, we have

$$\Pr\left[\left|\left\langle v, \hat{\Sigma}^{-1/2} \hat{X} \right\rangle\right| \geq T\right] \leq \Pr\left[\left|\left\langle v, \hat{\Sigma}^{-1/2} \hat{X} \right\rangle\right| \geq \tau + k\sqrt{\delta_t}\right] \leq \exp\left(-\Omega\left(\tau^{1/2}\right)\right) + k \times \text{poly}(n) \times \exp\left(-\Omega\left(\tau^{C/2}\right)\right)$$

completing our proof of Lemma E.6.

$\square$

### E.6.3    RENORMALIZATION

Our goal in this portion of the proof will be to show that the empirical covariance matrix $\Sigma = X^\mathsf{T} X$ is not far from the expectation $\Sigma_{\text{gt}} = \mathbb{E}_{X \sim \mathcal{X}^n}[X^\mathsf{T} X] = nI$. In particular, we prove the following claim:

**Claim E.7.** *Let $X_1, \ldots, X_n \sim \mathcal{X}$ be samples drawn iid from an almost well-behaved distribution $\mathcal{X}$ with covariance identity. Let $\Sigma = \sum_{i \in [n]} X_i X_i^\mathsf{T}$ denote the unnormalized empirical covariance, and set $\Sigma_{\text{gt}} \stackrel{\text{def}}{=} \mathbb{E}[\Sigma] = nI$. Then,*

*With very high probability, $\|\Sigma - \Sigma_{\text{gt}}\| < \widetilde{O}\left(\sqrt{nd} + d\right) = o(n)$.*

*Moreover, for all $i$, we have $\left\|\mathbb{E}\left[\|X_i\|^2 X_i X_i^\mathsf{T}\right]\right\| = O(d)$ and with very high probability $\|X_i\|^2 = \widetilde{O}(d)$.*

*Proof of Claim E.7.* We prove Claim E.7 using our adaptation of the matrix Bernstein inequality (see Lemma E.5). We will apply this inequality for the $Z_i = X_i X_i^\mathsf{T} - I_d$, allowing us to obtain probabilistic bounds on

$$\Sigma - \Sigma_{\text{gt}} = \sum_{i=1}^n Z_i$$

To use the approximate matrix Bernstein, we begin by proving a bound on the norm of $\max_i \|Z_i\|$ that holds with probability $1 - \frac{1}{\text{superpoly}(n)}$. Note that

$$\|Z_i\| \leq \|X_i X_i^\mathsf{T}\| + \|I\| = \|X_i\|^2 + 1 = \left(\sum_{j \in [d]} \langle e_j, X_i \rangle^2\right) + 1$$

Next, we use the fact that $X_i$ is drawn from a well-behaved distribution $\mathcal{X}$ to prove strong tail bounds on the distribution of its norm. In particular for any primary axis $e_j$ (for $j \in [d]$), we have $\langle e_j, X_i \rangle \leq \mathrm{polylog}(n)$ with very high probability. Therefore, using the union bound over all $i, j$, we also have that with very high probability

$$\max_{\substack{i \in [n] \\ j \in [d]}} \left\{ \langle e_j, X_i \rangle^2 \right\} \leq \mathrm{polylog}\,(n)\ .$$

Therefore, with very high probability

$$\|Z_i\| \leq \widetilde{O}\,(d) \tag{20}$$

Next, we consider the second moment of $Z_i$.

$$\mathbb{E}\left[Z_i^2\right] = \mathbb{E}\left[\|X_i\|^2 X_i X_i^\mathsf{T} - 2 X_i X_i^\mathsf{T} + I\right] = \mathbb{E}\left[\|X_i\|^2 X_i X_i^\mathsf{T}\right] - I$$

Fix some pair of primary axis $e_s$ and unit vector $v \in \mathbb{S}^{d-1}$. Recall that by our definition of $\mathcal{X}$ being a well-behaved distribution, we have an exponentially decaying concentration bound on the projection of our samples onto either of these axes

$$\Pr_{X \sim \mathcal{X}}\left[\max\left\{|\langle X, e_s\rangle|, |\langle X, v\rangle|\right\} \geq t\right] = \exp\left(-\Omega\left(t^C\right)\right)$$

Therefore, a similar exponential tail bound also holds on the square of the product of these projections

$$\Pr_{X \sim \mathcal{X}}\left[|\langle X, e_s\rangle|^2 |\langle X, v\rangle|^2 \geq t\right] = \exp\left(-\Omega\left(t^{C/4}\right)\right)$$

In particular, we can conclude the far milder bound that the expectation of the squared product of these projections has bounded mean:

$$\mathbb{E}_{X \sim \mathcal{X}}\left[|\langle X, e_s\rangle|^2 |\langle X, v\rangle|^2\right] = \int_{t \in [0, \infty)} \Pr_{X \sim \mathcal{X}}\left[|\langle X, e_s\rangle|^2 |\langle X, v\rangle|^2 \geq t\right] \leq \int_{t \in [0, \infty)} \exp\left(-\Omega\left(t^{C/4}\right)\right) = O(1) \tag{21}$$

Note that equation (21) is no longer a concentration bound that holds with high probability, but a bound on the expectation of a random variable, that holds for *any* such pair $v, e_s$. In particular, we have

$$\max_{v, e_s \in \mathbb{S}^{d-1}} \left\{ \mathbb{E}_{X \sim \mathcal{X}}\left[|\langle X, e_s\rangle|^2 |\langle X, v\rangle|^2\right] \right\} = O(1)$$

Therefore, for any $v \in \mathbb{S}^{d-1}$,

$$v^\mathsf{T} \mathbb{E}\left[\|X_i\|^2 X_i X_i^\mathsf{T}\right] v = \mathbb{E}\left[\langle v, X_i\rangle^2 \|X_i\|^2\right] = \mathbb{E}\left[\sum_{s \in [d]} \langle v, X_i\rangle^2 \langle e_s, X_i\rangle^2\right] = \sum_{s \in [d]} \mathbb{E}\left[\langle v, X_i\rangle^2 \langle e_s, X_i\rangle^2\right] = O(d)$$

and since this holds for all $v$, it is also true when maximizing over the unit sphere

$$\left\|\mathbb{E}\left[\|X_i\|^2 X_i X_i^\mathsf{T}\right]\right\| = \max_{v \in \mathbb{S}^{d-1}} \left\{v^\mathsf{T} \mathbb{E}\left[\|X_i\|^2 X_i X_i^\mathsf{T}\right] v\right\} = O(d)$$

Therefore

$$\left\|\sum_{i \in [n]} \mathbb{E}\left[Z_i^2\right]\right\| \leq \sum_{i \in [n]} \left\|\mathbb{E}\left[Z_i^2\right]\right\| = O(nd) \tag{22}$$

Combining equations (20) and (22) with Lemma E.5 yields Claim E.7 $\qquad\square$

### E.6.4 MAXIMAL NORM

We now proceed to prove that each of the conditions required for the regression to be well-behaved occurs with very high probability. The first (and easiest to prove) is the condition that $\max_{i \in [n]} X_i^\intercal \Sigma^{-1} X_i$ is bounded.

**Claim E.8.** *Let $X_1, \ldots, X_n \sim \mathcal{X}$ be samples drawn iid from an almost well-behaved distribution $\mathcal{X}$ with covariance identity. Let $\Sigma = \sum_{i \in [n]} X_i X_i^\intercal$ denote the unnormalized empirical covariance, and set $\Sigma_{\text{gt}} \stackrel{\text{def}}{=} \mathbb{E}[\Sigma] = nI$.*

*Then, with very high probability*

$$\max_{i \in [n]} \left\{ X_i^\intercal \Sigma^{-1} X_i \right\} = \widetilde{O}\left( \frac{d}{n} \right)$$

*Proof of Claim E.8.* Let $\lambda$ denote the spectrum of a matrix. In the proof of Claim E.7, we already showed that with very high probability $\|\Sigma - \Sigma_{\text{gt}}\| = o(n) = o\left(\min \lambda\left(\Sigma_{\text{gt}}\right)\right)$. When this holds, we also have

$$\lambda\left(\Sigma^{-1}\right) \subseteq (1 \pm o(1)) \times \frac{1}{n}.$$

Moreover, the second part of Claim E.7 states that with very high probability $\max_{i \in [n]} \|X_i\|^2 = \widetilde{O}(d)$.

Combining these results, we have that with very high probability

$$\max_{i \in [n]} \left\{ X_i^\intercal \Sigma^{-1} X_i \right\} \le \max \lambda\left(\Sigma^{-1}\right) \times \max_{i \in [n]} \left\{ \|X_i\|^2 \right\} = \widetilde{O}\left( \frac{d}{n} \right)$$

$\square$

### E.6.5 BOUNDED INNER PRODUCTS

For the next step of our proof, we show that the second condition of ACRE-friendliness of the regression holds with very high probability. In particular, we will show that

**Claim E.9.** *Let $X_1, \ldots, X_n \sim \mathcal{X}$ be samples drawn iid from an almost well-behaved distribution $\mathcal{X}$ with covariance identity. Let $\Sigma = \sum_{i \in [n]} X_i X_i^\intercal$ denote the unnormalized empirical covariance, and set $\Sigma_{\text{gt}} \stackrel{\text{def}}{=} \mathbb{E}[\Sigma] = nI$.*

*Then, with very high probability*

$$\max_{i \ne j \in [n]} \left\{ \left| X_i^\intercal \Sigma^{-1} X_j \right| \right\} = \widetilde{O}\left( \frac{\sqrt{d}}{n} \right).$$

*Proof of Claim E.9.* Denote $A = \Sigma - X_i X_i^\intercal = \Sigma_{[n] \setminus \{i\}}$, and $v = X_i$. Note that from Claim E.7, we know that with very high probability $v^\intercal A^{-1} v \le \|X_i\|^2 \|A^{-1}\| = \widetilde{O}\left(\frac{d}{n}\right) = o(1)$.

Therefore, in this regime we may apply the Sherman-Morrison formula to show that

$$\Sigma^{-1} = (A + vv^\intercal)^{-1} = A^{-1} - \frac{A^{-1} vv^\intercal A^{-1}}{1 + v^\intercal A^{-1} v^\intercal}.$$

Note that neither $A$ nor $X_j$ depend on $X_i$, so from our assumption that $\mathcal{X}$ is well behaved, with very high probability

$$\left| X_i^\intercal A^{-1} X_j \right| = \widetilde{O}\left(\left\| A^{-1} X_j \right\|\right) = \widetilde{O}\left( \frac{\sqrt{d}}{n} \right).$$

Similarly, for our target expression, with very high probability,

$$\left|X_i^\intercal \Sigma^{-1} X_j\right| = \left|X_i^\intercal \left(A^{-1} - \frac{A^{-1} X_i X_i^\intercal A^{-1}}{1 + X_i^\intercal A^{-1} X_i}\right) X_j\right| =$$

$$= \left(1 - \frac{X_i^\intercal A^{-1} X_i}{1 - X_i^\intercal A^{-1} X_i}\right) \left|X_i^\intercal A^{-1} X_j\right| = \widetilde{O}\left(\frac{\sqrt{d}}{n}\right)$$

$\square$

### E.6.6  Projection on the $e$ Axis

For the next property of well-behaved regressions, we will want to show that with very high probability $e^\intercal \Sigma^{-1} X_i$ is bounded for all $i$. Note that from the exponential decay assumption due to $\mathcal{X}$ being well-behaved would suffice to give a good bound on $e^\intercal \hat{\Sigma}^{-1} X_i$, but as before, the challenge will be to show that $\Sigma^{-1}$ doesn't rotate $X_i$ onto $e$.

**Claim E.10.** *Let $X_1, \ldots, X_n \sim \mathcal{X}$ be samples drawn iid from an almost well-behaved distribution $\mathcal{X}$ with covariance identity. Let $\Sigma = \sum_{i \in [n]} X_i X_i^\intercal$ denote the unnormalized empirical covariance, and set $\Sigma_{\mathrm{gt}} \stackrel{\text{def}}{=} \mathbb{E}[\Sigma] = nI$. Let $e \in \mathbb{S}^{d-1}$ be any fixed vector independent of the $X_i$.*

*Then, with very high probability*

$$\forall i \in [n] \quad e^\intercal \Sigma^{-1} X_i = \frac{e^\intercal X_i}{n} \pm o\left(\frac{1}{n}\right).$$

*In particular, with very high probability*

$$\max_{i \in [n]} \left|\langle \Sigma^{-1} X_i, e \rangle\right| = \widetilde{O}\left(\frac{1}{n}\right) = \sqrt{\widetilde{O}\left(\frac{1}{n}\right) \times e^\intercal \Sigma^{-1} e}$$

*Proof of Claim E.10.* As in the proof of Claim E.9, let $v = X_i$ and $A = \Sigma - vv^\intercal$. Moreover, because with very high probability $v^\intercal A^{-1} v = o(1) < 1$, we may apply the Sherman-Morrison formula

$$\Sigma^{-1} = (A + vv^\intercal)^{-1} = A^{-1} - \frac{A^{-1} vv^\intercal A^{-1}}{1 + v^\intercal A^{-1} v^\intercal}.$$

Therefore, with high very high probability,

$$e^\intercal \Sigma^{-1} X_i = e^\intercal A^{-1} X_i - \frac{e^\intercal A^{-1} X_i X_i^\intercal A^{-1} X_i^\intercal}{1 + X_i^\intercal A^{-1} X_i} =$$

$$= e^\intercal \Sigma_{\mathrm{gt}}^{-1} X_i + e^\intercal \left(A^{-1} - \Sigma_{\mathrm{gt}}^{-1}\right) X_i - \frac{e^\intercal A^{-1} X_i X_i^\intercal A^{-1} X_i^\intercal}{1 + X_i^\intercal A^{-1} X_i}$$

From Claim E.7, we know that with very high probability

$$\left\|A^{-1} - \Sigma_{\mathrm{gt}}^{-1}\right\| \le \frac{1}{2n^2} \left\|A - \Sigma_{\mathrm{gt}}\right\| = \widetilde{O}\left(\frac{d + \sqrt{nd}}{n^2}\right)$$

$$X_i^\intercal A^{-1} X_i \le \|X_i\|^2 \left\|A^{-1}\right\| = \widetilde{O}\left(\frac{d}{n}\right).$$

Therefore, using the fact that $X_i$ is well-behaved and independent of $e$ and $A$, we have that with very high probability

$$\left|e^\intercal \Sigma^{-1} X_i - e^\intercal \Sigma_{\mathrm{gt}}^{-1} X_i\right| \le \left|e^\intercal \left(A^{-1} - \Sigma_{\mathrm{gt}}^{-1}\right) X_i\right| + \left|\frac{e^\intercal A^{-1} X_i X_i^\intercal A^{-1} X_i^\intercal}{1 + X_i^\intercal A^{-1} X_i}\right| = \widetilde{O}\left(\frac{d + \sqrt{nd}}{n^2}\right) = o\left(\frac{1}{n}\right).$$

$\square$

### E.6.7 BOUNDED RESIDUALS

For the next step of our analysis we will show that the residuals are bounded with very high probability. Recall that under the assumptions of Claim E.3, we assume that the labels are drawn from the distribution

$$Y = X\beta_{\mathrm{gt}} + \zeta \sim X\beta_{\mathrm{gt}} + \mathcal{N}\left(\vec{0}, I_n\right)$$

Therefore, we clearly have that with very high probability $\max_{i \in [n]} |\zeta_i| \leq \log(n) = \widetilde{O}(1)$ as required. The issue is that the residuals of the regression are not necessarily equal to $\zeta$.

Recall that the residuals are equal to

$$R \overset{\text{def}}{=} Y - X\Sigma^{-1}X^{\mathsf{T}}Y = \zeta - X\Sigma^{-1}X^{\mathsf{T}}\zeta$$

**Claim E.11.** *With very high probability*

$$\forall i \quad |R_i - \zeta_i| = o(1)$$

*Proof of Claim E.11.* Fix some index $i \in [n]$.

$$R_i = \zeta_i - X_i^{\mathsf{T}}\Sigma^{-1}X^{\mathsf{T}}\zeta = \zeta_i - \sum_j X_i^{\mathsf{T}}\Sigma^{-1}X_j\zeta_j = \zeta_i - \zeta_i X_i^{\mathsf{T}}\Sigma^{-1}X_i - \sum_{j \neq i} \zeta_j X_i^{\mathsf{T}}\Sigma^{-1}X_j \quad (23)$$

Therefore, from Claim E.8, we have

$$|R_i - \zeta_i| = \left| \sum_{j \neq i} \zeta_j X_i^{\mathsf{T}}\Sigma^{-1}X_j \right| + \widetilde{O}\left(\frac{d}{n}\right) = \left| \sum_{j \neq i} \zeta_j X_i^{\mathsf{T}}\Sigma^{-1}X_j \right| + o(1)$$

This leaves us with the task of analyzing the term

$$X_i^{\mathsf{T}}\Sigma^{-1}\left(\sum_{j \neq i} \zeta_j X_j\right) = \sum_{j \neq i} Z_j$$

where $Z_j = X_i^{\mathsf{T}}\Sigma^{-1}X_j\zeta_j$. To bound this term, we view the process of generating the samples as first generating the covariates $X_i$, and then after fixing some values for the $X_i$, it generates the errors $\zeta_j$.

In other words, we will show that with very high probability over the $X_i$

$$\Pr_{\zeta_j \sim \mathcal{N}(0,1)}\left[\left| \sum_{j \neq i} Z_j \right| > o(1)\right] < \frac{1}{\text{super-poly}(n)} \; .$$

In particular, with very high probability over the covariates, we have

$$\forall j \neq i \quad \left(X_i^{\mathsf{T}}\Sigma^{-1}X_j\right)^2 = \widetilde{O}\left(\frac{d}{n^2}\right) \implies \sum_{j \in [n] \setminus \{i\}} \left(X_i^{\mathsf{T}}\Sigma^{-1}X_j\right)^2 = \widetilde{O}\left(\frac{d}{n}\right) \; .$$

Therefore, fixing the covariates $X_i$, and viewing $\sum_{j \neq i} Z_j$ as a random variable dependent on the randomness of the errors $\zeta_j$, we have

$$\sum_{j \neq i} Z_j \sim \mathcal{N}\left(0, \sum_{j \in [n] \setminus \{i\}} \left(X_i^{\mathsf{T}}\Sigma^{-1}X_j\right)^2\right) = \mathcal{N}\left(0, \widetilde{O}\left(\frac{d}{n}\right)\right) \; ,$$

yielding the claim.

$\square$

### E.6.8 LARGE INFLUENCE SCORES

For the final step of our proof of Claim E.3, we will show that with very high probability, there are many samples in the regression that have relatively high AMIP influence scores.

**Claim E.12.** *Let $\alpha_i = e^\intercal \Sigma^{-1} X_i R_i$ denote the AMIP influence score of the $i$th sample. Then with very high probability, there are is a set $T_0 \subseteq [n]$ of least $k_0 = \widetilde{\Omega}(n)$ "influential samples" – i.e., such that $\forall i \in T$ $\alpha_i \geq \frac{1}{10n}$.*

Proving Claim E.12 will also conclude our proof of Claim E.3, as this will show that all conditions required for a regression to be well-behaved are fulfilled with very high probability.

*Proof of Claim E.12.* To prove Claim E.12 we first show that a very large number of samples must have a relatively high inner product with the axis of interest. In other words, we will show that with very high probability

$$\left| \left\{ i \middle| \left| e^\intercal \Sigma^{-1} X_i \right| \geq \frac{1}{2n} \right\} \right| = \widetilde{\Omega}(n) \tag{24}$$

To prove equation (24), note that:

- $e^\intercal \Sigma^{-1} e = \frac{1 \pm o(1)}{n}$ (this follows immediately from Claim E.7).

- $\sum_{i \in [n]} \left| e^\intercal \Sigma^{-1} X_i \right|^2 = \sum_{i \in [n]} e^\intercal \Sigma^{-1} X_i X_i^\intercal \Sigma^{-1} e = e^\intercal \Sigma^{-1} e$.

- In Claim E.10, we showed that wvhp $\forall i$ $\left( e^\intercal \Sigma^{-1} X_i \right)^2 \leq \widetilde{O}\left( \frac{1}{n} \right) \times e^\intercal \Sigma^{-1} e$.

Therefore, we must have at least $\widetilde{\Omega}(n)$ samples with $\left| e^\intercal \Sigma^{-1} X_i \right| \geq \frac{1}{2n}$. Let $i$ be the index of such a sample. If sign $\left( e^\intercal \Sigma^{-1} X_i \right) \times R_i \geq \frac{1}{5}$, then we will also have $\alpha_i \geq \frac{1}{10n}$.

In the proof of Claim E.11, we showed that wvhp $|R_i - \zeta_i| = o(1)$ for all $i$ (where $\zeta_i = Y_i - X_i^\intercal \beta_{\text{gt}}$ are the "ground truth residuals", and are drawn iid from a normal distribution). In particular, wvhp $\forall i$ $|R_i - \zeta_i| < \frac{1}{4} - \frac{1}{5}$, so as long as sign $\left( e^\intercal \Sigma^{-1} X_i \right) \zeta_i \geq \frac{1}{4}$, we have $\alpha_i \geq \frac{1}{10n}$.

But $\zeta_i$ is drawn iid from a normal distribution, so sign $\left( e^\intercal \Sigma^{-1} X_i \right) \zeta_i \geq \frac{1}{4}$ has constant probability and is independent of $X_i$. Therefore, applying the Hoeffding-Chernoff bound, we can easily see that wvhp at least a constant fraction of the $\widetilde{\Omega}(n)$ samples for which $\left| e^\intercal \Sigma^{-1} X_i \right| \geq \frac{1}{2n}$ also have $\alpha_i \geq \frac{1}{10n}$, thus concluding our proof of Claims E.12 and E.3.

$\square$

## F TIGHTNESS OF OHARE

In the previous section, we proved Theorem E.1 which says that for "well-behaved" data, the ACRE algorithm outputs nearly tight bounds on the removal effects for a range of removal set sizes $k$. In this section, we will extend those results to the one-hot aware version of the algorithm – OHARE .

**Theorem F.1** (OHARE Bounds are Tight on Well-behaved Data)**.** *Consider a linear regression from a set of continuous features $X \in \mathbb{R}^{n \times d}$ and a set of $m$ dummy variables, representing a categorical feature $B_1 \sqcup \cdots \sqcup B_m = [n]$, to a target variable $Y$.*

*For any fixed $\varepsilon > 0$, there exists $\nu \in \text{polylog}(n)$ such that:*

*If $n_j = |B_j|$ denote the number of samples that take the value $j$ in the categorical feature, and for all $j \in [m]$, we have*

$$n^\varepsilon + \nu \sqrt{d} < n_j < 0.49n \,,$$

*that the dimension of the continuous features $d$ is at most $d \leq n^{4/5}/\nu$, and that the continuous features are then drawn iid from a well-behaved distribution $X_i \sim \mathcal{X}$ independently of their value on the categorical feature.*

*And if the outcomes $Y$ are drawn iid from a normal distribution around a linear model of the features*

$$Y_i \sim \underbrace{\mu_{j(i)}}_{\text{categorical contribution}} + \underbrace{\langle X_i, \beta_{\text{gt}} \rangle}_{\text{continuos contribution}} + \underbrace{\mathcal{N}(0,1)}_{\text{error}},$$

*for some unknown ground truth linear model $(\mu, \beta_{\text{gt}}) \in \mathbb{R}^{m+d}$.*

*Then, for any axis $e \in \mathbb{S}^{d-1}$, with very high probability, the upper and lower bounds produced by OHARE on this regression are close to tight*

$$\frac{U_k}{L_k} = 1 + O\left(\frac{\text{polyloglog}(n)}{\sqrt{\log(n)}}\right)$$

*for all $k < k_{\text{threshold}}$, where*

$$k_{\text{threshold}} = \widetilde{\Theta}\left(\min\left\{\frac{n}{\sqrt{d}}, \frac{n^2}{d^2}, n^{1-\varepsilon}\right\}\right).$$

Our goal for the rest of this section will be to prove Theorem F.1.

## F.1 Main Challenges and Proof Structure

Recall from Section F that the key idea of the OHARE algorithm is to analyse a process that is equivalent to the regression with a one-hot encoding. In this alternative formulation of one-hot controlled regression, we first split our samples into buckets $B_j \subseteq [n]$ corresponding to each of the potential values of the categorical feature, reaverage the samples in each bucket

$$\widetilde{X}_i = X_i - \mathbb{E}_{i' \in B_{j(i)}}[X_{i'}] \in \mathbb{R}^d$$

$$\widetilde{Y}_i = Y_i - \mathbb{E}_{i' \in B_{j(i)}}[Y_{i'}] \in \mathbb{R}$$

and perform a regression with just the reaveraged continuous features. The OHARE algorithm then computes the same MSN bounds as the ACRE algorithm would, but on these reaveraged continous features and combines them with terms corresponding to the effect a removal might have on the reaveraging process.

Our proof of Theorem F.1 will follow a similar path. We will first prove a claim very similar to Claim E.3 adapted to the OHARE case:

**Claim F.2** (Well-behaved distributions yield well-behaved regressions with high probability after reaveraging). *Let $n, d, m, X, Y$ be as in Theorem F.1.*

*Then, for any axis $e \in \mathbb{S}^{d-1}$, with very high probability, the regression $\widetilde{X}, \widetilde{Y}$ is ACRE-friendly for all $k \leq k_0$, for $k_0 = \widetilde{\Omega}(n^{1-\varepsilon})$.*

Note that Claim F.2 does not follow immediately from the corresponding Claim E.3 for ACRE, since the continuous features $\widetilde{X}$ are no longer drawn iid from a well-behaved distribution as the reaveraging step could have changed them and similarly the reaveraged labels $\widetilde{Y}$ are not drawn iid from a normal distribution around a linear combination of the continous features. The proof of Claim F.2 will follow a very similar path to the proof of Claim E.3, but will also have to account for these additional corrections.

Finally, we will prove that the additional corrections taken into account by OHARE will not change the upper and lower bounds too much, yielding Theorem F.1.

## F.2 Proof of Claim F.2

We begin by adapting the analysis from the continuous features (Claim E.3) to the reaveraged samples. Throughout this section, let $X \in \mathbb{R}^{n \times d}$ denote just the continuous covariates and $\widetilde{X} \in \mathbb{R}^{n \times d}$ denote the reaveraged continuous covariates.

As in the proof of Claim E.3, we normalize our samples so that the ground truth covariance of the continuous features is equal to the identity, and denote by $\Sigma = X^\intercal X \in \mathbb{R}^{d \times d}$ the unnormalized empirical covariance of $X$ and by $\widetilde{\Sigma} = \widetilde{X}^\intercal \widetilde{X} \in \mathbb{R}^{d \times d}$ denote the unnormalized empirical covariance of the reaveraged samples covariates.

### F.2.1 REAVERAGING

The first step of our analysis will be to show that with very high probability the reaveraging step makes only a small change to the covariates as well as the target variable. Let $j \in [m]$ be the index of any bucket of samples and let

$$\xi_j = \frac{1}{n_j} \sum_{i \in B_j} X_i = \mathbb{E}_{i \in B_j} [X_i]$$

$$y_j = \frac{1}{n_j} \sum_{i \in B_j} Y_i = \mathbb{E}_{i \in B_j} [Y_i]$$

denote the averaging effect for this bucket.

**Claim F.3.** *The mean and covariance of $\xi_j$ are*

$$\mathbb{E}[\xi_j] = 0$$

$$\mathbb{E}[\xi_j \xi_j^\intercal] = \frac{1}{n_j} I$$

*Moreover, with very high probability*

$$\|\xi_j\| \leq \widetilde{O}\left(\sqrt{d/n_j}\right) = o(\sqrt{d})$$

$$|y_j - \mu_j - \langle \xi_j, \beta_{\mathrm{gt}} \rangle| \leq \widetilde{O}\left(1/\sqrt{n_j}\right) = o(1)$$

*Finally, the fourth moments of $\xi_j$ are also bounded*

$$\left\| \mathbb{E}\left[ \|\xi_j\|^2 \xi_j \xi_j^\intercal \right] \right\| = O\left(\frac{d}{n_j^2}\right)$$

*Proof of Claim F.3.* First, recall that our covariate distribution $\mathcal{X}$ was normalized to have mean $0$ and covariance identity, yielding the first part of Claim F.3 immediately from the fact that $\xi_j$ is the empirical average of $n_j$ iid samples drawn from $\mathcal{X}$.

Recall that we assumed the target variable was drawn from a normal distribution

$$Y_i \sim \mu_j + \langle X_i, \beta_{\mathrm{gt}} \rangle + \mathcal{N}(0, 1)$$

In particular,

$$\mathbb{E}_{i \in B_j} [Y_i] - \mu_j - \mathbb{E}_{i \in B_j} [\langle X_i, \beta_{\mathrm{gt}} \rangle]$$

is the empirical average over $n_j$ samples of this normal distribution, yielding the claim that with very high probability

$$|y_j - \mu_j - \langle \xi_j, \beta_{\mathrm{gt}} \rangle| \leq \widetilde{O}\left(1/\sqrt{n_j}\right) = o(1) \, .$$

The rest of Claim F.3 will follow from a combination of Claim E.7 (which bounds the norms and higher moments of well-behaved distributions), and Lemma E.6 (which states that the sum over iid samples from a well-behaved distribution is also well-behaved).

Indeed $\xi_j$ is the empirical average over $n_j = \mathrm{poly}(n)$ samples from a well-behaved distribution $\mathcal{X}$, so Lemma E.6 ensures that for all $j$, the variable $\sqrt{n_j} \xi_j$ is well behaved, and, as noted above, it has covariance identity. Therefore, it follows from Claim E.7 that

$$\left\| \mathbb{E}\left[ \|\xi_j\|^2 \xi_j \xi_j^\intercal \right] \right\| \leq O\left(\frac{d}{n_j^2}\right)$$

and that with very high probability

$$\|\xi_j\| \le \widetilde{O}\left(\sqrt{d/n_j}\right)$$

$\square$

### F.2.2 Renormalization

Let $\Sigma = X^\intercal X = \sum_{i \in [n]} X_i X_i^\intercal$ be the unnormalized empirical covariance of the continuous features, and let $\Sigma_{\text{gt}} = nI = \mathbb{E}[\Sigma]$ be the ground truth mean of these features. We continue along the same lines as the proof of Claim E.3 by adapting Claim E.7 to the reaveraged setting:

**Claim F.4.** *With very high probability, $\left\|\widetilde{\Sigma} - \Sigma\right\| = \widetilde{O}(d + m)$.*

*In particular, due to Claim E.7 and the triangle inequality*

$$\left\|\widetilde{\Sigma} - \Sigma_{\text{gt}}\right\| \le \left\|\widetilde{\Sigma} - \Sigma\right\| + \|\Sigma - \Sigma_{\text{gt}}\| = \widetilde{O}\left(\sqrt{nd} + d + m\right) = \widetilde{O}\left(\sqrt{nd} + m\right) = o(n)$$

*Proof of Claim F.4.* We have

$$\widetilde{\Sigma} = \sum_{i \in [n]} \widetilde{X}_i \widetilde{X}_i^\intercal = \sum_{i \in [n]} \left(X_i - \xi_{j(i)}\right)\left(X_i - \xi_{j(i)}\right)^\intercal = \sum_{i \in [n]} X_i X_i^\intercal - \sum_{i \in [n]} \xi_{j(i)} \xi_{j(i)}^\intercal = \Sigma - \sum_{j \in [m]} n_j \xi_j \xi_j^\intercal$$

Therefore, it only remains to bound

$$\left\|\sum_{i \in [n]} \xi_{j(i)} \xi_{j(i)}^\intercal\right\| = \left\|\sum_{j \in [m]} n_j \xi_j \xi_j^\intercal\right\|$$

We do this using our approximate matrix Bernstein inequality – Lemma E.5. Let $Z_j = n_j \xi_{j(i)} \xi_{j(i)}^\intercal$.

In Claim F.3, we showed that $\mathbb{E}[Z_j] = I$, that $\mathbb{E}\left[Z_j^2\right] = \mathbb{E}\left[n_j^2 \|\xi_j\|^2 \xi_j \xi_j^\intercal\right]$ has bounded norm

$$\left\|\mathbb{E}\left[n_j^2 \|\xi_j\|^2 \xi_j \xi_j^\intercal\right]\right\| = O(d) ,$$

and that with very high probability

$$\|Z_j\| = n_j \|\xi_j\|^2 = \widetilde{O}(d) .$$

Therefore, from Lemma E.5, with very high probability

$$\left\|\sum_{j \in [m]} Z_j\right\| \le \left\|\sum_{j \in [m]} Z_j - \sum_{j \in [m]} \mathbb{E}[Z_j]\right\| + m = \widetilde{O}\left(d + \sqrt{md} + m\right) = \widetilde{O}(d + m)$$

Finally, recall that we assumed to have at least $n^\varepsilon$ samples in the smallest category. Therefore, $m \le n^{1-c} \ll n$, completing the proof.

$\square$

### F.2.3 Maximal Norm

We proceed to prove that each of the conditions required for the regression to be well-behaved occurs with very high probability. The first (and easiest to prove) is the condition that $\max_{i \in [n]} X_i^\intercal \Sigma^{-1} X_i$ is bounded.

**Claim F.5.** *With very high probability*

$$\max_{i \in [n]} \left\{\widetilde{X}_i^\intercal \widetilde{\Sigma}^{-1} \widetilde{X}_i\right\} = \widetilde{O}\left(\frac{d}{n}\right)$$

*Proof of Claim F.5.* Let $\lambda$ denote the spectrum of a matrix. In the proof of Claim F.4, we already showed that with very high probability $\left\|\widetilde{\Sigma} - \Sigma_{\text{gt}}\right\| = o(n) = o\left(\min \lambda\left(\Sigma_{\text{gt}}\right)\right)$ (where $\Sigma_{\text{gt}} = nI$). When this holds, we also have

$$\lambda\left(\widetilde{\Sigma}^{-1}\right) \subseteq (1 \pm o(1)) \times \frac{1}{n}.$$

Moreover, in Claim E.7 we show that with very high probability $\max_{i \in [n]} \|X_i\|^2 = \widetilde{O}(d)$. Combined with Claim F.3 and the triangle inequality, we can derive a bound on $\left\|\widetilde{X}_i\right\| < \|X_i\| + \left\|\xi_{j(i)}\right\| = \widetilde{O}\left(\sqrt{d}\right)$ with very high probability.

Therefore, with very high probability

$$\max_{i \in [n]} \left\{\widetilde{X}_i^\intercal \widetilde{\Sigma}^{-1} \widetilde{X}_i\right\} \leq \max\left\{\lambda\left(\widetilde{\Sigma}^{-1}\right)\right\} \times \max_{i \in [n]} \left\{\left\|\widetilde{X}_i\right\|^2\right\} = \widetilde{O}\left(\frac{d}{n}\right)$$

$\square$

### F.2.4 BOUNDED INNER PRODUCTS

For the next step of our proof, we show that the second condition of ACRE-friendliness of the regression holds with very high probability. In particular, we will show that

**Claim F.6.** *For $M \in \left\{\Sigma^{-1}, \widetilde{\Sigma}^{-1}\right\}$, the following inequalities hold with very high probability*

•

$$\max_{j_1 \neq j_2 \in [m]} \left\{\left|\xi_{j_1}^\intercal M \xi_{j_2}\right|\right\} = \widetilde{O}\left(\frac{\sqrt{d}}{n\sqrt{n_{j_1} n_{j_2}}} \times \left(1 + \frac{d\sqrt{n_{j_1}}}{n}\right)\right)$$

•

$$\forall i \in [n], j \in [m] \quad |X_i^\intercal M \xi_j| = \widetilde{O}\left(\frac{\sqrt{d}}{n\sqrt{n_j}} + \frac{d}{nn_j} 1_{i \in B_j}\right)$$

•

$$\max_{i_1 \neq i_2 \in [n]} \left\{\left|\widetilde{X}_{i_1}^\intercal M^{-1} \widetilde{X}_{i_2}\right|\right\} = \widetilde{O}\left(\frac{\sqrt{d}}{n}\right)$$

Claim E.9 proves the third inequality of Claim F.6 for the case where $M = \Sigma^{-1}$. The following Lemma F.7 will prove the first inequality for this same case.

**Lemma F.7.** *Let $j_1 \neq j_2 \in [m]$ be the indices of two distinct buckets, and let $\Sigma = X^\intercal X = \sum_{i \in [n]} X_i X_i^\intercal$ denote the unnormalized empirical covariance of the unaveraged samples.*

*With very high probability*

$$\left|\xi_{j_1}^\intercal \Sigma^{-1} \xi_{j_2}\right| = \widetilde{O}\left(\frac{\sqrt{d}}{\sqrt{n_{j_1} n_{j_2}} \times n} \times \left(1 + \frac{d\sqrt{n_{j_1}}}{n}\right)\right).$$

The proof of Lemma F.7 is very long and technical, and we devote Section F.3 to it. For now, let us continue with our proof of the rest of the inequalities in Claim F.6 assuming Lemma F.7.

*Proof of Claim F.6.* We begin by bounding the inner product between the bucket averages through the covariance $\widetilde{\Sigma}^{-1}$. Define

$$\eta \stackrel{\text{def}}{=} \max_{j_1, j_2} \left\{\frac{\left|\xi_{j_1}^\intercal \widetilde{\Sigma}^{-1} \xi_{j_2}\right|}{\frac{\sqrt{d}}{\sqrt{n_{j_1} n_{j_2}} \times n} \times \left(1 + \frac{d\sqrt{n_{j_1}}}{n}\right)}\right\}.$$

Our first goal will be to show that with very high probability $\eta = \widetilde{O}(1)$.

Consider the following identity (where $A$ and $B$ are $d \times d$ matrices such that $A$ and $A - B$ are invertible):

$$A = (A - B) + B \,.$$

Multiplying this equation by $(A - B)^{-1}$ from the left and $A^{-1}$ yields the following identity that we will use in our analysis.

$$(A - B)^{-1} = A^{-1} + (A - B)^{-1} B A^{-1} \tag{25}$$

We apply equation (25) for $A = \Sigma, B = \Sigma - \widetilde{\Sigma} = \sum_{j \in [m]} n_j \xi_j \xi_j^\mathsf{T}$, giving us the identity

$$\widetilde{\Sigma}^{-1} = \Sigma^{-1} + \sum_{j \in [m]} \widetilde{\Sigma}^{-1} \xi_j n_j \xi_j^\mathsf{T} \Sigma^{-1} \,.$$

Therefore, from the triangle inequality,

$$\eta \times \frac{\sqrt{d}}{\sqrt{n_{j_1} n_{j_2}} n} \times \left(1 + \frac{d\sqrt{n_{j_1}}}{n}\right) = \left|\xi_{j_1}^\mathsf{T} \widetilde{\Sigma}^{-1} \xi_{j_2}\right| \leq \left|\xi_{j_1}^\mathsf{T} \Sigma^{-1} \xi_{j_2}\right| + \sum_{j \in [m]} \left|\xi_{j_1}^\mathsf{T} \widetilde{\Sigma}^{-1} \xi_j n_j \xi_j^\mathsf{T} \Sigma^{-1} \xi_{j_2}\right| \,.$$

For all $j \neq j_1, j_2$, from Lemma F.7, with very high probability

$$\left|\xi_{j_1}^\mathsf{T} \widetilde{\Sigma}^{-1} \xi_j n_j \xi_j^\mathsf{T} \Sigma^{-1} \xi_{j_2}\right| = \widetilde{O}\left(\eta \frac{d}{n^2 \sqrt{n_{j_1} n_{j_2}}} \left(1 + \frac{d^2 n_j}{n^2}\right)\right) \,.$$

Therefore, with very high probability

$$\sum_{j \neq j_1, j_2} \left|\xi_{j_1}^\mathsf{T} \widetilde{\Sigma}^{-1} \xi_j n_j \xi_j^\mathsf{T} \Sigma^{-1} \xi_{j_2}\right| = \widetilde{O}\left(\frac{md}{n^2 \sqrt{n_{j_1} n_{j_2}}} + \frac{d^3}{n^3 \sqrt{n_{j_1} n_{j_2}}}\right) \times \eta = \widetilde{O}\left(\frac{\sqrt{d}}{\sqrt{n_{j_1} n_{j_2}} n} \eta\right) \,,$$

where the last step utilized our assumptions that $m \leq \frac{n}{n_{\min}} \ll \frac{n}{\sqrt{d}}$ and that $n^4 \gg d^5$.

When $j = j_1$, we have

$$\left|\xi_{j_1}^\mathsf{T} \widetilde{\Sigma}^{-1} \xi_j n_j \xi_j^\mathsf{T} \Sigma^{-1} \xi_{j_2}\right| = \left|\xi_{j_1}^\mathsf{T} \widetilde{\Sigma}^{-1} \xi_{j_1} n_{j_1} \xi_{j_1}^\mathsf{T} \Sigma^{-1} \xi_{j_2}\right| = \widetilde{O}\left(\frac{d}{n \times n_{j_1}} \times n_{j_1} \times \frac{\sqrt{d}}{\sqrt{n_{j_1} n_{j_2}}} \left(1 + \frac{d\sqrt{n_{j_1}}}{n}\right)\right) =$$

$$= \widetilde{O}\left(\frac{\sqrt{d}}{\sqrt{n_{j_1} n_{j_2}}} \left(1 + \frac{d\sqrt{n_{j_1}}}{n}\right)\right) \,.$$

Similarly, for $j = j_2$, we have

$$\left|\xi_{j_1}^\mathsf{T} \widetilde{\Sigma}^{-1} \xi_j n_j \xi_j^\mathsf{T} \Sigma^{-1} \xi_{j_2}\right| = \left|\xi_{j_1}^\mathsf{T} \widetilde{\Sigma}^{-1} \xi_{j_2} n_{j_2} \xi_{j_2}^\mathsf{T} \Sigma^{-1} \xi_{j_2}\right| = \widetilde{O}\left(\frac{d}{n \times n_{j_2}} \times n_{j_2} \times \frac{\sqrt{d}\eta}{\sqrt{n_{j_1} n_{j_2}}} \left(1 + \frac{d\sqrt{n_{j_1}}}{n}\right)\right) =$$

$$= o\left(\frac{\sqrt{d}\eta}{\sqrt{n_{j_1} n_{j_2}}} \left(1 + \frac{d\sqrt{n_{j_1}}}{n}\right)\right) \,.$$

Therefore, with very high probability

$$\eta = \frac{\left|\xi_{j_1}^\mathsf{T} \widetilde{\Sigma}^{-1} \xi_{j_2}\right|}{\frac{\sqrt{d}}{\sqrt{n_{j_1} n_{j_2}} n} \times \left(1 + \frac{d\sqrt{n_{j_1}}}{n}\right)} \leq \frac{\left|\xi_{j_1}^\mathsf{T} \Sigma^{-1} \xi_{j_2}\right| + \sum_{j \in [m]} \left|\xi_{j_1}^\mathsf{T} \widetilde{\Sigma}^{-1} \xi_j n_j \xi_j^\mathsf{T} \Sigma^{-1} \xi_{j_2}\right|}{\frac{\sqrt{d}}{\sqrt{n_{j_1} n_{j_2}} n} \times \left(1 + \frac{d\sqrt{n_{j_1}}}{n}\right)} = \widetilde{O}(1) + o(\eta) \,.$$

Therefore, with very high probability $\eta = \widetilde{O}(1)$, proving the first portion of our claim.

Next, consider a term of the form $X_i^\mathsf{T} \widetilde{\Sigma}^{-1} \xi_j$. To bound this term, we open $\widetilde{\Sigma}^{-1}$ again using equation (25). Indeed, we have

$$\left|X_i^\mathsf{T} \widetilde{\Sigma}^{-1} \xi_j\right| \leq \left|X_i^\mathsf{T} \Sigma^{-1} \xi_j\right| + \sum_{j'} \left|X_i^\mathsf{T} \Sigma^{-1} \xi_{j'} n_{j'} \xi_{j'}^\mathsf{T} \widetilde{\Sigma}^{-1} \xi_j\right| \,.$$

Define

$$\zeta_{i,j} = \xi_j - \frac{1}{n_j} X_i 1_{i \in B_j} = \begin{cases} \xi_j - \frac{1}{n_j} X_i & i \in B_j \\ \xi_j & i \notin B_j \end{cases}$$

$\zeta_{i,j}$ over $B_j$ bucket had the $X_i$ sample been replaced with the 0 vector. Note that with very high probability

$$\left| X_i^\intercal \Sigma^{-1} (\xi_j - \zeta_{i,j}) \right| \leq \frac{1_{i \in B_j}}{n_j} \|X_i\|^2 \|\Sigma^{-1}\| = \widetilde{O}\left( \frac{d 1_{i \in B_j}}{n n_j} \right).$$

To proceed, we apply the Sherman-Morrison identity with $v = X_i$ and $A = \Sigma_{[n] \setminus \{i\}} = \Sigma - v v^\intercal$, to show that

$$\Sigma^{-1} = A^{-1} - \frac{A^{-1} v v^\intercal A^{-1}}{1 - v^\intercal A^{-1} v}.$$

Moreover, because both $A$ and $\zeta_{i,j}$ are independent of the $i$th sample and $\mathcal{X}$ is well-behaved, with very high probability

$$\left| X_i^\intercal A^{-1} \zeta_{i,j} \right| = \widetilde{O}\left( \left\| A^{-1} \zeta_{i,j} \right\| \right) = \widetilde{O}\left( \frac{\sqrt{d}}{\sqrt{n_j} n} \right)$$

Combining this with the Sherman-Morrison formula, we have

$$\left| X_i^\intercal \Sigma^{-1} \zeta_{i,j} \right| = \left( 1 - \frac{X_i^\intercal A^{-1} X_i}{1 - X_i^\intercal A^{-1} X_i} \right) \left| X_i^\intercal A^{-1} \zeta_{i,j} \right| = \widetilde{O}\left( \frac{\sqrt{d}}{\sqrt{n_j} n} \right).$$

Therefore, for all $j \in [m]$, with very high probability

$$\left| X_i^\intercal \Sigma^{-1} \xi_j \right| = \widetilde{O}\left( \frac{\sqrt{d}}{\sqrt{n_j} n} + \frac{1_{i \in B_j} d}{n_j n} \right).$$

Therefore, with very high probability

$$\sum_{j' \in [m]} \left| X_i^\intercal \Sigma^{-1} \xi_{j'} n_{j'} \xi_{j'}^\intercal \widetilde{\Sigma}^{-1} \xi_j \right| \leq$$

$$\leq \sum_{j' \in [m]} \widetilde{O}\left( \left( \frac{\sqrt{d}}{n \sqrt{n_{j'}}} + \frac{d 1_{i \in B_{j'}}}{n n_{j'}} \right) \times n_{j'} \times \left( \frac{\sqrt{d}}{\sqrt{n_j n_{j'}} n} \left( 1 + \frac{d \sqrt{n_{j'}}}{n} \right) + \frac{1_{j = j'} d}{\sqrt{n_j n_{j'}} n} \right) \right) =$$

$$= \sum_{j' \in [m]} \widetilde{O}\left( \frac{d}{n^2 \sqrt{n_j}} + \frac{d^2 \sqrt{n_{j'}}}{n^3 \sqrt{n_j}} + \frac{d^{3/2} 1_{j=j'}}{\sqrt{n_j} n^2} + \frac{d^{3/2} 1_{i \in B_{j'}}}{\sqrt{n_j n_{j'}} n^2} \times \left( 1 + \frac{d \sqrt{n_{j'}}}{n} \right) + \frac{d^2 1_{i \in B_j} 1_{j=j'}}{n_j n^2} \right) =$$

$$= \widetilde{O}\left( \frac{\sqrt{d}}{\sqrt{n_j} n} + \frac{1_{i \in B_j} d}{n_j n} \right) + \widetilde{O}\left( \frac{md}{n^2 \sqrt{n_j}} + \frac{\sqrt{mn} d^2}{n^3 \sqrt{n_j}} + \frac{d^{3/2}}{\sqrt{n_{j'}} n^2} + \frac{d 1_{i \in B_j}}{n n_j} \right) = \widetilde{O}\left( \frac{\sqrt{d}}{\sqrt{n_j} n} + \frac{1_{i \in B_j} d}{n n_j} \right),$$

where the last step utilizes our assumptions that $m = \widetilde{O}\left( \frac{n}{\sqrt{d}} \right)$ and that $d^5 = \widetilde{O}\left( n^4 \right)$.

Finally, let $i_1 \neq i_2$ be the indices of two samples. From Claim E.9, we have that with very high probability $\left| X_{i_1}^\intercal \Sigma^{-1} X_{i_2} \right| = \widetilde{O}\left( \frac{\sqrt{d}}{n} \right)$. Opening $\widetilde{\Sigma}^{-1}$ again using equation (25), we see that with very high probability

$$\left| X_{i_1}^\intercal \widetilde{\Sigma}^{-1} X_{i_2} \right| \leq \left| X_{i_1}^\intercal \Sigma^{-1} X_{i_2} \right| + \sum_{j \in [m]} \left| X_{i_1}^\intercal \Sigma^{-1} \xi_j n_j \xi_j^\intercal \widetilde{\Sigma}^{-1} X_{i_2} \right| =$$

$$= \widetilde{O}\left( \frac{\sqrt{d}}{n} \right) + \sum_{j \in [m]} \widetilde{O}\left( \left( \frac{\sqrt{d}}{\sqrt{n_j} n} + \frac{1_{i_1 \in B_j} d}{n_j n} \right) \times \left( \frac{\sqrt{d}}{\sqrt{n_j} n} + \frac{1_{i_2 \in B_j} d}{n_j n} \right) \times n_j \right) =$$

$$= \widetilde{O}\left( \frac{\sqrt{d}}{n} \right) + \widetilde{O}\left( \frac{md}{n^2} + \frac{d^2}{n^2 n_{\min}} \right) = \widetilde{O}\left( \frac{\sqrt{d}}{n} \right).$$

where $n_{\min} \stackrel{\text{def}}{=} \min_{j \in [m]} \{n_j\} = \widetilde{\Omega}\left(\sqrt{d}\right)$.

$\square$

### F.2.5 PROJECTION ON THE $e$ AXIS

For the next property of well-behaved regressions, we will want to show that with very high probability $e^{\mathsf{T}}\Sigma^{-1}X_i$ is bounded for all $i$. Note that from the exponential decay assumption due to $\mathcal{X}$ being well-behaved would suffice to give a good bound on $e^{\mathsf{T}}\Sigma_{\text{gt}}^{-1}X_i$ (where $\Sigma_{\text{gt}} = nI$), but as before, the challenge will be to show that $\Sigma^{-1}$ doesn't rotate $X_i$ onto $e$.

**Claim F.8.** *With very high probability*

$$\forall i \in [n] \quad \left\langle \widetilde{\Sigma}^{-1}\widetilde{X}_i, e \right\rangle = \frac{e^{\mathsf{T}}X_i}{n} \pm o\left(\frac{1}{n}\right) .$$

*In particular, with very high probability*

$$\max_{i \in [n]} \left| \left\langle \widetilde{\Sigma}^{-1}\widetilde{X}_i, e \right\rangle \right| = \widetilde{O}\left(\frac{1}{n}\right) = \sqrt{\widetilde{O}\left(\frac{1}{n}\right) \times e^{\mathsf{T}}\widetilde{\Sigma}^{-1}e}$$

Our proof of Claim F.8, will make use of the following Lemma F.9 that will also be proved in Section F.3.

**Lemma F.9.** *For any $j \in [m]$, with very high probability,*

$$\left| e^{\mathsf{T}}\Sigma^{-1}\xi_j \right| = \widetilde{O}\left(\frac{1}{n\sqrt{n_j}} \times \left(1 + \frac{d\sqrt{n_j}}{n}\right)\right) .$$

*Proof of Claim F.8.* As in the proof of Claim F.6, we will use the matrix identity in equation (25) to show that

$$\widetilde{\Sigma}^{-1} = \Sigma^{-1} + \Sigma^{-1}C\widetilde{\Sigma}^{-1} ,$$

where $C = \Sigma - \widetilde{\Sigma} = \sum_{j \in [m]} n_j \xi_j \xi_j^{\mathsf{T}}$.

Define

$$\eta \stackrel{\text{def}}{=} \max_{j \in [m]} \left\{ \left| e^{\mathsf{T}}\widetilde{\Sigma}^{-1}\xi_j \right| \right\}$$

Using Lemma F.9, we see that with very high probability

$$\eta = \left| e^{\mathsf{T}}\widetilde{\Sigma}^{-1}\xi_j \right| \leq \left| e^{\mathsf{T}}\Sigma^{-1}\xi_j \right| + \sum_{j'} \left| e^{\mathsf{T}}\Sigma^{-1}\xi_{j'}n_{j'}\xi_{j'}^{\mathsf{T}}\widetilde{\Sigma}^{-1}\xi_j \right| = \widetilde{O}\left(\frac{1}{n} \times \left(\frac{m\sqrt{d} + d}{n\sqrt{n_j}}\right)\right) = \widetilde{O}\left(\frac{1}{n\sqrt{n_j}}\right) .$$

Opening $\widetilde{\Sigma}^{-1}$ again, we have

$$\left| e^{\mathsf{T}}\widetilde{\Sigma}^{-1}X_i - e^{\mathsf{T}}\Sigma^{-1}X_i \right| = \sum_{j \in [m]} \left| e^{\mathsf{T}}\widetilde{\Sigma}^{-1}\xi_j n_j \xi_j^{\mathsf{T}}\Sigma^{-1}X_i \right|$$

From Claim E.10, we know that with very high probability

$$\left| e^{\mathsf{T}}\Sigma^{-1}X_i - \frac{e^{\mathsf{T}}X_i}{n} \right| = o\left(\frac{1}{n}\right) .$$

Therefore, from Claim F.6, with very high probability

$$\left| e^{\mathsf{T}}\widetilde{\Sigma}^{-1}X_i - e^{\mathsf{T}}\Sigma^{-1}X_i \right| \leq \sum_{j \in [m]} \left| e^{\mathsf{T}}\widetilde{\Sigma}^{-1}\xi_j n_j \xi_j^{\mathsf{T}}\Sigma^{-1}X_i \right| =$$

$$= \sum_{j \in [m]} \widetilde{O}\left(\frac{1}{n\sqrt{n_j}} \times n_j \times \left(\frac{\sqrt{d}}{n\sqrt{n_j}} + \frac{d1_{i \in B_j}}{nn_j}\right)\right) =$$

$$= \widetilde{O}\left(\frac{\sqrt{d}m}{n^2} + \frac{d}{n^2\sqrt{n_{\min}}}\right) = o\left(\frac{1}{n}\right) .$$

Altogether, we have that with very high probability

$$\left| e^\mathsf{T} \widetilde{\Sigma}^{-1} \widetilde{X}_i - \frac{e^\mathsf{T} X_i}{n} \right| \leq \left| e^\mathsf{T} \widetilde{\Sigma}^{-1} \xi_{j(i)} \right| + \left| e^\mathsf{T} \widetilde{\Sigma}^{-1} X_i - e^\mathsf{T} \Sigma^{-1} X_i \right| + \left| e^\mathsf{T} \Sigma^{-1} X_i - \frac{e^\mathsf{T} X_i}{n} \right| = o\left( \frac{1}{n} \right).$$

$\square$

### F.2.6 BOUNDED RESIDUALS

Recall that we assumed that the samples of our regression were drawn from a ground truth linear model plus an iid normally distributed error. Therefore, in order to bound the empirical residuals, it suffices to show that they are close to the ground truth residuals with very high probability.

In particular, in the setting of Theorem F.1, we assumed that

$$Y_i \sim \underbrace{\mu_{j(i)}}_{\text{categorical contribution}} + \underbrace{\langle X_i, \beta_{\mathrm{gt}} \rangle}_{\text{continuous contribution}} + \underbrace{\mathcal{N}(0,1)}_{\text{error}}$$

Define the ground truth residuals to be

$$R_{\mathrm{gt}} = \left( Y_i - \mu_{j(i)} - \langle X_i, \beta_{\mathrm{gt}} \rangle \right)_{i \in [n]},$$

and the empirical residuals to be

$$R = \widetilde{Y} - \widetilde{X}\beta = \widetilde{Y} - \widetilde{X}\widetilde{\Sigma}^{-1}\widetilde{X}^\mathsf{T}\widetilde{Y}.$$

**Claim F.10.** *With very high probability,*

$$\|R - R_{\mathrm{gt}}\|_\infty = o(1).$$

*Proof of Claim F.10.* Denote

$$\mu_j^* \stackrel{\text{def}}{=} \frac{1}{n_j} \sum_{i \in B_j} Y_i = \mu_j + \xi_j^\mathsf{T} \beta_{\mathrm{gt}} + \frac{1}{n_j} \sum_{i \in B_j} (R_{\mathrm{gt}})_i$$

From the definitions of the residuals and the ground truth residuals, we have

$$R_i - (R_{\mathrm{gt}})_i = \left( Y_i - \mu_{j(i)}^* \right) - \left( X_i - \xi_{j(i)} \right)^\mathsf{T} \beta - \left( Y_i - \mu_{j(i)} - X_i^\mathsf{T} \beta_{\mathrm{gt}} \right) =$$

$$= -\xi_{j(i)}^\mathsf{T} \beta_{\mathrm{gt}} - \frac{1}{n_{j(i)}} \sum_{i' \in B_{j(i)}} (R_{\mathrm{gt}})_{i'} - X_i^\mathsf{T} (\beta - \beta_{\mathrm{gt}}) + \xi_{j(i)}^\mathsf{T} \beta =$$

$$= -\frac{1}{n_{j(i)}} \sum_{i' \in B_{j(i)}} (R_{\mathrm{gt}})_{i'} - \widetilde{X}_i^\mathsf{T} (\beta - \beta_{\mathrm{gt}}).$$

Expanding on the difference between the results of the reaveraged OLS and the ground truth linear model, we have

$$\beta - \beta_{\mathrm{gt}} = \widetilde{\Sigma}^{-1} \widetilde{X}^\mathsf{T} \left( \widetilde{Y} - \widetilde{X}\beta_{\mathrm{gt}} \right) = \widetilde{\Sigma}^{-1} \widetilde{X}^\mathsf{T} \left( Y_i - \mu_{j(i)}^* - \left( Y_i - \mu_{j(i)} - (R_{\mathrm{gt}})_i \right) + \xi_{j(i)}^\mathsf{T} \beta_{\mathrm{gt}} \right)_{i \in [n]} =$$

$$= \widetilde{\Sigma}^{-1} \widetilde{X}^\mathsf{T} \left( (R_{\mathrm{gt}})_i - \xi_{j(i)}^\mathsf{T} \beta_{\mathrm{gt}} - \frac{1}{n_{j(i)}} \sum_{i' \in B_{j(i)}} (R_{\mathrm{gt}})_{i'} + \xi_{j(i)}^\mathsf{T} \beta_{\mathrm{gt}} \right)_{i \in [n]} =$$

$$= \widetilde{\Sigma}^{-1} \widetilde{X}^\mathsf{T} \left( (R_{\mathrm{gt}})_i - \frac{1}{n_{j(i)}} \sum_{i' \in B_{j(i)}} (R_{\mathrm{gt}})_{i'} \right)_{i \in [n]}.$$

Therefore,

$$R_i - (R_{\mathrm{gt}})_i = -\frac{1}{n_{j(i)}} \sum_{i' \in B_{j(i)}} (R_{\mathrm{gt}})_{i'} - \widetilde{X}_i^\mathsf{T} \widetilde{\Sigma}^{-1} \widetilde{X}^\mathsf{T} \left( (R_{\mathrm{gt}})_{i^*} - \frac{1}{n_{j(i^*)}} \sum_{i' \in B_{j(i^*)}} (R_{\mathrm{gt}})_{i'} \right)_{i^* \in [n]}.$$

Summing over the contributions of the second term, we have

$$\sum_{i^*\in[n]} \widetilde{X}_i^\intercal \widetilde{\Sigma}^{-1} \widetilde{X}_{i^*} \left( (R_{\text{gt}})_{i^*} - \frac{1}{n_{j(i^*)}} \sum_{i'\in B_{j(i^*)}} (R_{\text{gt}})_{i'} \right) = \sum_{i^*\in[n]} \widetilde{X}_i^\intercal \widetilde{\Sigma}^{-1} \left( \widetilde{X}_{i^*} - \xi_{j(i^*)} \right) (R_{\text{gt}})_{i^*} \sim$$

$$\sim \mathcal{N}\left( 0, \sum_{i^*\in[n]} \left( \widetilde{X}_i^\intercal \widetilde{\Sigma}^{-1} \left( \widetilde{X}_{i^*} - \xi_{j(i^*)} \right) \right)^2 \right) = \mathcal{N}\left( 0, \widetilde{O}\left( \frac{d}{n} + \frac{d}{n n_{\min}} + \frac{d^2}{n^2} \right) \right),$$

so with very high probability this contribution is bounded in absolute value by $o(1)$.

Similarly,

$$\frac{1}{n_{j(i)}} \sum_{i'\in B_{j(i)}} (R_{\text{gt}})_{i'} \sim \mathcal{N}\left( 0, \frac{1}{n_{j(i)}} \right) = \mathcal{N}\left( 0, O\left( \frac{1}{n^\varepsilon} \right) \right),$$

so with very high probability this term is also $o(1)$, concluding our proof of Claim F.10.

$\square$

### F.2.7 LARGE INFLUENCE SCORES

For the final step of our proof of Claim F.2, we will show that with very high probability, there are many samples in the regression that have relatively high AMIP influence scores.

**Claim F.11.** *Let $\alpha_i = e^\intercal \widetilde{\Sigma}^{-1} \widetilde{X}_i R_i$ denote the AMIP influence score of the ith sample. Then with very high probability, there are is a set $T_0 \subseteq [n]$ of least $k_0 = \widetilde{\Omega}(n^{1-\varepsilon})$ "influential samples" – i.e., such that*

$$\forall i \in T \;\; \alpha_i \geq \left( \frac{\sqrt{\log(n)}}{n} \right) = \omega\left( \frac{1}{n} \right)$$

Proving Claim F.11 will also conclude our proof of Claim F.2, as this will show that all conditions required for a regression to be well-behaved are fulfilled with very high probability.

*Proof of Claim F.11.* The proof of Claim F.11 will follow the same approach as our proof of its ACRE counterpart – Claim E.12. Indeed, we first note that from the definition of $\widetilde{\Sigma}$ and Claim F.4, with very high probability,

$$\sum_{i\in[n]} \left( e^\intercal \widetilde{\Sigma}^{-1} \widetilde{X}_i \right)^2 = \sum_{i\in[n]} e^\intercal \widetilde{\Sigma}^{-1} \widetilde{X}_i \widetilde{X}_i^\intercal \widetilde{\Sigma}^{-1} e = e^\intercal \widetilde{\Sigma} e = \frac{1 \pm o(1)}{n}.$$

Moreover, from Claim F.8, with very high probability, the contribution of each individual sample to this sum is at most $\widetilde{O}\left( \frac{1}{n} \right)$-fraction of this total. Therefore, with very high probability

$$\left| \left\{ i \;\middle|\; \left| e^\intercal \widetilde{\Sigma}^{-1} \widetilde{X}_i \right| \geq \frac{1}{2n} \right\} \right| = \widetilde{\Omega}(n).$$

As in the proof of Claim E.12, we note that Claim F.10 guarantees that with very high probability $\forall i \;\; \left| R_i - (R_{\text{gt}})_i \right| = o(1) < \frac{1}{20}$. Moreover, because the ground truth residuals $(R_{\text{gt}})_i$ are normally distributed independently of anything else, it follows that so are $\rho_i \overset{\text{def}}{=} \text{sign}\left( e^\intercal \widetilde{\Sigma}^{-1} \widetilde{X}_i \right) \times (R_{\text{gt}})_i$.

Therefore, with very high probability, at least $\widetilde{\Omega}\left( n^{1-\varepsilon} \right)$ of the samples such that $\left| e^\intercal \widetilde{\Sigma}^{-1} \widetilde{X}_i \right| \geq \frac{1}{2n}$ have $\rho_i \geq \Omega(\sqrt{\log(n)})$ (this is because we can set the constants in the $\Omega$ to be such that the probability of each of $\rho_i$ being above this threshold is $\gg n^{-\varepsilon}$, allowing us to apply Hoeffding on the $\widetilde{\Omega}(n)$ iid $\rho_i$).

Therefore, for this set of samples it holds that

$$\alpha_i = e^\intercal \widetilde{\Sigma}^{-1} \widetilde{X}_i R_i = (1 \pm o(1)) \left| e^\intercal \widetilde{\Sigma}^{-1} \widetilde{X}_i \right| \rho_i = \Omega\left( \frac{\sqrt{\log(n)}}{n} \right).$$

$\square$

## F.3 PROOF OF LEMMAS F.7 AND F.9

Recall the lemma we wish to prove:

**Lemma F.12.** *Let $X_1, \ldots, X_n \sim \mathcal{X}$ be $n$ iid samples of a well-behaved distribution $\mathcal{X}$ with covariance $I_{d \times d}$. Let $\xi = \frac{1}{k} \sum_{i \in [k]} X_i$ be the empirical average over the first $k < 0.49n$ of these samples, and let $v \in \mathbb{R}^d$ be any vector that is independent of these first $k$ samples (but may depend on the other samples). Finally, denote by $\Sigma \overset{\text{def}}{=} \sum_{i \in [n]} X_i X_i^{\mathsf{T}}$ the unnormalized empirical second moment of these samples.*

*Then, with very high probability,*

$$\left| \xi^{\mathsf{T}} \Sigma^{-1} v \right| = \widetilde{O} \left( \frac{\|v\|}{\sqrt{k}n} \times \left( 1 + \frac{d\sqrt{k}}{n} \right) \right) .$$

Clearly, Lemma F.12 implies Lemma F.7 (set $\xi = \xi_{j_1}$ and $v = \xi_{j_2}$) and Lemma F.9 (set $\xi = \xi_j$ and $v = e$).

### F.3.1 PROOF SKETCH

We will split the proof of Lemma F.7 into 3 main steps. To do this, let $S = \sum_{i=k+1}^{n} X_i X_i^{\mathsf{T}} = \Sigma_{[n] \setminus [k]}$ be the contributions to the empirical covariance due to samples not amongst the first $k$, and let $C = \Sigma - S$ be the contributions from within the bucket. From a standard analysis, so long as $C \prec S$, we have

$$\Sigma^{-1} = S^{-1} - \Sigma^{-1} C S^{-1} = S^{-1} - S^{-1} C S^{-1} - \left( \Sigma^{-1} - S^{-1} \right) C S^{-1} ,$$

and these components will correspond to the main components of our analysis. We will show that the inequality $C \prec S$ holds with very high probability, before proving the following claims over the rest of this section:

**Claim F.13.** *With very high probability*

$$\left| \xi^{\mathsf{T}} S^{-1} v \right| = \widetilde{O} \left( \frac{\|v\|}{\sqrt{k} \times n} \right)$$

**Claim F.14.** *Let $w \in \mathbb{R}^d$ be any vector that does not depend on the first $k$ samples.*

*Then, with very high probability,*

$$\left| \xi^{\mathsf{T}} S^{-1} C w \right| = \widetilde{O} \left( \frac{\|w\|}{\sqrt{k}} \left( 1 + \frac{d\sqrt{k}}{n} \right) \right)$$

**Claim F.15.** *Let $w \in \mathbb{R}^d$ be any vector that does not depend on the first $k$ samples. Then, with very high probability,*

$$\left| \xi^{\mathsf{T}} \left( S^{-1} - \Sigma^{-1} \right) C w \right| = \widetilde{O} \left( \frac{\|w\|}{\sqrt{k}} \left( 1 + \frac{d\sqrt{k}}{n} \right) \right)$$

The proofs of Claims F.13, F.14 and F.15 will grow progressively more complex and each claim will build on ideas from the previous one. Throughout the latter two, the key challenge will be to deal with cases where both the $\zeta \overset{\text{def}}{=} k\xi$ term and the $C$ or $\Sigma$ multiplicand may depend on the same samples.

In these cases, we will proceed by applying a sort of divide-and-conquer approach by splitting the bucket into two subsets. For instance, instead of analyzing $\zeta^{\mathsf{T}} S^{-1} C w$ directly, we will split the samples in the bucket into two subsets and track their contributions to both $\zeta = \zeta_0 + \zeta_1$ and $C = C_0 + C_1$.

$$\zeta^{\mathsf{T}} S^{-1} C w = (\zeta_0 + \zeta_1) S^{-1} (C_0 + C_1) w =$$
$$= \underbrace{\zeta_0^{\mathsf{T}} S^{-1} C_0 w + \zeta_1^{\mathsf{T}} S^{-1} C_1 w}_{\text{diagonal terms}} + \underbrace{\zeta_1^{\mathsf{T}} S^{-1} C_0 w + \zeta_0^{\mathsf{T}} S^{-1} C_1 w}_{\text{off-diagonal terms}} \qquad (26)$$

The "off-diagonal" subsets will be relatively simple to bound as they contain inner products of independent vectors in $\mathbb{R}^d$ (and this independence will give us a $1/\sqrt{d}$ scaling to their inner product),

and the diagonal elements will be split again recursively. This will also leave us with a large number of "single sample" diagonal elements of the form

$$X_i^\mathsf{T} S^{-1} X_i X_i^\mathsf{T} w$$

These single-sample terms will no longer enjoy the same $1/\sqrt{d}$ scaling the other terms gain due to independence, but will instead gain a sort of $1/k$ scaling, because instead of having $k^2$ sample-times-sample contributions in

$$\zeta^\mathsf{T} S^{-1} C = \sum_{i,i' \in [k]} X_i^\mathsf{T} A X_{i'} X_{i'}^\mathsf{T} w \,,$$

we have only $k$ terms in the sum

$$\sum_{i \in [k]} X_i^\mathsf{T} A X_i X_i^\mathsf{T} w$$

Finally, in Claim F.15, we will have 2 types of diagonal vs off-diagonal splits. The first will be to track the cases where $\zeta$ and $C$ may depend on the same samples and will be very similar to our analysis of Claim F.14. The second and much more difficult of the two will be dealing with dependencies between $\Sigma$ and $\zeta$.

### F.3.2 Setup and Proof of Claim F.13

Recall our assumption from Lemma F.7 that $k \le 0.49n$. Therefore $n - k \ge 0.51n > k + \Omega(n)$.

Let $S = \sum_{i \in [n] \setminus [k]} X_i X_i^\mathsf{T} = \Sigma_{[n] \setminus [k]}$. From Claim E.7, we have that with very high probability

$$\|S - (n - k) I\| = o(n) \,.$$

Combined with Lemma E.6, which shows that $\xi$ is well-behaved, we have that with high probability

$$\left| \xi^\mathsf{T} S^{-1} v \right| = \widetilde{O}\left( \frac{1}{\sqrt{k}} \times \|S^{-1} v\| \right) = \widetilde{O}\left( \frac{1}{\sqrt{k}} \times \max\left\{ \lambda\left(S^{-1}\right) \right\} \times \|v\| \right) = \widetilde{O}\left( \frac{\|v\|}{\sqrt{k} \times n} \right)$$

Our goal for the rest of the proof will be to show a similar bound for $\xi^\mathsf{T} \Sigma^{-1} v$. Let $S$ and $C = \Sigma - S$ be our "main" and "correction" terms. Applying Claim E.7 again, we have that with very high probability $C \preceq kI + o(n) \prec (n - k)I - o(n) \preceq S$, so both $S$ and $I \pm S^{-1} C$ are invertible. We have

$$\Sigma^{-1} = (S + C)^{-1} = \left(I + S^{-1} C\right)^{-1} S^{-1} = S^{-1} - \left(I + S^{-1} C\right)^{-1} S^{-1} C S^{-1} = S^{-1} - \Sigma^{-1} C S^{-1}$$

Therefore, it remains to bound $\xi^\mathsf{T} \Sigma^{-1} C S^{-1} v = \xi^\mathsf{T} \Sigma^{-1} C w$ (where $w = S^{-1} v$) in absolute value. To do this, we once again use the intuition that in some sense $S \approx \Sigma$, and first bound $\xi^\mathsf{T} S^{-1} C S^{-1} v$. We will then slowly break down the difference between $\xi^\mathsf{T} S^{-1} C S^{-1} v$ and $\xi_{j_1}^\mathsf{T} \Sigma^{-1} C S^{-1} v$ into a series of corrections and bound each of these corrections in absolute value.

### F.3.3 Proof of Claim F.14

Before proving Claim F.14, we prove a lemma that will help us in our analysis.

**Lemma F.16.** *With very high probability*

$$\|Cw\| = \widetilde{O}\left( \left(k + \sqrt{kd}\right) \|w\| \right) \,.$$

*Proof of Lemma F.16.* At first glance, it might seem like Lemma F.16 should follow immediately from Claim E.7, but this is only true when $k = \widetilde{\Omega}(d)$, and we will want to apply Lemma F.16 even when $k \ll d$.

We prove Lemma F.16 using the approximate matrix Bernstein inequality (Lemma E.5). Indeed,

$$Cw = \sum_{i \in [k]} X_i X_i^\mathsf{T} w = \sum_{i \in [k]} v_i \,,$$

can be written as the sum of $k$ iid vectors $v_i \overset{\text{def}}{=} X_i X_i^\mathsf{T} w$.

From the fact that the $X_i$s are well behaved and independent of $w$, we have that with very high probability,

$$\|v_i\| \leq \sqrt{d}\, \|w\|\,.$$

Moreover, from our assumption that $\mathcal{X}$ has covariance identity, we have that

$$\mathbb{E}\left[v_i\right] = \mathbb{E}\left[X_i X_i^\mathsf{T}\right] w = Iw = w\,.$$

Finally, from Claim E.7, we have that

$$\mathbb{E}\left[\|v_i\|^2\right] = w^\mathsf{T} \mathbb{E}\left[\|X_i\|^2 X_i X_i^\mathsf{T}\right] w = \widetilde{O}\left(d\,\|w\|^2\right)\,.$$

Therefore, applying the approximate matrix Bernstein inequality on the standard embedding matrix

$$V_i = \begin{pmatrix} 0 & v_i^\mathsf{T} \\ v_i & 0 \end{pmatrix} \in \mathbb{R}^{(d+1)\times(d+1)}\,,$$

yields the desired result. $\qquad\qquad\square$

We now return to the proof of of Claim F.14.

*Proof of Claim F.14.* Let $\zeta \overset{\text{def}}{=} k\xi = \sum_{i\in[k]} X_i$. Using this notation, we have

$$\xi^\mathsf{T} S^{-1} C w = \frac{1}{k}\zeta^\mathsf{T} S^{-1} C w \tag{27}$$

We will bound the RHS of equation (27) by breaking the contributions to $\zeta$ and to $C$ into smaller and smaller subsets of the bucket $[k]$.

We will split the contributions from the first $k$ samples into "clusters" based on their index modulus some number, and the assumption above is just to ensure that the subsets are of roughly equal size.

Indeed, for any string of bits $a = (a_0, \ldots, a_t) \in \{0,1\}^*$, define the $a$th *cluster* of samples to be the set of indices from $[k]$ whose bitwise representation ends with the string $a$:

$$\mathcal{C}_a \overset{\text{def}}{=} \left\{i \in [k] \,\middle|\, i \mod 2^{t+1} = a_0 + a_1 2^1 + \cdots a_t 2^t\right\}$$

If $\epsilon$ is the empty string, then $\mathcal{C}_\epsilon = [k]$, and for all $a$, we have

$$\mathcal{C}_a = \mathcal{C}_{a0} \sqcup \mathcal{C}_{a1}$$

Similarly, we may split the contributions of samples in $[k]$ to $C$ and $\zeta$ based on their cluster

$$\begin{aligned} C_a &\overset{\text{def}}{=} \sum_{i\in\mathcal{C}_a} X_i X_i^\mathsf{T} \\ \zeta_a &\overset{\text{def}}{=} \sum_{i\in\mathcal{C}_a} X_i \end{aligned} \tag{28}$$

Moreover, we have the property that $C_a = C_{a0} + C_{a1}$ and $\zeta_a = \zeta_{a0} + \zeta_{a1}$. Using this property, we may begin to split the RHS of equation (27) to smaller components

$$\begin{aligned} \zeta^\mathsf{T} S^{-1} C w &= (\zeta_0 + \zeta_1)\, A\, (C_0 + C_1)\, w = \\ &= \underbrace{\zeta_0^\mathsf{T} S^{-1} C_0 w + \zeta_1^\mathsf{T} S^{-1} C_1 w}_{\text{diagonal terms}} + \underbrace{\zeta_1^\mathsf{T} S^{-1} C_0 w + \zeta_0^\mathsf{T} S^{-1} C_1 w}_{\text{off-diagonal terms}} \end{aligned} \tag{29}$$

We split the terms in the RHS of equation (29) into "diagonal" terms which correspond to the contributions where the $C$ term and the $\zeta$ term correspond to the same samples and "off-diagonal" terms where $\zeta$ and $C$ depend on disjoint sets of samples.

To bound the contribution of the off-diagonal terms, we note that for any bitstring $a = (a_0, \ldots, a_t)$, it holds that $\zeta_a$ is well-behaved (as it is the sum over $|\mathcal{C}_a|$ iid samples from $\mathcal{X}$) and has covariance $|\mathcal{C}_a| I = \Theta\left(\frac{k}{2^t}\right) I$ (this is because $|\mathcal{C}_a| \approx \frac{k}{2^{t+1}}$). Moreover, the other terms in the product do not depend on the samples in $\mathcal{C}_a$, so with very high probability

$$\left|\zeta_0^\mathsf{T} S^{-1} C_1 w\right| = \widetilde{O}\left(\sqrt{k} \times \left\|S^{-1} C_1 w\right\|\right).$$

Moreover, applying Claim E.7, we have that with very high probability

$$\|S - (n-k)I\| = o(n) \Rightarrow \left\|S^{-1}\right\| = O\left(\frac{1}{n}\right),$$

and applying Lemma F.16, we have that with very high probability,

$$\|C_1 w\| = \widetilde{O}\left(\left(k + \sqrt{kd}\right)\|w\|\right).$$

Therefore, with very high probability

$$\left|\zeta_0^\mathsf{T} S^{-1} C_1 w\right| = \widetilde{O}\left(\sqrt{k} \times \left\|S^{-1} C_1 w\right\|\right) = \widetilde{O}\left(\sqrt{k}\frac{k + \sqrt{kd}}{n}\|w\|\right),$$

and similarly for the other off-diagonal term, with very high probability

$$\left|\zeta_1^\mathsf{T} S^{-1} C_0 w\right| = \widetilde{O}\left(\sqrt{k}\frac{k + \sqrt{kd}}{n}\|w\|\right).$$

It now remains to bound the diagonal terms. Consider $\zeta_0^\mathsf{T} S^{-1} C_0 w$. We can open up the next bit of the indices of the samples to obtain

$$\begin{aligned}
\zeta_0^\mathsf{T} S^{-1} C_0 w &= (\zeta_{00} + \zeta_{01})^\mathsf{T} A (C_{00} + C_{01}) w = \\
&= \zeta_{00}^\mathsf{T} S^{-1} C_{00} w + \zeta_{01}^\mathsf{T} S^{-1} C_{01} w + \zeta_{00}^\mathsf{T} S^{-1} C_{01} w + \zeta_{01}^\mathsf{T} S^{-1} C_{00} w
\end{aligned} \tag{30}$$

We split the RHS of equation (30) again into diagonal and off-diagonal terms. The off-diagonal terms can be bounded again in exactly the same manner and the off diagonal can again be split by specifying another bit of the sample indices. Applying this logic recursively, we have

$$\text{Diagonal}_\epsilon = \zeta^\mathsf{T} S^{-1} C w = \underbrace{\zeta_0^\mathsf{T} S^{-1} C_0 w}_{\text{Diagonal}_0} + \underbrace{\zeta_1^\mathsf{T} S^{-1} C_1 w}_{\text{Diagonal}_1} + \underbrace{\zeta_1^\mathsf{T} S^{-1} C_0 w}_{\text{Off-Diagonal}_{1,0}} + \underbrace{\zeta_0^\mathsf{T} S^{-1} C_1 w}_{\text{Off-Diagonal}_{0,1}}$$

$$= \text{Diagonal}_{00} + \text{Diagonal}_{01} + \text{Diagonal}_{10} + \text{Diagonal}_{11} + \text{Off-Diagonal}_{1,0} + \text{Off-Diagonal}_{0,1} +$$

$$+ \text{Off-Diagonal}_{00,01} + \text{Off-Diagonal}_{01,00} + \text{Off-Diagonal}_{10,11} + \text{Off-Diagonal}_{11,10} = \cdots$$

$$\cdots = \sum_{i\in[k]} \text{Diagonal}_i + \sum_{t=1}^{t=\lceil \log_2(k)\rceil} \sum_{a\in\{0,1\}^t} \text{Off-Diagonal}_{a|0,a|1} + \text{Off-Diagonal}_{a|1,a|0} \tag{31}$$

From the same analysis as the one above, we see that for any bitstring $a \in \{0,1\}^t$ and bit $b \in \{0,1\}$, it holds that with very high probability

$$\left|\text{Off-Diagonal}_{a|b,a|\bar{b}}\right| = \widetilde{O}\left(\sqrt{|\mathcal{C}_a|}\frac{|\mathcal{C}_a| + \sqrt{|\mathcal{C}_a|d}}{n}\|w\|\right) = \widetilde{O}\left(2^{-t}\sqrt{k}\frac{k + \sqrt{kd}}{n}\|w\|\right)$$

Union bounding over the $O\left(2^t\right) = \text{poly}(n)$ off-diagonal combinations, we see that with very high probability they are all bounded. Summing over these off-diagonal terms gives us

$$\left|\sum_{t=1}^{t=\lceil \log_2(k)\rceil} \sum_{a\in\{0,1\}^t} \text{Off-Diagonal}_{a|0,a|1} + \text{Off-Diagonal}_{a|1,a|0}\right| \leq$$

$$\leq \sum_{t=1}^{t=\lceil \log_2(k)\rceil} 2^t \widetilde{O}\left(2^{-t}\sqrt{k}\frac{k + \sqrt{kd}}{n}\|w\|\right) = \widetilde{O}\left(\sqrt{k}\frac{k + \sqrt{kd}}{n}\|w\|\right).$$

Now, consider a Diagonal$_i$ term $X_i^\mathsf{T} S^{-1} X_i X_i^\mathsf{T} w$. Because $w$ is independent of the $i$th sample $X_i$, with very high probability,

$$|X_i^\mathsf{T} w| = \widetilde{O}\left(\|w\|\right) ,$$

and from Claim E.7, we have that with very high probability,

$$\left|X_i^\mathsf{T} S^{-1} X_i\right| \le \|X_i\|^2 \left\|S^{-1}\right\| = \widetilde{O}\left(\frac{d}{n}\right) .$$

Therefore, from the triangle inequality, with very high probability

$$\left|\sum_{i\in[k]} \text{Diagonal}_i\right| \le \sum_{i\in[k]} |\text{Diagonal}_i| = \widetilde{O}\left(\frac{kd}{n}\|w\|\right) .$$

Altogether, we have

$$\left|\xi^\mathsf{T} S^{-1} C w\right| = \widetilde{O}\left(\frac{\|w\|}{\sqrt{k}} \times \left(1 + \frac{d\sqrt{k}}{n}\right)\right) ,$$

concluding the proof of Claim F.14. □

### F.3.4 PROOF OF CLAIM F.15

In the previous portion of the proof, we bounded $\xi^\mathsf{T} S^{-1} C w$, when $w$ is independent of $\xi$. The rest of our analysis will be devoted to bounding the effect that replacing $S^{-1}$ with $\Sigma^{-1}$ will not make this inner product much larger.

As in the previous portion of the proof, we set $\zeta = k\xi$, and separate samples into clusters based on the least significant bits of the bitwise representations of their indices. In particular, for any bitstring $a$, let $\mathcal{C}_a$, $\zeta_a$ and $C_a$ be as defined above, and define

$$S_a = \Sigma - C_a = \Sigma_{[n]\setminus\mathcal{C}_a} = \sum_{i\in[n]\setminus\mathcal{C}_a} X_i X_i^\mathsf{T}$$

to be the unnormalized empirical covariance of the samples *not* in the $\mathcal{C}_a$ cluster.

As in the proof of Claim F.14, we will separate the contributions to $\zeta^\mathsf{T}\left(S^{-1} - \Sigma^{-1}\right) C w$ based on the cluster of the samples and label these contributions as diagonal or off-diagonal based on whether or not the same samples were used in $\zeta$ and $C$.

$$\zeta^\mathsf{T}\left(S^{-1} - \Sigma^{-1}\right) C w = (\zeta_0 + \zeta_1)^\mathsf{T}\left(S^{-1} - \Sigma^{-1}\right)(C_0 + C_1) w =$$
$$= \underbrace{\zeta_0^\mathsf{T}\left(S^{-1} - \Sigma^{-1}\right) C_0 w + \zeta_1^\mathsf{T}\left(S^{-1} - \Sigma^{-1}\right) C_1 w}_{\text{diagonal terms}} +$$
$$+ \underbrace{\zeta_0^\mathsf{T}\left(S^{-1} - \Sigma^{-1}\right) C_1 w + \zeta_1^\mathsf{T}\left(S^{-1} - \Sigma^{-1}\right) C_0 w}_{\text{off-diagonal terms}} \tag{32}$$

In other words, we have the recursive formula that for all $a \in \{0,1\}^*$,

$$\text{Diagonal Term}_a = \text{Diagonal Term}_{a|0} + \text{Diagonal Term}_{a|1} +$$
$$+ \text{Off-Diagonal Term}_{a|0,a|1} + \text{Off-Diagonal Term}_{a|1,a|0} \tag{33}$$

Applying equation (33) recursively, we have

$$\zeta^\mathsf{T}\left(S^{-1} - \Sigma^{-1}\right) C w = \text{Diagonal Term}_\epsilon = \cdots = \sum_{i\in[k]} \text{Diagonal Term}_i +$$
$$+ \sum_{t=0}^{\lceil\log(k)\rceil} \sum_{a\in\{0,1\}^t} \left(\text{Off-Diagonal Term}_{a|0,a|1} + \text{Off-Diagonal Term}_{a|1,a|0}\right)$$

In Claim F.17, we will bound the off-diagonal terms. We will show that with very high probability,

$$\left| \text{Off-Diagonal Term}_{a,b} \right| = \widetilde{O}\left( \frac{\sqrt{|\mathcal{C}_a|}}{n} \times \left( |\mathcal{C}_b| + \sqrt{|\mathcal{C}_b|d} \right) \times \|w\| \times \left( 1 + \frac{d\sqrt{k}}{n} \right) \right) =$$

$$= \widetilde{O}\left( 2^{-(|a|+|b|)/2} \frac{\sqrt{k}}{n^2} \times \left( k + \sqrt{kd} \right) \times \|w\| \times \left( 1 + \frac{d\sqrt{k}}{n} \right) \right) .$$

Since in our case $|a| = |b| = t$, we may conclude that with very high probability

$$\left| \text{Off-Diagonal Term}_{a,b} \right| = \widetilde{O}\left( 2^{-t} \frac{\sqrt{k}}{n^2} \times \left( k + \sqrt{kd} \right) \times \|w\| \times \left( 1 + \frac{d\sqrt{k}}{n} \right) \right) .$$

Applying the triangle inequality, with very high probability the total contribution of all the off-diagonal terms is of order

$$\left| \sum_{t=0}^{\lceil \log(k) \rceil} \sum_{a \in \{0,1\}^t} \left( \text{Off-Diagonal Term}_{a|0,a|1} + \text{Off-Diagonal Term}_{a|1,a|0} \right) \right| =$$

$$= \sum_{t=0}^{\lceil \log(k) \rceil} \sum_{a \in \{0,1\}^t} \left| \text{Off-Diagonal Term}_{a|0,a|1} \right| + \left| \text{Off-Diagonal Term}_{a|1,a|0} \right| =$$

$$= \sum_{t=0}^{\lceil \log(k) \rceil} \sum_{a \in \{0,1\}^t} \widetilde{O}\left( 2^{-t} \frac{\sqrt{k}}{n^2} \times \left( k + \sqrt{kd} \right) \times \|w\| \times \left( 1 + \frac{d\sqrt{k}}{n} \right) \right) =$$

$$= \sum_{t=0}^{\lceil \log(k) \rceil} \widetilde{O}\left( \frac{\sqrt{k}}{n^2} \times \left( k + \sqrt{kd} \right) \times \|w\| \times \left( 1 + \frac{d\sqrt{k}}{n} \right) \right) =$$

$$= \widetilde{O}\left( \frac{\sqrt{k}}{n^2} \times \left( k + \sqrt{kd} \right) \times \|w\| \times \left( 1 + \frac{d\sqrt{k}}{n} \right) \right)$$

This leaves us with only the "single-sample" diagonal terms

$$\text{Diagonal Term}_i = X_i^\mathsf{T} \left( S^{-1} - \Sigma^{-1} \right) X_i X_i^\mathsf{T} w .$$

To analyse this term, simply note that from Claim E.7 and the CS inequality, with very high probability,

$$\left| X_i^\mathsf{T} \left( S^{-1} - \Sigma^{-1} \right) X_i \right| \le \|X_i\|^2 \left( \|S^{-1}\| + \|\Sigma^{-1}\| \right) = \widetilde{O}\left( \frac{d}{n} \right) .$$

Moreover, from our assumption that $X_i \sim \mathcal{X}$ is well-behaved and that $w$ is independent of $X_i$, we have that with very high probability

$$\left| X_i^\mathsf{T} w \right| = \widetilde{O}\left( \|w\| \right) .$$

Therefore, with very high probability

$$\frac{1}{k} \sum_{i \in [k]} \left| \text{Diagonal Term}_i \right| = \widetilde{O}\left( \frac{\|w\|}{\sqrt{k}} \times \frac{d\sqrt{k}}{n} \right) .$$

**Bounding the Off-Diagonal Terms**

**Claim F.17.** *Consider the off-diagonal term*

$$\zeta_a^\mathsf{T} \left( S^{-1} - \Sigma^{-1} \right) \mathcal{C}_b w ,$$

*where $a \ne b$ are bitstrings representing disjoint clusters $\mathcal{C}_a \cap \mathcal{C}_b = \emptyset$.*

*With very high probability,*

$$\zeta_a^\mathsf{T} \left( S^{-1} - \Sigma^{-1} \right) \mathcal{C}_b w = \widetilde{O}\left( \frac{\sqrt{|\mathcal{C}_a|}}{n} \times \left( |\mathcal{C}_b| + \sqrt{|\mathcal{C}_b|d} \right) \times \|w\| \times \left( 1 + \frac{d\sqrt{k}}{n} \right) \right) .$$

*Proof of Claim F.17.* Our goal is to bound the following term in absolute value

$$\zeta_a^\mathsf{T} \left( S^{-1} - \Sigma^{-1} \right) C_b w \,.$$

The difficulty in analysing this term is that both $\zeta_a$ and $\left( S^{-1} - \Sigma^{-1} \right)$ may depend on the same samples. To circumvent this, we begin by splitting $\left( S^{-1} - \Sigma^{-1} \right)$ into a term that is easy to deal with and a small correction:

$$\left( S^{-1} - \Sigma^{-1} \right) = \left( S^{-1} - S_a^{-1} \right) + \left( S_a^{-1} - \Sigma^{-1} \right) \,,$$

where $S_a = \Sigma - C_a = \sum_{i \notin \mathcal{C}_a} X_i X_i^\mathsf{T}$.

The first component has the property that it does not depend on the samples in $\mathcal{C}_a$, while $\zeta_a$ depends only on the samples in $a$. Therefore, because $\zeta_a$ is well-behaved with covariance $|\mathcal{C}_a| I$, with very high probability

$$\left| \zeta_a^\mathsf{T} \left( S^{-1} - S_a^{-1} \right) C_b w \right| = \widetilde{O} \left( \sqrt{|\mathcal{C}_a|} \left\| \left( S^{-1} - S_a^{-1} \right) C_b w \right\| \right)$$

We begin by bounding the norm of $\left( S^{-1} - S_a^{-1} \right) C_b w$. First, we note that Lemma F.16, with very high probability

$$\| C_b w \| = \widetilde{O} \left( \left( |\mathcal{C}_b| + \sqrt{|\mathcal{C}_b| d} \right) \| w \| \right) \,.$$

To continue, we bound the norm of $\left( S^{-1} - S_a^{-1} \right)$. Let $A = S_a$ and $B = S_a - S = \sum_{i \in [k] \setminus \mathcal{C}_a} X_i X_i^\mathsf{T}$. We utilize the identity

$$(A - B)^{-1} - A^{-1} = (A - B)^{-1} B A^{-1} \,,$$

as well as Claim E.7 which states that with very high probability $\left\| (A - B)^{-1} \right\|, \left\| A^{-1} \right\| = O \left( \frac{1}{n} \right)$ and that $\| B \| = \widetilde{O} (k + d)$ to show that with very high probability

$$\left\| S^{-1} - S_a^{-1} \right\| = \widetilde{O} \left( \frac{k + d}{n^2} \right)$$

Therefore, with very high probability

$$\left| \zeta_a^\mathsf{T} \left( S^{-1} - S_a^{-1} \right) C_b w \right| = \widetilde{O} \left( \sqrt{|\mathcal{C}_a|} \times \frac{k + d}{n^2} \times \left( |\mathcal{C}_b| + \sqrt{|\mathcal{C}_b| d} \right) \times \| w \| \right) =$$

$$= \widetilde{O} \left( \frac{\sqrt{|\mathcal{C}_a|}}{n} \times \left( |\mathcal{C}_b| + \sqrt{|\mathcal{C}_b| d} \right) \times \| w \| \right) \,.$$

It remains to bound the contribution of the $S_a^{-1} - \Sigma^{-1}$ term, which brings with it the added difficulty that it may depend on the samples in $\mathcal{C}_a$. To bound the effect of this term, we split the contributions to $\zeta$ once more

$$
\begin{aligned}
\text{Hard Term}_{a,b} &\stackrel{\text{def}}{=} \zeta_a^\mathsf{T} \left( S_a^{-1} - \Sigma^{-1} \right) C_b w = \\
&= (\zeta_{a0} + \zeta_{a1})^\mathsf{T} \left( S_a^{-1} - \Sigma^{-1} \right) C_b w = \\
&= \underbrace{\zeta_{a0}^\mathsf{T} \left( S_a^{-1} - S_{a0}^{-1} \right) C_b w + \zeta_{a1}^\mathsf{T} \left( S_a^{-1} - S_{a1}^{-1} \right) C_b w}_{\text{Easy Terms}} + \\
&\quad + \underbrace{\zeta_{a0}^\mathsf{T} \left( S_{a0}^{-1} - \Sigma^{-1} \right) C_b w + \zeta_{a1}^\mathsf{T} \left( S_{a1}^{-1} - \Sigma^{-1} \right) C_b w}_{\text{Hard Terms}} = \\
&= \text{Easy Term}_{a0,b} + \text{Easy Term}_{a1,b} + \text{Hard Term}_{a0,b} + \text{Hard Term}_{a1,b}
\end{aligned}
\tag{34}
$$

We split the right hand side of equation (34) into "easy" terms which can be dealt with using the same logic above and hard terms which can again be split into smaller easy and hard terms. Applying equation (34) recursively, we have that:

$$\text{Hard Term}_{a,b} = \text{Easy Term}_{a,0,b} + \text{Easy Term}_{a,1,b} + \text{Hard Term}_{a0,b} + \text{Hard Term}_{a1,b} =$$

$$\vdots$$

$$= \sum_{i \in \mathcal{C}_a} \text{Hard Term}_{i,b} + \sum_{t=1}^{\log(|\mathcal{C}_a|)+1} \sum_{\substack{a' \in \{0,1\}^{t-1} \\ z \in \{0,1\}}} \text{Easy Term}_{a|a',z,b} \tag{35}$$

To bound the easy terms, we note that as in the analysis above, the samples included in their $\zeta$ are disjoint from the samples included in their other factors. Therefore, with very high probability

$$\left| \text{Easy Term}_{a,z,b} \right| \overset{\text{def}}{=} \left| \zeta_{az}^{\mathsf{T}} \left( S_a^{-1} - S_{az}^{-1} \right) C_b w \right| = \widetilde{O} \left( \sqrt{|\mathcal{C}_{az}|} \times \left\| \left( S_a^{-1} - S_{az}^{-1} \right) C_b w \right\| \right)$$

Recall that we showed above that with very high probability

$$\|C_b w\| = \widetilde{O} \left( \left( |\mathcal{C}_b| + \sqrt{|\mathcal{C}_b|d} \right) \times \|w\| \right) .$$

From here, we set $A = S_{az}$ and $B = C_{a\bar{z}} = S_{az} - S_a$, and recall the identity

$$(A - B)^{-1} - A^{-1} = A^{-1} B (A - B)^{-1}$$

to obtain the equation

$$S_a^{-1} - S_{az}^{-1} = S_{az}^{-1} C_{a\bar{z}} S_a^{-1} .$$

From here, it might be tempting to simply bound the norm of this product, but that would result in too loose of a bound. Instead, we perform the somewhat finer relaxation

$$\left\| \left( S_a^{-1} - S_{az}^{-1} \right) C_b w \right\| = \left\| S_{az}^{-1} C_{a\bar{z}} S_a^{-1} C_b w \right\| \le \left\| S_{az}^{-1} \right\| \times \left\| C_{a\bar{z}} S_a^{-1} C_b w \right\| .$$

The advantage of this finer analysis is that we can now use Lemma F.16 again, but this time on the matrix $C_{a\bar{z}}$. Indeed, $S_a$ depends only on the samples outside $\mathcal{C}_a \supseteq \mathcal{C}_{a\bar{z}}$, $C_b$ depends only on the samples in $\mathcal{C}_b$ which is disjoint from $\mathcal{C}_a$, and $S$ and $w$ do not depend on the first $k$ samples. Therefore, $C_{a\bar{z}}$ is independent of them all, so applying Lemma F.16, we have that with very high probability

$$\left\| C_{a\bar{z}} S_a^{-1} C_b w \right\| = \widetilde{O} \left( |\mathcal{C}_{a\bar{z}}| + \sqrt{|\mathcal{C}_{a\bar{z}}|d} \right) \times \left\| S_a^{-1} C_b w \right\| \le \widetilde{O} \left( |\mathcal{C}_{a\bar{z}}| + \sqrt{|\mathcal{C}_{a\bar{z}}|d} \right) \times \left\| S_a^{-1} \right\| \|C_b w\| =$$

$$= \widetilde{O} \left( \frac{|\mathcal{C}_{a\bar{z}}| + \sqrt{|\mathcal{C}_{a\bar{z}}|d}}{n} \times \left( |\mathcal{C}_b| + \sqrt{|\mathcal{C}_b|d} \right) \times \|w\| \right)$$

Altogether, we have that with very high probability

$$\left| \text{Easy Term}_{a,z,b} \right| = \widetilde{O} \left( \sqrt{|\mathcal{C}_a|} \times \frac{|\mathcal{C}_a| + \sqrt{|\mathcal{C}_a|d}}{n^2} \times \left( |\mathcal{C}_b| + \sqrt{|\mathcal{C}_b|d} \right) \times \|w\| \right) .$$

Therefore, for $a'$ of length $t-1$, we have $\left| \mathcal{C}_{a|a'} \right| \approx 2^{-t} |\mathcal{C}_a|$, so with very high probability

$$\left| \text{Easy Term}_{a|a',z,b} \right| = \widetilde{O} \left( \sqrt{|\mathcal{C}_{a|a'}|} \times \frac{|\mathcal{C}_{a|a'}| + \sqrt{|\mathcal{C}_{a|a'}|d}}{n^2} \times \left( |\mathcal{C}_b| + \sqrt{|\mathcal{C}_b|d} \right) \times \|w\| \right) =$$

$$= \widetilde{O} \left( 2^{-t} \sqrt{|\mathcal{C}_a|} \times \frac{2^{-t/2}|\mathcal{C}_a| + \sqrt{|\mathcal{C}_a|d}}{n^2} \times \left( |\mathcal{C}_b| + \sqrt{|\mathcal{C}_b|d} \right) \times \|w\| \right)$$

Therefore, summing over all of the easy terms and applying the triangle inequality, we have that with very high probability

$$\left| \sum_{t=1}^{\log(|\mathcal{C}_a|)+1} \sum_{\substack{a' \in \{0,1\}^{t-1} \\ z \in \{0,1\}}} \text{Easy Term}_{a|a',z,b} \right| \leq \sum_{t=1}^{\log(|\mathcal{C}_a|)+1} \sum_{\substack{a' \in \{0,1\}^{t-1} \\ z \in \{0,1\}}} \left| \text{Easy Term}_{a|a',z,b} \right| =$$

$$= \widetilde{O}\left( \sqrt{|\mathcal{C}_a|} \frac{|\mathcal{C}_a| + \sqrt{d|\mathcal{C}_a|}}{n^2} \times \left( |\mathcal{C}_b| + \sqrt{|\mathcal{C}_b|d} \right) \times \|w\| \right) =$$

$$= \widetilde{O}\left( \frac{\sqrt{|\mathcal{C}_a|}}{n} \times \left( |\mathcal{C}_b| + \sqrt{|\mathcal{C}_b|d} \right) \times \|w\| \right)$$

It remains to bound the single-sample hard terms. That is, we want to bound

$$\left| X_i^{\mathsf{T}} \left( \Sigma_{[n]\setminus\{i\}}^{-1} - \Sigma^{-1} \right) C_b w \right|.$$

Using the same matrix identity as in the previous terms, we have that

$$\Sigma_{[n]\setminus\{i\}}^{-1} - \Sigma^{-1} = \Sigma^{-1} X_i X_i^{\mathsf{T}} \Sigma_{[n]\setminus\{i\}}^{-1}$$

Therefore, with very high probability, we have

$$\left| X_i^{\mathsf{T}} \left( \Sigma_{[n]\setminus\{i\}}^{-1} - \Sigma^{-1} \right) C_b w \right| \leq \|X_i\| \left\| \left( \Sigma_{[n]\setminus\{i\}}^{-1} - \Sigma^{-1} \right) C_b w \right\| =$$

$$= \|X_i\| \left\| \Sigma^{-1} X_i X_i^{\mathsf{T}} \Sigma_{[n]\setminus\{i\}}^{-1} C_b w \right\| \leq$$

$$\leq \|X_i\| \left\| \Sigma^{-1} \right\| \|X_i\| \left| X_i^{\mathsf{T}} \Sigma_{[n]\setminus\{i\}}^{-1} C_b w \right| \leq$$

$$\leq \|X_i\| \left\| \Sigma^{-1} \right\| \|X_i\| \times \widetilde{O}\left( \left\| \Sigma_{[n]\setminus\{i\}}^{-1} C_b w \right\| \right) =$$

$$= \widetilde{O}\left( \|X_i\|^2 \left\| \Sigma^{-1} \right\| \left\| \Sigma_{[n]\setminus\{i\}}^{-1} \right\| \|C_b w\| \right) = \widetilde{O}\left( \frac{d\left( |\mathcal{C}_b| + \sqrt{|\mathcal{C}_b|d} \right)}{n^2} \|w\| \right)$$

$$(36)$$

Therefore, from the triangle inequality, with very high probability

$$\left| \sum_{i \in \mathcal{C}_a} \text{Hard Term}_{i,b} \right| \leq \sum_{i \in \mathcal{C}_a} |\text{Hard Term}_{i,b}| = \widetilde{O}\left( |\mathcal{C}_a| \frac{d\left( |\mathcal{C}_b| + \sqrt{\mathcal{C}_b d} \right)}{n^2} \|w\| \right),$$

completing our proof of Claim F.17. □

## F.4 Proof of Theorem F.1

Recall that the OHARE algorithm works by computing each of the MSN style bounds produced by the ACRE algorithm and then adds a correction term to each one, where these correction terms correspond to the indirect removal effects due to the change in the reaveraging step.

In the previous subsection, we proved Claim F.2 which shows that with very high probability the ACRE components of the OHARE algorithm will produce good bounds on well-behaved regressions with categorical features. In order to conclude Theorem F.1, we would also need to bound the higher order corrections that OHARE takes into account.

Finally, note that for the case of an unweighted one-hot encoding, we have $u_{i,j} = 1_{i \in B_j}$ (i.e., the columns corresponding to the dummy variables are indicators of their respective buckets).

### F.4.1 INDIRECT CONTRIBUTIONS TO THE FIRST ORDER TERM

Analysing the indirect contributions to the first order term in OHARE will require significantly more care than our analysis of the higher order terms. This is because the first order term is the dominant one to begin with and the indirect contributions to it are smaller than the main effect by only a $\sqrt{\log(n)}$ factor, forcing us to track $\mathrm{polylog}(n)$ factors much more carefully.

Recall from Section A.2, that the OHARE algorithm bounds the first order effect of removals on the regression result from above/below by

$$\mathrm{bound}_j^\pm(k_j) = d_j(k_j) + \frac{c_j^\pm(k_j)}{n_j - k_j}$$

where $d_j$ represents AMIP gradients on the reaveraged samples $\widetilde{X}_i$:

$$d_j(k_j) = \max_{\substack{T_j \subseteq B_j \\ |T_j|=k_j}} \sum_{i \in T_j} R_i e^\intercal \widetilde{\Sigma}^{-1} \widetilde{X}_i \,,$$

and $c_j^\pm(k_j)$ represents the effect that reaveraging after the removals can have on the AMIP gradients of the retained samples:

$$c_j^+(k_j) = \max \left\{ \left( \max_{\substack{T_j \subseteq B_j \\ |T_j|=k_j}} \sum_{i \in T_j} R_i \right) \left( \max_{\substack{T_j \subseteq B_j \\ |T_j|=k_j}} \sum_{i \in T_j} e^\intercal \widetilde{\Sigma}^{-1} \widetilde{X}_i \right), \left( \min_{\substack{T_j \subseteq B_j \\ |T_j|=k_j}} \sum_{i \in T_j} R_i \right) \left( \min_{\substack{T_j \subseteq B_j \\ |T_j|=k_j}} \sum_{i \in T_j} e^\intercal \widetilde{\Sigma}^{-1} \widetilde{X}_i \right) \right\}$$

$$c_j^-(k_j) = \max \left\{ \left( \min_{\substack{T_j \subseteq B_j \\ |T_j|=k_j}} \sum_{i \in T_j} R_i \right) \left( \max_{\substack{T_j \subseteq B_j \\ |T_j|=k_j}} \sum_{i \in T_j} e^\intercal \widetilde{\Sigma}^{-1} \widetilde{X}_i \right), \left( \max_{\substack{T_j \subseteq B_j \\ |T_j|=k_j}} \sum_{i \in T_j} R_i \right) \left( \min_{\substack{T_j \subseteq B_j \\ |T_j|=k_j}} \sum_{i \in T_j} e^\intercal \widetilde{\Sigma}^{-1} \widetilde{X}_i \right) \right\}$$

For $k_j = n_j$, there is no reaveraging effect and $\mathrm{bound}_j^\pm(n_j) = d_j(n_j)$.

Our goal will be to show that $d_j(k_j)$, which is the contribution of the AMIP gradients to this first order effect, is the dominant effect. In particular, the main claim we will prove in this subsection is

**Claim F.18.** *Let $k_{\mathrm{threshold}} = \widetilde{\Omega}\left(n^{1-\varepsilon}\right)$ be as promised by Claim F.11.*

*Then, with very high probability*

$$\forall k \leq k_{\mathrm{threshold}}, \quad \max_{k_1+\cdots+k_m=k} \left\{ \sum_{j \in [m]} \mathrm{bound}_j^\pm(k_j) \right\} = \left(1 \pm O\left(\frac{1}{\sqrt{\log(n)}}\right)\right) \mathrm{AMIP}(k) \,,$$

*where*

$$\mathrm{AMIP}(k) = \max_{T \in \binom{[n]}{k}} \left\{ \sum_{i \in T} \alpha_i \right\}$$

*is the sum over the $k$ largest AMIP influence scores*

$$\alpha_i = e^\intercal \widetilde{\Sigma}^{-1} \widetilde{X}_i R_i \,.$$

*Proof of Claim F.18.* We expect $d_j(k_j)$ to grow roughly linearly with $k_j$, which motivates us to focus on the expression

$$\text{Scaled Indirect Effect} = \eta_j(k_j) \stackrel{\mathrm{def}}{=} \frac{\max\left\{\left|c_j^+(k_j)\right|, \left|c_j^-(k_j)\right|\right\}}{k_j(n_j - k_j)} \,.$$

We will bound these $\eta_j$ in the following lemma:

**Claim F.19.** *There exists some $\nu = \mathrm{polylog}(n)$ such that for all $k_j \leq \frac{n_j}{\nu}$, with very high probability*

$$\eta_j(k_j) = O\left(\frac{1}{n}\right) \,.$$

*Moreover, for any $\nu' = \text{polylog}(n)$ there exists some threshold $\tau = \text{polyloglog}(n)$, such that with very high probability,*

$$\left|\{i \in B_j \mid |R_i| > \tau\}\right| < \frac{n_j}{\nu'}$$

$$\left|\left\{i \in B_j \mid \left|e^{\mathsf{T}}\widetilde{\Sigma}^{-1}\widetilde{X}_i\right| > \frac{\tau}{n}\right\}\right| < \frac{n_j}{\nu'}$$

*In particular, for all $k_j \geq \frac{n_j}{\nu}$, with very high probability*

$$\max_{T_j \in \binom{B_j}{k_j}} \left\{\sum_{i \in T_j} \alpha_i\right\} = O\left(\frac{\text{polyloglog}(n)}{n} \times k_j\right)$$

$$\eta_j(k_j) = O\left(\frac{\text{polyloglog}(n)}{n}\right),$$

*where $\alpha_i = e^{\mathsf{T}}\widetilde{\Sigma}^{-1}\widetilde{X}_i R_i$ are the AMIP influence scores.*

The first part of Claim F.19 promises that the $\eta_j$ components to the bound are very small in any bucket for which there are not too many removals $k_j < \frac{n_j}{\text{polylog}(n)}$, while the second portion of the claim will help us bound the total contribution of buckets from which more than $\frac{n_j}{\text{polylog}(n)}$ have been removed.

Let $T = \text{argmax}_{T \in \binom{[n]}{k}} \left\{\sum_{i \in T} \alpha_i\right\}$ denote the set of $k$ samples with largest AMIP influences, and let $\kappa_j = |B_j \cap T|$ denote the distribution of these samples across the buckets. By definition, we have

$$\sum_{i \in T} \alpha_i = \text{AMIP}(k).$$

From Claim F.11, we know that with very high probability all the samples in $T$ must have influence at least $\min_{i \in T}\{\alpha_i\} = \Omega\left(\frac{\sqrt{\log(n)}}{n}\right)$. But from the second part of Claim F.19, we know that with very high probability, for all $j$, the $j$th bucket cannot have more than $\frac{n_j}{\nu}$ such samples, so with very high probability $\kappa_j < \frac{n_j}{\nu}$.

Consider our maximization problem

$$\text{MaxScore} = \max_{k_1 + \cdots + k_m = k} \left\{\sum_{j \in [m]} \text{bound}_j^{\pm}(k_j)\right\}.$$

This maximization is lower bounded by every valid assignment to $k_1, \ldots, k_m$, so in particular it is lower bounded by the score of $\kappa_1, \ldots, \kappa_m$. Utilising the first part of Claim F.19 to bound $\eta_j(\kappa_j)$, we have

$$\max_{k_1 + \cdots + k_m = k} \left\{\sum_{j \in [m]} \text{bound}_j^{\pm}(k_j)\right\} \geq \sum_{j \in [m]} \text{bound}_j^{\pm}(\kappa_j) = \sum_{i \in T} \alpha_i - \sum_{j \in [m]} \kappa_j \eta_j(\kappa_j) \in \text{AMIP}(k) - O\left(\frac{k}{n}\right).$$

Now, consider any other assignment $k_1, \ldots, k_m$. If we still have $k_j \leq \frac{n_j}{\nu}$ for all $j$, then $\sum_{j \in [m]} \text{bound}_j^{\pm}(k_j)$ will still be bounded by $\text{AMIP}(k) \pm O\left(\frac{k}{n}\right)$ following the same logic as above.

Otherwise, let $j_1, \ldots, j_\ell$ denote the set of buckets for whick $k_{j_i} > \frac{n_{j_i}}{\nu}$, and define $k' \stackrel{\text{def}}{=} k - k_{j_1} - \cdots - k_{j_\ell}$. From the same analysis as above, we have

$$\sum_{j \in [m]} \text{bound}_j^{\pm}(k_j) \leq \text{AMIP}(k') + \sum_{i \in [\ell]} \left\{\text{bound}_{j_i}^{\pm}(k_{j_i})\right\} + O\left(\frac{k'}{n}\right).$$

Next, note that we know that with very high probability

$$\text{AMIP}(k) \geq \text{AMIP}(k') + \Omega\left(\frac{\sqrt{\log(n)}\,(k - k')}{n}\right),$$

because Claim F.11 tells us that with very high probability, there are at least $k_{\text{threshold}} \geq k$ samples, each of which has a sufficiently large contribution to the AMIP score.

Therefore, using the last part of Claim F.19, which bounds with very high probability every term in the bounds of buckets with more than $\frac{n_j}{\nu}$ removals, we have

$$\sum_{j \in [m]} \text{bound}_j^{\pm}(k_j) \leq \text{AMIP}(k') + \sum_{i \in [\ell]} \left\{\text{bound}_{j_i}^{\pm}(k_{j_i})\right\} + O\left(\frac{k'}{n}\right) \leq$$

$$\leq \text{AMIP}(k) + \underbrace{O\left(\frac{\text{polyloglog}(n) \times (k - k')}{n}\right) - \Omega\left(\frac{\sqrt{\log(n)}\,(k - k')}{n}\right)}_{\leq 0} \pm O\left(\frac{k'}{n}\right) \leq$$

$$\leq \text{AMIP}(k) + O\left(\frac{k}{n}\right).$$

Altogether, we have bounded our maximization target from above and from below by $\text{AMIP}(k) + O\left(\frac{k}{n}\right) = \left(1 \pm O\left(\frac{1}{\sqrt{\log(n)}}\right)\right) \text{AMIP}(k)$, completing our proof.

$\square$

*Proof of Claim F.19.* If $k_j < \frac{n_j}{\nu} \ll n_j$, then we use the bound

$$\left|c_j^{\pm}(k_j)\right| < \frac{k_j^2}{n_j - k_j} \max_{i \in [n]} \left\{|R_i|\right\} \max_{i \in [n]} \left\{\left|e^{\intercal} \widetilde{\Sigma}^{-1} \widetilde{X}_i\right|\right\} = \widetilde{O}\left(\frac{k_j}{\nu n}\right) = o\left(\frac{k_j}{n}\right).$$

Next, note that due to Claim F.10 (which states that with very high probability the empirical residuals $R_i$ is close to the ground truth residual $(R_{\text{gt}})_i$), we have that with very high probability for all $i$:

$$\left|R_i - (R_{\text{gt}})_i\right| = o(1) < 1.$$

From our assumption that the ground truth residuals were normally distributed, we have that

$$\Pr_{(R_{\text{gt}})_i \sim \mathcal{N}(0,1)}\left[\left|(R_{\text{gt}})_i\right| > \tau - 1\right] < \frac{1}{2\nu'}.$$

Combining the two, along with the Hoeffding bound which promises us that with very high probability

$$\left|\{i \in B_j \mid |R_i| > \tau - 1\}\right| \leq n_j \Pr_{(R_{\text{gt}})_i \sim \mathcal{N}(0,1)}\left[\left|(R_{\text{gt}})_i\right| > \tau - 1\right] \pm \widetilde{O}\left(\sqrt{n_j}\right),$$

we have that with very high probability

$$\left|\{i \in B_j \mid |R_i| > \tau\}\right| < \left|\left\{i \in B_j \mid \left|\widetilde{R}_i\right| > \tau - 1\right\}\right| = \frac{n_j}{\nu'}.$$

We bound $\left|\left\{i \in B_j \mid \left|e^{\intercal} \widetilde{\Sigma}^{-1} \widetilde{X}_i\right| > \frac{\tau}{n}\right\}\right|$ in much the same manner, by utilizing Claim F.8 which states that with very high probability $\left|e^{\intercal} \widetilde{\Sigma}^{-1} \widetilde{X}_i - \frac{e^{\intercal} X_i}{n}\right| = o\left(\frac{1}{n}\right)$, and our assumption that the $X_i$ are very well behaved, which shows that there can't be too many samples in a given bucket for which $|e^{\intercal} X_i| > \text{polyloglog}(n)$.

$\square$

### F.4.2 INDIRECT CONTRIBUTIONS TO THE HIGHER ORDER TERMS

**Indirect Contributions to the Covariance Shift Term** Recall from Section A.3 that we bound the covariance shift as

$$\max\left\{\lambda\left(\hat{\Sigma}_S^{-1}\right)\right\} \leq \frac{1}{1 - \max\left\{\lambda\left(\widetilde{\Sigma}_T\right)\right\} - \max_{k_1+\cdots+k_m=k}\left\{\sum_{j\in[m]} \frac{1}{n_j-k_j}\left\|\sum_{i\in T\cap B_j} \widetilde{\Sigma}^{-1/2}\widetilde{X}_i\right\|\right\}} .$$

Each component in the denominator is bounded separately by running an MSN-bounding algorithm. The first MSN is run on the Gram matrix $G_{X\otimes X}$ whose $i, j$ entry is:

$$\left(\widetilde{X}_i^\mathsf{T}\widetilde{\Sigma}^{-1}\widetilde{X}_j\right)^2 .$$

From Claim F.2, the regression $\widetilde{X}, \widetilde{Y}$ is well-behaved, allowing us to use the same analysis as in Claim E.2 to bound the output of this MSN by

$$M_{X\otimes X}(k) = \widetilde{O}\left(\sqrt{k\frac{d^2}{n^2} + k^2\frac{d}{n^2}}\right) .$$

Similarly, we define $M_j$ to be the MSN bound achieved by RTI on the Gram matrix

$$G_j[i_1, i_2] = \widetilde{X}_{i_1}^\mathsf{T}\widetilde{\Sigma}^{-1}\widetilde{X}_{i_2} .$$

Claims F.5 and F.6 promise us that with very high probability the largest diagonal entry of this Gram matrix is at most $\widetilde{O}\left(\frac{d}{n}\right)$ and its largest off-diagonal entry is at most $\widetilde{O}\left(\frac{\sqrt{d}}{n}\right)$. Therefore, the resulting MSN bounds are at most

$$M_j(k_j) = \widetilde{O}\left(\sqrt{\frac{k_jd + k_j^2\sqrt{d}}{n}}\right) .$$

Recall that for our actual OHARE bound, we also utilize the symmetry that allows us to replace $M_j(k_j)$ with $\overline{M}_j(k_j) = \min\{M_j(k_j), M_j(n_j - k_j)\}$.

For all $k_j \leq \frac{n_j}{2}$, we have

$$\frac{\overline{M}_j(k_j)^2}{(n_j - k_j)k_j} \leq \frac{M_j(k_j)^2}{(n_j - k_j)k_j} = \widetilde{O}\left(\frac{d + k_j\sqrt{d}}{n(n_j - k_j)}\right) = \widetilde{O}\left(\frac{\sqrt{d}}{n}\right) ,$$

and for all $k_j > \frac{n_j}{2}$, we have

$$\frac{\overline{M}_j(k_j)^2}{(n_j - k_j)k_j} \leq \frac{M_j(n_j - k_j)^2}{(n_j - k_j)k_j} = \widetilde{O}\left(\frac{\sqrt{d}}{n}\right) .$$

Therefore, for all $k < k_{\text{threshold}}$

$$\max_{k_1+\cdots+k_m=k}\left\{\sum_{j\in[m]} \frac{\overline{M}_j(k_j)^2}{n_j - k_j}\right\} = \widetilde{O}\left(\frac{\sqrt{d}k}{n}\right) = o\left(M_{X\otimes X}(k)^2\right) = o(1) .$$

Therefore, in this regime the covariance shift term does not contribute a factor of more than 3.

**Indirect Contributions to the $XR$ Term** Recall from Section A.3 that the residual contributions to the indirect XR term are bounded by

$$\rho_j(k_j) \stackrel{\text{def}}{=} \min\left\{\max_{T_j\in\binom{B_j}{k_j}}\left\{\sum_{i\in T_j} |R_i|\right\}, \max_{T_j\in\binom{B_j}{k_j}}\left\{\sum_{i\in B_j\setminus T_j} |R_i|\right\}\right\} ,$$

where $R_i$ are the empirical residuals.

With an analysis similar to the one above, we can utilize Claim F.10 which states that with very high probability $\max_{i \in [n]} \{|R_i|\} = \widetilde{O}(1)$ to show that with very high probability

$$\frac{\rho_j(k_j)}{\sqrt{k_j(n_j - k_j)}} = \widetilde{O}(1) .$$

Combining this with our bound on $\frac{\overline{M}_j(k_j)}{\sqrt{k_j(n_j-k_j)}}$, we have

$$\max_{k_1 + \cdots + k_m = k} \left\{ \sum_{j \in [m]} \frac{\overline{M}_j(k_j)\rho_j(k_j)}{n_j - k_j} \right\} = \widetilde{O}\left( \sqrt{\frac{\sqrt{d}}{n}} k \right) .$$

Note that this is also smaller than the bound we proved for the direct contribution to the XR term in Section E.5,

$$M_{XR} = \widetilde{O}\left( \sqrt{\frac{kd}{n}} + \sqrt{\frac{\sqrt{d}}{n}} k \right) ,$$

and which we can apply after reaveraging due to Claim F.2 which states that $\widetilde{X}, \widetilde{Y}$ is an ACRE-friendly with very high probability.

**Indirect Contributions to the $XZ$ Term**   Finally, recall from Section A.3 that the $Z$ component of the indirect effect on the $XZ$ term was bounded by

$$\zeta_j(k_j) = \min \left\{ \max_{T_j \in \binom{B_j}{k_j}} \left\{ \sum_{i \in T_j} |Z_i| \right\}, \max_{T_j \in \binom{B_j}{k_j}} \left\{ \sum_{i \in B_j \setminus T_j} |Z_i| \right\} \right\} ,$$

where $Z_i = e^\intercal \widetilde{\Sigma}^{-1} \widetilde{X}_i$

Using the same analysis as above and Claim F.8 which states that with very high probability $|Z_i| = \widetilde{O}\left(\frac{1}{n}\right)$ for all $i \in [n]$, we have that with very high probability

$$\frac{\zeta_j(k_j)}{\sqrt{k_j(n_j - k_j)}} = \widetilde{O}\left(\frac{1}{n}\right) .$$

Combining this with our bound on $M_j(k_j)$, we have

$$\max_{k_1 + \cdots + k_m = k} \left\{ \sum_{j \in [m]} \frac{\overline{M}_j(k_j)\zeta_j(k_j)}{n_j - k_j} \right\} = \widetilde{O}\left( \sqrt{\frac{\sqrt{d}}{n^2}} k \right) .$$

As before, this is smaller than our bound on the direct effect

$$M_{XZ} = \widetilde{O}\left( \sqrt{\frac{kd}{n^2}} + \frac{k^2 \sqrt{d}}{n^2} \right) .$$

**Putting it all together**   In Claim F.18 and over the last few paragraphs, we have bounded all the individual terms that go into generating the OHARE bounds. In particular, we have shown that

$$U_k, L_k = \underbrace{\text{First Order}}_{\left(1 \pm \left(\frac{1}{\sqrt{\log(n)}}\right)\right) \times \text{AMIP}(k)} \pm \frac{|\text{Direct XR} + \text{Indirect XR}| \times |\text{Direct XZ} + \text{Indirect XZ}|}{1 - \text{Direct CS} - \text{Indirect CS}} =$$

$$= \left(1 \pm \left(\frac{1}{\sqrt{\log(n)}}\right)\right) \times \text{AMIP}(k) \pm \widetilde{O}\left(\frac{kd}{n^{3/2}} + \frac{k^2\sqrt{d}}{n^{3/2}}\right) =$$

$$= \left(1 \pm \left(\frac{1}{\sqrt{\log(n)}}\right)\right) \times \text{AMIP}(k) ,$$

for all $k \leq k_{\text{threshold}}$, concluding our proof of Theorem F.1.

## G    ONE-HOT ENCODINGS ARE ALMOST BRITTLE

In this section, we will prove our claim from the introduction that datasets with one-hot encodings are arbitrarily close to being extremely brittle. In particular, we will show that

**Claim G.1.** *Let $X \in \mathbb{R}^{n \times d}$ be an array of features and let $Y \in \mathbb{R}^n$ be some labels, such that one of the features is $0$ on all but $k_{\text{bucket}} < n - d$ of the samples. In other words, for some set $S \subseteq [n]$ of size $|S| = n - k_{\text{bucket}}$, we have*

$$\forall i \in S \quad X_{i,d} = 0$$

*Then, for all $\gamma \in \mathbb{R}^{d-1}$ and for all $c > 0$, there exists a linear regression $X', Y'$ such that $\|X' - X\| + \|Y' - Y\| < c$, and $OLS\left(X'_S, Y'_S\right)_{[d-1]} = \gamma$.*

The reason regressions become so close to brittle is that with one of the features being always $0$, we can make a very small change to its value, in a way that creates very strong correlations.

*Proof of Claim G.1.*  The proof of Claim G.1 is relatively simple.

First, we want to ensure that the original regression problem has no other degeneracies. We do this to ensure that the resulting OLS has a unique solution.

Let $V \subseteq \mathbb{R}^{n - k_{\text{bucket}}}$ be the linear space spanned by the columns of $X_{S,[d-1]}$ and $Y_S$. $V$ is spanned by $d$ vectors. Therefore, $\dim V \le d$.

If $\dim V = d$, we can skip this step, and if this inequality is strict we say that $X_{S,[d-1]}, Y_S$ are degenerate. If $\dim V < d$, it is easy to see that almost all $X'_{S,[d-1]}, Y'_S$ in $B_{c/2}\left(X_{S,[d-1]}, Y_S\right)$ (i.e. in the ball of radius $c/2$ around the original regression) are non-degenerate. Therefore, let $X'_{S,[d-1]}, Y'_S$ be such a non-degenerate pair.

Now, consider the vector $R = Y'_S - X'_{S,[d-1]}\gamma$. This is the residual vector for the linear model $Y'_S \approx X'_{S,[d-1]}\gamma$, and by our assumption that $X'_{S,[d-1]}, Y'_S$ are non-degenerate, $R \neq 0$ and is not in the span of the columns of $X'_{S,[d-1]}$.

It is only left to decide the values of $X'_{S,d}$. Setting $X'_{S,d} = \frac{c}{2\|R\|}R$, our regression has a perfect fit on the samples in $S$

$$Y'_S = X'_S \begin{pmatrix} \gamma \\ \frac{2\|R\|}{c} \end{pmatrix}$$

By our construction, $\Sigma = (X'_S)^\mathsf{T} X'_S$ is full rank, making this OLS solution unique.    □

