# OpenReview forum: "Robustness Auditing for Linear Regression: To Singularity and Beyond"
_ICLR.cc/2025/Conference — ICLR 2025 Poster_

### Official Review · Reviewer_UrpZ · 2024-11-02

**Soundness:** 2
**Presentation:** 2
**Contribution:** 2
**Rating:** 5
**Confidence:** 3

**Summary:**

The paper presents two algorithms, ACRE (Algorithm for Certifying Robustness Efficiently) and OHARE (One-Hot aware Algorithm for certifying Robustness Efficiently), for certifying the robustness of Ordinary Least Squares (OLS) regression to sample removal in the dataset with upper and lower bounds. The first algorithm is shown to work for continuous features and the second one with a mixture of categorical and one-hot encoded features.
Given a dataset with features $X_1,...,X_n \in \mathbb{R}^d$ and labels $Y_1,...,Y_n \in \mathbb{R}$, the algorithms provide bounds on how much the OLS coefficients can change when removing $k$ samples. The authors call the maximal discrepancy between the OLS with full dataset and the one where $k$ data points are removed $\Delta_k(e)$ along the direction $e$.

The algorithms provide upper and lower bounds $U_k, L_k$ such that $L_k \leq \Delta_k(e) \leq U_k$, where $\Delta_k(e)$ represents the maximum possible change in the projection of $\beta$ onto direction $e$ when removing $k$ samples. The bounds are proven to be (logarithmically) tight under mild distributional assumptions. One main characteristic of the algorithm is Minimal Subset-sum Norm bounding.

The authors provide some empirical analysis of the efficiency of these algorithms and an explanation of the correctness of ACRE.
The analysis is based on the decomposition of $\beta - \beta_S$ into linear order and higher-order terms where $\beta_S$ is the OLS estimator after the removal of samples.

**Strengths:**

* The authors provide an implementation in their anonymous repository. The algorithm proposed by them is able to run on consumer grade machines for datasets of sizes of order $n = 10^{4}$.
* The main text provides an excellent introduction to ACRE, explaining the idea behind its definition. The connection to MSN (Maximal Subset-sum Norm) is clearly explained.

**Weaknesses:**

* The clarity of the paper is a central issue. In Section 3, the intuition for the actual definition of the algorithm explains the setting and assumptions under which the result is obtained, and the authors believe the algorithm's work is not precise.
The connection between theoretical guarantees and practical performance needs to be better explained. for example Theorem1.2 and 1.3 could be tested in varying the numbers $n,d$ used for Table 1

* The process of obtaining the values in Table 1 needs to be clearly explained, and the table needs to be clearer. For example, while the algorithms output bounds $(U_k, L_k)$ for each $k$, how these translate to the final numbers reported needs to be clarified. The comparison methodology with baseline methods needs more detail. Any form of cross-validation missing from the experimental results

* Additional experiments can be conducted to strengthen the results.
Performance could be assessed under different types of data distributions using synthetic datasets with known ground truth.
Comparison could also be made with simple baselines (e.g., random sampling).

**Questions:**

* How sensitive are the bounds values $U_k$ and $L_k$ to the choice of MSN-bounding algorithm?
* How do the algorithms perform when the assumptions about data distribution are violated? In lines 252-253, the authors mention that the result is valid also without the exponential decaying tail assumption. Have the authors tried the algorithm on PowerLaw synthetic data?

---

> ### Author Response · Authors · 2024-11-19
>
> Thank you for your time and effort in reviewing our submission!
>
>
> ---
>
> #### **Clarity Issues in Section 3 and Explanation of Theorems 1.2 and 1.3**
>
> **Reviewer Comment:**
> The clarity of the paper is a central issue. In Section 3, the intuition for the actual definition of the algorithm explains the setting and assumptions under which the result is obtained, and the authors believe the algorithm's work is not precise. The connection between theoretical guarantees and practical performance needs to be better explained. For example, Theorems 1.2 and 1.3 could be tested by varying the numbers $n, d$ used for Table 1.
>
> **Response:**
> **Clarity:** We would appreciate any concrete suggestions on how to improve the clarity of the paper further.
>
> **Theory versus practical performance:** We agree that additional experiments can help elucidate the relationship between the theoretical guarantees we prove and our experimental results. To this end, we are working on synthetic experiments and aim to complete these before the end of the discussion period.
>
> That said, we would like to clarify that Table 1 reports results on real-world regressions. In these cases, we do not control $n$ and $d$, so it is not possible to test Theorems 1.2 and 1.3 by varying these values using the datasets shown in Table 1.
>
> ---
>
> #### **Clarity of Table 1 and Comparison with Baseline Methods**
>
> **Table 1 Methodology:**
> Thank you for highlighting this clarity issue. We will update the submission to better explain the methodology used to produce the values in Table 1.
>
> The bounds $(U_k, L_k)$ translate to the numbers in the "OHARE" column of Table 1 as follows:
>
> 1. We fix the sign of $e$ (WLOG) such that $\langle \beta, e \rangle > 0$.
> 2. For the bounds $(U_0, L_0), \dots, (U_n, L_n)$ produced by OHARE, we identify the smallest $k$ such that $U_k > \langle \beta, e \rangle$.
>
> For AMIP and KZC, both methods output candidate sets of samples to remove, one set for each value of $k$. The "AMIP" and "KZC" columns in Table 1 contain the size of the smallest such candidate set whose removal flips the sign of $\langle \beta, e \rangle$.
>
> **Comparison with Baseline Methods:**
> As noted above, we compare to AMIP and KZC using the size of the candidate sets they generate. The algorithm of Moitra and Rohatgi is computationally infeasible for datasets of the scale we consider, as its runtime is exponential in the data dimension $d$, and the algorithm of Freund and Hopkins yielded only trivial bounds ($k \geq 1$) for all of the datasets in this experiment.
>
> **Cross-validation:**
> We are unsure of what type of cross-validation the reviewer is referring to. Could you clarify? Cross-validation is not typically used in the robustness-auditing setting we consider.
>
> ---
>
> #### **Additional Experiments: Synthetic Datasets and Baselines**
>
> We are working on experiments with synthetic data, which will be added to the submission. We aim to complete these before the end of the discussion period.
>
> However, we would appreciate clarification on two points:
>
> 1. It is unclear how to construct synthetic datasets with known ground-truth values for $\Delta_k(e)$. Even if we know the ground-truth $\beta$, computing $\Delta_k(e)$ for sample removals in synthetic datasets is nontrivial.
> 2. Regarding random sampling: If by "random sampling" you mean removing small random subsets of data, this would generate candidate sets for removal, but such an approach does not provide the lower bounds our algorithms compute (i.e., the minimum number of samples that must be removed to achieve a desired effect).
>
> ---
>
> #### **Sensitivity of Bounds to MSN-Bounding Algorithm**
>
>
> The quality of the bounds $U_k$ and $L_k$ is indeed sensitive to the underlying MSN-bounding algorithm. There is a range of MSN-bounding algorithms available, each with trade-offs in performance and runtime:
>
> - Spectral algorithms yield moderately tighter bounds in synthetic datasets but are computationally expensive.
> - The RTI algorithm, which we used, strikes a balance between runtime efficiency, ease of implementation, and practical performance.
>
> We chose to focus on RTI in this work but leave the exploration of alternative MSN-bounding algorithms to future work.
>
> ---
>
> #### **Assumptions and PowerLaw Synthetic Data**
>
> We are currently conducting experiments on synthetic data to explore the performance of our algorithms under such scenarios. We plan to include these results in an updated version of the submission before the discussion period ends.
>
> ---

---

### Official Review · Reviewer_SWFy · 2024-11-04

**Soundness:** 2
**Presentation:** 2
**Contribution:** 2
**Rating:** 5
**Confidence:** 3

**Summary:**

The paper presents two algorithms, ACRE and OHARE, designed to audit the robustness of Ordinary Least Squares (OLS) regression models. This robustness is assessed by determining how removing a small subset of data points can alter the regression parameter vector. For detailed comments, please see strengths, weakness, questions section.

**Strengths:**

1) The paper addresses an interesting problem of auditing in linear regression.
2) Rigorous theoretical analysis using interesting proof techniques like bounds via maximal subset sum norm (MSN) or knapsack-style dynamic programming.
3) The authors have tried experiments on real datasets.

**Weaknesses:**

1) Line 22: Instead of saying hundreds of dimensions, the authors should specify the exact value of d and n. Later in the paper, the authors have specified that they assume n>>d. Hence, saying hundreds of dimensions sounds slightly misleading.
2) Line 135: The authors claim no assumptions whatsoever on X and Y. Aren’t you assuming well-behaved in Definition 1 and also iid? Also, see line 159 about assumptions.
3) Line 253: Authors claim distributions with heavier tails than sub-exponential can still be well-behaved.
While this is true, the authors should provide more information or intuition on which heavy-tailed distributions can specifically be well-behaved and which are not. For example, the counter-statement with some heavy-tailed distribution not being well-behaved can also be true.
4) Line 295: The authors believe that the error term 1/\sqrt(\log(n)) is nearly tight. Is there a formal proof for the tightness of the bound?
5) Line 322: The authors claim that loose bounds suffice. Why? Are the proposed bounds loose? Because the authors have also claimed the bounds to be nearly tight. This is slightly contradictory. What are the challenges in deriving tighter bounds?
6) Line 374: The authors bound the first term using a greedy algorithm. What will the computational complexity be for that? Is this greedy step just for theoretical analysis?
7) Line 390: The authors assume the max eigenvalue of a matrix which is a function of T (an unknown variable or variable of interest). This is slightly unconventional.
8) Line 399: Can the computational complexity be improved from O(n^2 d) to something better?
9) Line 440: Typo: direct -> direction
10) Line 441: The assumption on $e$ should be specified in the main theorem.
11) Line 497: The authors criticize existing works due to their reliance on semi-definite programming. Is the computational complexity of ACRE and OHARE better than existing works?

Although the contributions are noteworthy, it is slightly challenging to follow the paper:
1) It is tough to follow the story. The authors propose ACRE and then criticize it for proposing OHARE.
2) Theorem 1.2 and 1.3 are presented even before describing the respective methods.
3) This makes it tough to develop an intuitive understanding of theorems. For example, please explain the implication of bucket size from Theorem 1.3. What role does it play? What happens if all the bucket sizes are different or if all of them are the same?
4) It feels like the authors have tried to squeeze in a lot of information, breaking the story's flow.

**Questions:**

1) Line 79: The authors talk about crossing a decision boundary. In terms of the regression problem addressed in the paper, a decision boundary is slightly inappropriate. It suits better for classification problem.
2) Line 158: ACRE and OHARE are claimed to produce nearly matching upper and lower bounds. When do they (ACRE and OHARE) produce the same bounds, and when do they perform differently?
3) Line 267: ACRE and OHARE are proposed for n>>d. What happens for the more interesting case when n is comparable to d or a high dimensional setting?
4) Line 352: The norm is not specified in the equation. Is it Euclidean?
5) Line 385: The authors assume the knowledge of the population covariance matrix.  What is the approach when it is unknown?
6) Line 464: The authors use a greedy algorithm to obtain the bound. What is the computational complexity?

---

> ### Author Response · Authors · 2024-11-19
>
> Thank you for your time and effort in reviewing our submission!
>
> ---
> #### **Line 22**
>
> **Reviewer Comment:**
> Instead of saying hundreds of dimensions, the authors should specify the exact value of $d$ and $n$. Later in the paper, the authors have specified that they assume $n \gg d$. Hence, saying hundreds of dimensions sounds slightly misleading.
>
> **Response:**
> Thanks for pointing this out. We ran both the ACRE and OHARE algorithms on regressions with dimension $d = 200$ and $n = 30,000$ samples. We will amend the submission to include these exact numbers.
>
> The goal of this line is to highlight the practicality of our algorithm in comparison with algorithms such as MR22 and BP21, which are too computationally intensive to be run beyond 3 dimensions or tens of samples.
>
> ---
>
> #### **Line 135 vs Line 159: clarifying our assumptions**
>
>
> These lines refer to two separate theorems:
>
> - Theorem 1.1 states that the bounds produced by ACRE and OHARE are unconditionally *true*.
> - In Theorems 1.2 and 1.3, we show that under some assumptions, these bounds are also *tight*.
>
> This distinction is crucial, as it allows us to use the ACRE and OHARE algorithms even when we have no guarantees about the input.
>
> ---
>
> #### **Line 253: Clarify well-behaved heavy-tailed distributions**
>
>
> This is true— not all heavy-tailed distributions are well-behaved.
>
> Our goal in Theorems 1.2 and 1.3 is to analyze the ACRE and OHARE algorithms in reasonable cases. We highlight that the set of cases analyzed includes all sub-exponential and sub-Gaussian distributions but is not limited to them.
>
> For instance:
> - A distribution where the entries of each sample are drawn i.i.d. by taking the cube of a standard normal variable is well-behaved but not sub-exponential or sub-Gaussian.
> - Conversely, a distribution where each entry is drawn i.i.d. from a power-law distribution is not well-behaved.
>
> ---
>
> #### **Line 295: Tightness of the $1/\sqrt{\log(n)}$ term**
>
>
> We did not include a formal proof of this conjecture in the submission. A relatively simple analysis of the case where the continuous features $X_i$ are drawn iid from the unit sphere and the axis of interest $e$ is the 1st primary axis shows that this $1/\sqrt{\log(n)}$ term is asymptotically tight, and we will add this to the submission.
>
> ---
>
> #### **Line 322: loose bounds on MSN vs tight bounds on robustness**
>
>
> These statements refer to two different bounds.
>
> - ACRE utilizes **loose MSN bounds** to produce **nearly tight robustness bounds**.
> - This ability to leverage loose bounds is crucial because deriving tight or even nearly tight MSN bounds is believed to be computationally intractable.
>
> ---
>
> #### **Line 374: Computational complexity of the greedy algorithm**
>
> The first-order term is computed using the influence scores $\alpha_i = R_i \langle X_i, e \rangle$, each of which is a real number.
>
> To maximize $\sum_{i \in T} \alpha_i$ for $T \subset [n]$ of size $k$, we:
> 1. Sort the $\alpha_i$ in descending order.
> 2. Compute the cumulative sum of the top $k$ values.
>
> This process has a time complexity of $O(n \log(n))$, which is much smaller than the overall runtime of the algorithm ($\Theta(n^2 d  + n^2 \log(n))$).
>
> ---
>
> #### **Line 399: Improving computational complexity**
>
>
> The computational bottleneck in Algorithm 1 (ACRE) is the pairwise inner product computation $\langle X_i, X_j \rangle$ for all $i, j$.
>
> Significant runtime improvements would likely come at the cost of bound quality. Additionally, the current runtime is not a bottleneck for our experiments and is comparable to the $\Theta(nd^2)$ complexity of the OLS regression itself. We leave potential runtime improvements to future work.
>
> ---
>
> #### **Line 440: Typo**
>
> Thanks! We will correct this error.
>
> ---
>
> #### **Line 441: Assumptions in the main theorem**
>
> Thank you for pointing this out! We mention this assumption in Line 145 (right before Theorem 1.1), but we will make it more explicit in the main text and also include it in Theorem 1.3.
>
> ---

---

> > ### Author Response · Authors · 2024-11-19
> >
> > #### **Line 497: Comparison with SDP-based methods**
> >
> > **Reviewer Comment:**
> > The authors criticize existing works for reliance on semi-definite programming. Is the computational complexity of ACRE and OHARE better than existing works?
> >
> > **Response:**
> > Yes, the runtime of ACRE and OHARE is significantly faster than that of previous work.
> >
> > Computing the exact polynomial power of an SOS / SDP based algorithm typically requires a very deep analysis of the details of their SOS proofs. In particular, both BP21 and KKM18 do not state the runtime of their algorithm beyond the fact that it is polynomial in $n$ and $d$, so it is difficult to get a precise comparison with their algorithm.
> >
> > As a rough comparison, BP21 for instance rely on a degree $k \geq 4$ SOS, which involves solving a $\binom{n}{k}$-variable linear program, even writing the constraints for this linear program would require an extremely high memory complexity, and this gives a conservative runtime estimate of at least $n^{>12}$.
> >
> > ---
> >
> > #### **Improving clarity of presentation**
> >
> > **Reviewer Comment:**
> > It is slightly challenging to follow the paper:
> > - ACRE is proposed, then criticized in favor of OHARE.
> > - Theorems 1.2 and 1.3 are presented before describing the respective methods.
> >
> > **Response:**
> > Yes, the paper contains two new algorithms. Both are improve upon the state-of-the-art, and both are useful for different purposes in our experiments and beyond them.
> >
> > Given that ACRE is a building-block in the OHARE algorithm, it seems reasonable to present it first. That said, if you have suggestion for how to improve the presentation, they would be greatly appreciated!
> >
> > Regarding Theorems 1.2 and 1.3: It is common in the literature to state results before the pseudo-code. However, we will add clearer pointers to the sections that describe the algorithms to improve clarity.
> >
> > ---
> >
> > #### **Role of bucket sizes in Theorem 1.3**
> >
> > The bucket size assumptions are designed to ensure:
> > 1. That each bucket has at least $n^{\Omega(1)} \sqrt{d}$ samples so that means and covariances concentrate for each bucket.
> > 2. Each bucket has at most $0.49n$ samples so that no single bucket dominates the overall covariance (this is needed for Lemma E.12).
> >
> > Beyond these requirements, bucket size variation does not significantly impact $\frac{U_k}{L_k}$.
> >
> > ---

---

> > > ### Author Response · Authors · 2024-11-19
> > >
> > > #### **Line 79: Decision boundary terminology**
> > >
> > > **Reviewer Comment:**
> > > The authors talk about crossing a decision boundary. In terms of the regression problem addressed in the paper, a decision boundary is slightly inappropriate. It suits better for classification problems.
> > >
> > > **Response:**
> > > The ACRE and OHARE algorithms can be used to bound the change in a given OLS coefficient.
> > >
> > > The regressions we considered in this paper are taken from econometrics studies, where there is a clear decision boundary.
> > >
> > > In applications without such a decision boundary, these tools may still be used to bound changes in the regression coefficients.
> > >
> > > ---
> > >
> > > #### **Line 158: When do ACRE and OHARE produce matching bounds?**
> > >
> > > **Reviewer Comment:**
> > > ACRE and OHARE are claimed to produce nearly matching upper and lower bounds. When do they (ACRE and OHARE) produce the same bounds, and when do they perform differently?
> > >
> > > **Response:**
> > > The ACRE algorithm is designed to analyze regressions with continuous features (lines 132–141), where it produces nearly tight bounds.
> > > The OHARE algorithm, on the other hand, is designed to analyze regressions that include both continuous features and a discrete feature (lines 142–146), where it also produces nearly tight bounds.
> > >
> > > ---
> > >
> > > #### **Line 267: Performance in $n \approx d$ or high-dimensional settings**
> > >
> > > **Reviewer Comment:**
> > > ACRE and OHARE are proposed for $n \gg d$. What happens for the more interesting case when $n$ is comparable to $d$ or in a high-dimensional setting?
> > >
> > > **Response:**
> > > Which of the $n \leq d$ or $n \gg d$ regimes is more interesting is perhaps in the eye of the beholder.
> > >
> > > There are many applications where $n \gg d$, and providing robustness guarantees even for this "classical" regime remains challenging.
> > >
> > > When $n \leq d$, traditional OLS is generally not robust. Any robustness analysis for this regime would likely focus on LASSO or ridge regressions. Given the added complexity of these methods, it makes sense from an algorithm design perspective to first address the simpler case of OLS.
> > >
> > > ---
> > >
> > > #### **Line 352: Norm specification**
> > >
> > > **Reviewer Comment:**
> > > The norm is not specified in the equation. Is it Euclidean?
> > >
> > > **Response:**
> > > Yes, this is an $L_2$ norm. We will clarify this in the submission.
> > >
> > > ---
> > >
> > > #### **Line 385: Population covariance matrix assumption**
> > >
> > > **Reviewer Comment:**
> > > The authors assume knowledge of the population covariance matrix. What is the approach when it is unknown?
> > >
> > > **Response:**
> > > Neither the ACRE nor the OHARE algorithms assume knowledge of the covariance of the sample distribution (or even that the samples were drawn from a distribution).
> > >
> > > The matrix $\Sigma_S$ refers to the **empirical covariance** of the samples ($\sum_{i \in S} X_i X_i^T$). As noted in lines 300–305, we normalize the samples so that their empirical second moment is the identity matrix.
> > >
> > > ---
> > >
> > > #### **Line 464: Computational complexity of the greedy algorithm**
> > >
> > > **Reviewer Comment:**
> > > The authors use a greedy algorithm to obtain the bound. What is the computational complexity?
> > >
> > > **Response:**
> > > The computational complexity of the RTI algorithm is $O(n^2 \log(n))$.
> > >
> > > The algorithm is described in lines 468–480. It includes $2n$ sort and cumulative sum operations on lists of length $n$. Each sort operation takes $O(n \log(n))$, resulting in an overall runtime of $O(n^2 \log(n))$.
> > >
> > > #### **Line 390: Unconventional assumption on the max eigenvalue**
> > >
> > > **Reviewer Comment:**
> > > The authors assume the max eigenvalue of a matrix which is a function of $T$ (an unknown variable or variable of interest). This is slightly unconventional.
> > >
> > > **Response:**
> > > Thank you for pointing this out! We will add a clarification to our submission:
> > >
> > > The very next lines (391–402) describe how our ACRE algorithm utilizes the MSN-bounding algorithm to bound the maximal eigenvalue of $\Sigma_T$ for any given $k$. When the bound produced in this step exceeds 1, our algorithm returns $U_k, L_k = \pm \infty$ for this $k$.

---

### Official Review · Reviewer_CjMq · 2024-11-06

**Soundness:** 4
**Presentation:** 4
**Contribution:** 3
**Rating:** 8
**Confidence:** 4

**Summary:**

This paper studies data-based robustness for linear regression. Given an OLS solution that predicts $Y$ from $X$ (based on dataset $(X_1, Y_1), \dots, (X_n, Y_n)$), target direction $e$, and $1 \leq k \leq n$, they estimate $\Delta_k(e)$, the maximum possible change along the direction $e$ in $\beta$, the solution to the OLS, that can be achieved by removing $k$ points from the dataset. In essence, $\Delta_k(e)$ measures the robustness of a linear regression to potential training-time adversarial attacks.

Their main contribution is an algorithm, ACRE, which computes upper and lower bounds $U_k$ and $L_k$ for $\Delta_k(e)$. ACRE operates by bounding the difference between $\beta_S$, the value of $\beta$ after restricting to a subset, $S$ of the data, and $\beta$ by expressing it with two terms: first, a combination of the residuals of the original regression, and second, a higher order term, that factors in matrix terms based on the dataset $X$.

Next, ACRE bounds each of these terms by reducing them to Maximum Subset Norm (MSN) problems. The MSN problem consists of computing a subset of vectors from a set of vectors with the largest possible $L_2$-norm of their sum. This problem, while still posing significant combinatorial difficulty, is subsequently solved by a Refined Triangle Inequality (RTI)-based method which essentially uses a greedy approach.

Their first theoretical result shows that the upper and lower bounds absolutely hold for any data-set. The quality of the solution, however, is based on the difference between $U_k$ and $L_k$. They then show that for exponentially tailed distributions, the ratio $\frac{U_k}{L_k}$ is at most $1 + O \left( \frac{d + k \sqrt{d}}{n} \right)$.

This paper also includes an adaptation of their algorithm for regressions that involve categorical data called OHARE. For this algorithm, they show that for datasets satisfying certain technical properties, they can obtain a bound of $\frac{U_k}{L_k}$ of at most $1 + O\left(\frac{1}{\sqrt{\log n}}\right)$. They then experimentally validate this algorithm by demonstrating its effectiveness in the Nightlights study, where they obtain a provable lower bound on the minimum number of examples that need to be changed in order to reverse the main findings of the study.

They also include audits of 13 other linear regressions.

**Strengths:**

This paper is well written and proposes a nice solution to a simple fundamental problem. Due to the challenging nature of this problem, their results still hinge on a reduction to an NP-complete problem. However, they are able to give performance bounds on their algorithm's tightness (both theoretically and experimentally), and crucially their algorithm improves over the naive approach (trying all subsets of size $k$) enough to provide meaningful audits of how robust the results from real-life linear regressions are.

**Weaknesses:**

There is, of course, a gap between the theoretical bounds on $\frac{U_k}{L_k}$ and its behavior in practice (probably due to the nature of the assumptions needed to prove bounds on this quantity). Additionally, this algorithm is probably still not feasible for very large values of $k$. However, I do view this last weakness to be quite thoroughly addressed with the comprehensive empirical evaluation provided on real regressions.

**Questions:**

Do you have a comment on how to expand your theoretical tightness results to broader types of assumptions on $X$ and $Y$. In particular, can you show a more general (but perhaps weaker) bound on $\frac{U_k}{L_k}$.

---

> ### Author Response · Authors · 2024-11-19
>
> Thank you for your time and effort to review our submission!
>
>
> ---
>
> #### **Expanding theoretical tightness results to broader assumptions**
>
> **Reviewer Comment:**
> Do you have a comment on how to expand your theoretical tightness results to broader types of assumptions on $X$ and $Y$? In particular, can you show a more general (but perhaps weaker) bound on $\frac{U_k}{L_k}$?
>
> **Response:**
> We believe that weakening our assumptions on the tails of well-behaved distributions to a bound on a sufficiently high moment will suffice. However, deriving such a general bound would require more careful analysis, which we felt was beyond the scope of this paper.
>
> That said, we are conducting a numerical experiment on synthetic power-law data, which we expect to support our hypothesis. We aim to complete this before the end of the discussion period.

---

### Author Response · Authors · 2024-11-25
**Change Log**

1. **Lines 1284–1428:** Added Appendix D to test the ACRE algorithm on synthetic datasets. These numerical experiments evaluate the tightness of the ACRE bounds with covariates drawn from normal and power-law distributions. Results indicate that, for both normally distributed covariates and those drawn from a power-law with a sufficiently high exponent, the scaling of $k_{\text{threshold}}$—the largest $k$ for which $U_k / L_k$ remains close to $1$—matches the predictions of Theorem 1.2, even though power-law distributions are not directly covered by the theorem.

2. **Lines 99–103:** Clearly stated the values of $n$ and $d$ considered in our experiments in the main text.

3. **Lines 139–142:** Clarified the distinction between ACRE and OHARE.

4. **Lines 139–149:** Added explicit pointers to the ACRE and OHARE algorithms before the theorem statements.

5. **Lines 201–205 and 1101–1144:** Improved the caption of Table 1 and added a pointer to Appendix C.1, which explains the methodology in detail.

6. **Lines 252–254:** Clarified the comparison between our well-behaved assumption and sub-Gaussian/sub-exponential distributions.

7. **Lines 278–283:** Clearly stated the assumptions of Theorem 1.3 regarding the direction of interest.

8. **Lines 374–376:** Expanded the explanation of the runtime for the greedy algorithm used in analyzing the first-order term.

9. **Lines 380–382:** Explained how the $\Sigma_T < 1$ assumption is verified.

---

### Meta-Review · Area_Chair_2QVY · 2024-12-25

**Metareview:**

The paper considers the following very natural and simple problem regarding robustness of linear regression: Given n points, and a budget k for points to remove and a direction e, what is the maximum change that you can make along this direction by removing k points from the training data. This is a very natural robustness question which has only recently been studied. The authors introduce an algorithm to compute upper and lower bounds on this measure. There is also substantial empirical evaluation.

**Additional Comments On Reviewer Discussion:**

The reviewers did not respond to the detailed responses submitted by the authors; this was particularly unfortunate for the negative reviews. In the end, I read several parts of the reviewer and sided with the positive reviewer.

---

### Decision · Program_Chairs · 2025-01-22

Accept (Poster)